

# Noncommutative resolutions and CICY quotients from a non-Abelian GLSM

Johanna Knapp[⋆] and Joseph McGovern[†]

School of Mathematics and Statistics, University of Melbourne, Parkville, VIC 3010, Australia

⋆ johanna.knapp@unimelb.edu.au , † mcgovernjv@gmail.com

## Abstract

We discuss a one-parameter non-Abelian GLSM with gauge group $(U(1)\times U(1)\times U(1))\rtimes\mathbb{Z}_3$ and its associated Calabi-Yau phases. The large volume phase is a free $\mathbb{Z}_3$-quotient of a codimension 3 complete intersection of degree-$(1,1,1)$ hypersurfaces in $\mathbb{P}^2 \times \mathbb{P}^2 \times \mathbb{P}^2$. The associated Calabi-Yau differential operator has a second point of maximal unipotent monodromy, leading to the expectation that the other GLSM phase is geometric as well. However, the associated GLSM phase appears to be a hybrid model with continuous unbroken gauge symmetry and cubic superpotential, together with a Coulomb branch. Using techniques from topological string theory and mirror symmetry we collect evidence that the phase should correspond to a non-commutative resolution, in the sense of Katz-Klemm-Schimannek-Sharpe, of a codimension two complete intersection in weighted projective space with 63 nodal points, for which a resolution has $\mathbb{Z}_3$-torsion. We compute the associated Gopakumar-Vafa invariants up to genus 11, incorporating their torsion refinement. We identify two integral symplectic bases constructed from topological data of the mirror geometries in either phase.

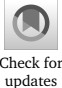

# 1  Introduction and summary

Calabi-Yau manifolds and their moduli spaces have played a central role in string theory and the associated mathematics for more than three decades. More recently, it has been appreciated that new phenomena and correspondences can occur when one goes beyond the well-studied framework of smooth complete intersections in toric ambient spaces. A valuable tool is Witten's gauged linear sigma model (GLSM) [1]. It allows to explore the stringy Kähler moduli space beyond the boundaries of a Kähler cone associated to a Calabi-Yau. In this way one can establish connections between Calabi-Yaus that are located at different limiting regions of a shared moduli space. Physically, the Calabi-Yaus are target spaces for non-linear sigma models appearing as low-energy effective theories, or phases, of GLSMs at different limiting values of the FI-theta parameters, which get identified with the complexified Kähler moduli. Moreover, the Coulomb branch of the GLSM encodes information about singularities in the moduli space. Mirror symmetry combines the different chambers of Kähler structure moduli space into the single mirror complex structure moduli space, which allows one to analyse the different regions of moduli space using differential equations which encode this same singularity structure.

Not every phase of a Calabi-Yau GLSM has to be a non-linear sigma model with a smooth target geometry. Other well-studied examples are for instance Landau-Ginzburg orbifolds. More generic phases are still rather poorly understood. Typically, one expects them to be some type of hybrid theory, i.e. a Landau-Ginzburg model fibred over a geometric base, but even more exotic configurations can occur. Of particular interest are GLSMs that have more than one phase that is geometric in some suitable sense. It has been shown in various examples that GLSMs can have two Calabi-Yau phases that are not necessarily birational to each other. This has connections to active research areas in mathematics such as non-commutative algebraic geometry and homological projective duality. The torsion-refinement of the Gopakumar-Vafa formula [2] turns out to be a very useful tool for analysing these cases. An important example

in the context of Abelian GLSMs was studied in [3], where it was shown that a fairly simple complete intersection of quadrics in $\mathbb{P}^7$ shares its moduli space with a non-commutative resolution of a double cover of $\mathbb{P}^3$, branched over a singular octic hypersurface in $\mathbb{P}^3$. This singular variety, which is not a smooth manifold, is still an acceptable target space for a supersymmetric nonlinear sigma model because nonsingular geometries are not a prerequisite for nonsingular physics. This construction and its generalisations have recently been studied by Katz-Klemm-Schimannek-Sharpe (KKSS) [4], see also [5,6].

The mechanism by which such a singular geometry remains a valid NLSM target is through "fractional" B-fields, which in [4] are argued to generalise the notion of discrete torsion [7,8]. Fractional B-fields can be supported on exceptional curves, that are torsion in homology, on a non-Kähler resolution of the singular geometry, and in the singular limit where the exceptional curves shrink the effects of this B-field persist in the worldsheet theory. This is the string-theoretic realisation of noncommutative resolution, which the authors of [4] advanced and then used to define torsion-refined Gopakumar-Vafa invariants by an analysis of topological string theory in the presence of these fractional B-fields. This follows on from the work [2], which introduced this torsion refinement.

Another source of non-birational Calabi-Yaus sharing the same moduli space are non-Abelian GLSMs. A first construction was provided by Hori and Tong [9], who gave a physical realisation of the Pfaffian-Grassmannian correspondence first observed by Rødland [10] and later formulated in terms of homological projective duality [11]. Another pair of non-birational Calabi-Yaus is due to Hosono and Takagi [12] and was described in terms of GLSMs in [13]. More constructions have been given since then [14–19]. Typically one geometric phase realises its geometry through purely perturbative means, as the vanishing locus of a set of polynomial equations given by the critical locus of the GLSM superpotential, while the other phase realises its geometry through a nonperturbative mechanism enabled by the nonAbelian dynamics of a strongly coupled phase.

The aim of this work is to study a new pair of one-parameter Calabi-Yau threefolds that share the same moduli space, which generalise the known constructions in a non-trivial way. One can search for such examples by investigating the set of GLSMs that realise one-parameter Calabi-Yaus which are not complete intersections in toric ambient spaces. A well-studied source of examples of this are free quotients of complete intersections in products of projective spaces. The GLSM for a complete intersection in a product of projective spaces is Abelian, with gauge group $U(1)^m$ where $m$ is the number of $\mathbb{P}^n$ factors in the ambient variety. But to realise freely acting quotients by symmetry groups $\mathbb{Z}_M$ that cyclically permute $M$ of the ambient $\mathbb{P}^n$, one must replace this gauge group with the nonAbelian group $U(1)^m \rtimes \mathbb{Z}_M$. This provides a natural generalisation of the Hosono-Takagi examples with their gauge group $U(1)^2 \rtimes \mathbb{Z}_2$, as described in [13].

Complete Intersection Calabi-Yau threefolds (in products of projective spaces), or CICYs, were the first substantial database of Calabi-Yau threefolds assembled [20]. The set of freely acting symmetries of CICYs that descend from automorphisms of the ambient product of projective spaces was classified in [21], and all of their Hodge numbers are computed by the means of [22]. Prior to the complete solutions of these latter two works, [23–25] impressed the significance of systematically studying these quotient threefolds and obtaining their Hodge numbers. The tables of [26] collect, from the previously mentioned and further additional sources, Calabi-Yaus with small Hodge numbers which one can peruse with a mind to finding new GLSMs to study.

In general, one should not expect to find a second geometry in the moduli space of a CICY quotient. To find potential candidates for this, it is useful in the one-parameter case to study the associated Calabi-Yau differential operators [27,28]. In all known examples with two geometries the associated differential operator has two points of maximal unipotent monodromy

(MUM points). Searching the relevant databases, and also informed by the considerations in [29], we were led to the following Calabi-Yau:

$$Y \cong \begin{matrix} \mathbb{P}^2 \\ \mathbb{P}^2 \\ \mathbb{P}^2 \end{matrix} \begin{bmatrix} 1 & 1 & 1 \\ 1 & 1 & 1 \\ 1 & 1 & 1 \end{bmatrix}^{h^{1,1}=1, h^{2,1}=16}_{/\mathbb{Z}_3} . \tag{1}$$

This notation indicates that we take the intersection of three hypersurfaces in $\mathbb{P}^2 \times \mathbb{P}^2 \times \mathbb{P}^2$, each hypersurface is the vanishing locus of an equation that is degree $(1, 1, 1)$ in the homogeneous coordinates of $\mathbb{P}^2 \times \mathbb{P}^2 \times \mathbb{P}^2$, and we quotient by a freely acting $\mathbb{Z}_3$ symmetry. This $\mathbb{Z}_3$ symmetry of the complete intersection is induced by the symmetry of the ambient $\mathbb{P}^2 \times \mathbb{P}^2 \times \mathbb{P}^2$ that cycles the three $\mathbb{P}^2$ factors, which gives a freely acting symmetry of the intersection for suitable choices of defining polynomials. This Calabi-Yau and its simply connected cover have recently been discussed in the context of type IIB flux compactifications [30].

The Picard-Fuchs operator for the mirror manifold of (1) has AESZ number 17.

$$\begin{aligned} \mathcal{L}^{\text{AESZ17}} &= 25\theta^4 - 15\varphi(5 + 30\theta + 72\theta^2 + 84\theta^3 + 51\theta^4) \\ &\quad + 6\varphi^2(15 + 155\theta + 541\theta^2 + 828\theta^3 + 531\theta^4) \\ &\quad - 54\varphi^3(1170 + 3795\theta + 4399\theta^2 + 2160\theta^3 + 423\theta^4) \\ &\quad + 243\varphi^4(402 + 1586\theta + 2270\theta^2 + 1368\theta^3 + 279\theta^4) - 59049\varphi^5(1 + \theta^4), \end{aligned} \tag{2}$$

$$\theta = \varphi \frac{\mathrm{d}}{\mathrm{d}\varphi}.$$

As can be seen by collecting like powers of $\varphi$ and inspecting the polynomials in $\theta$ that multiply the extreme powers $\varphi^0$ and $\varphi^5$, this differential operator has a MUM point at $\varphi = \infty$ in addition to the expected one at $\varphi = 0$. This Picard-Fuchs operator was first studied in [31], where the simply connected cover of (1) was considered. Monodromies for solutions obtained as expansions about $\varphi = 0$ have been analysed in [32]. Below we reproduce the Riemann symbol for AESZ17.

Table 1: Riemann symbol for AESZ17.

| 0 | $\frac{1}{27}$ | $\frac{i}{3\sqrt{3}}$ | $\frac{-i}{3\sqrt{3}}$ | $\frac{5}{9}$ | $\infty$ |
|---|---|---|---|---|---|
| 0 | 0 | 0 | 0 | 0 | 1 |
| 0 | 1 | 1 | 1 | 1 | 1 |
| 0 | 1 | 1 | 1 | 3 | 1 |
| 0 | 2 | 2 | 2 | 4 | 1 |

This example (1) has seen additional previous study. The mirror variety's moduli space was found to possess a rank-two attractor point in [29,33], but more relevant to our current paper is the realisation in those works that it is highly nontrivial to construct an integral symplectic basis by making a linear transformation on a Frobenius basis of solutions expanded about $\varphi = \infty$. The argument of [29,33], which we recall and add to in §3.3, runs as follows. Assuming that there is a smooth mirror geometry associated to both MUM points $\varphi = 0, \infty$, then they must both have the same Euler characteristic because mirror symmetry exchanges Hodge numbers. The Euler characteristic of (1) is computed to be $-30$. Now seek a change of basis matrix that acts on the Frobenius basis associated to $\varphi = \infty$, and appeal to the results of [34–36] that provide such a change of basis matrix whose entries are topological data of the mirror manifold based on the structure of the genus 0 prepotential. This allows one to read

off the ratio of the triple intersection number and the Euler characteristic, leading to a value of $-30/13$ for the triple intersection when $\chi = -30$ is imposed. Clearly an assumption must fail. We will resolve this using considerations of [4] that hold for noncommutative resolutions of singular Calabi-Yau threefolds.

We begin this paper with a study of the GLSM. To analyse the stringy Kähler moduli space of (1), we show that it can be realised as the "large volume" phase of a non-Abelian GLSM with gauge group $(U(1) \times U(1) \times U(1)) \rtimes \mathbb{Z}_3$. A Coulomb branch analysis confirms the location of three singular points at the phase boundary which coincide with the three points in the table above that have indices (0,1,1,2). Against our expectations, the other phase does not look like a geometry at all. Rather, we find a hybrid model: The vacuum manifold is a $\mathbb{P}^2$. It forms the base of a Landau-Ginzburg fibration with a cubic potential. To our knowledge, all the multiple-MUM models studied so far could, at intermediate energy scales, be interpreted as hybrid models with quadric potentials, meaning that these theories are massive. The geometry deep in the IR is then encoded in the properties of the mass matrix. This mechanism does not apply to our model. However, instead of the mass matrix, there is a rank three tensor governing the couplings of the cubic potential. A possible generalisation of the examples with quadratic potential would be to consider the hyperdeterminant of this tensor which determines loci where certain couplings vanish. Unfortunately, dealing with hyperdeterminants poses computational challenges, and we can only give an incomplete analysis. Further complications come from the fact that the phase has a unbroken continuous non-Abelian gauge symmetry. The Landau-Ginzburg fibre is therefore not an orbifold, which would be fairly straightfoward to analyse [37–39]. Instead, we have to quotient by a continuous group. Therefore we are faced with an interacting gauge theory, which is why we refer to this phase as strongly coupled. But there is more: In contrast to the Rødland, Hosono-Takagi and KKSS-models, this example in addition has a non-compact Coulomb branch in the strongly coupled phase. GLSMs that exhibit this phenomenon are called non-regular [9,13] and are poorly understood (see [40] for a recent analysis of a non-regular GLSM). We have not succeeded in understanding the physics of this phase well enough to extract a geometry out of it. Especially given the appearance of Coulomb branch, it is remarkable that the model still has a MUM-point associated to this phase.

After our inconclusive GLSM analysis we return to the problems identified in [29, 33], where naive attempts failed to produce an integral symplectic basis that could be related to topological quantities of a mirror threefold. The resolution to this conundrum lies in a modification to the prepotential identified in [4], which must be taken into account when the MUM point is mirror not to a smooth threefold, but to the noncommutative resolution of a singular threefold.

With this in mind, we proceed to analyse the MUM point $\varphi = \infty$ using the tools of [4]. Instead of directly realising a geometry in the GLSM, we discuss in Appendix §A how to obtain topological string free energies up to genus 4 using the approaches of [41–44] to solving the holomorphic anomaly equations [45,46]. The topological string free energy can be analytically continued from the geometric phase that we understand at $\zeta \gg 0$ to the phase that we aim to better understand at $\zeta \ll 0$. This provides enough information for us to bootstrap topological data for the smooth deformation of whichever singular threefold is hiding at $\varphi = \infty$ (or equivalently, $\zeta \ll 0$). In this way we recognise a familiar example, namely we anticipate that $\varphi = \infty$ corresponds to a noncommutative resolution of an intersection

$$\mathbb{WP}^5_{111223}[4,6], \tag{3}$$

with 63 nodal singularities, where a non-Kähler resolution has $\mathbb{Z}_3$ torsion. Now the constant term formula in [4] solves our monodromy problem, and allows us to proceed with the genus expansion up to $g = 11$ as tabulated in Appendix §B. Beyond genus 11, there is insufficient information for us to fix the holomorphic ambiguity of [41]. We take this successful expansion

up to genus 11, including the highly nontrivial reproduction of the constant term in the topological string free energy at each lower genus as computed in [4], to justify the assumptions we make in using the results of [4].

## 1.1 Dramatis personae

Here we outline this paper's geometries. In §2.1 we construct a GLSM whose large volume phase realises $Y$. We study the other phase, for which we denote the anticipated geometry by $X$. $Y$ is the $\mathbb{Z}_3$ quotient of $\widetilde{Y}$, for which we get a mirror $\widetilde{\Lambda}$ from toric mirror symmetry in §3. We identify $\Lambda \cong \widetilde{\Lambda}/\mathbb{Z}_3$ as the mirror of $Y$. The PF operator of $\Lambda$ is AESZ17, with two MUM points. By expanding topological string free energies about both MUM points we read off data for $X$, which we combine with a monodromy analysis for AESZ17 to formulate a conjecture regarding $X$. We find that integral invariants for $X$, and an integral basis of periods for AESZ17, can be obtained by following the torsion-refined prescription of [2, 4]. This suggests that $X$ is a singular member of the family $X_{\mathrm{def}} \cong \mathbb{WP}^5_{111223}[4,6]$ with 63 nodal singularities, for which $c_1 = 0$ resolutions $\widehat{X}$ are non-Kähler. $X_{\mathrm{def}}$ has its own mirror, with PF operator AESZ12. We combine data from BPS expansions for AESZ17 and AESZ12 to obtain torsion-refined invariants for $X$, tabulated in §B, following [2, 4].

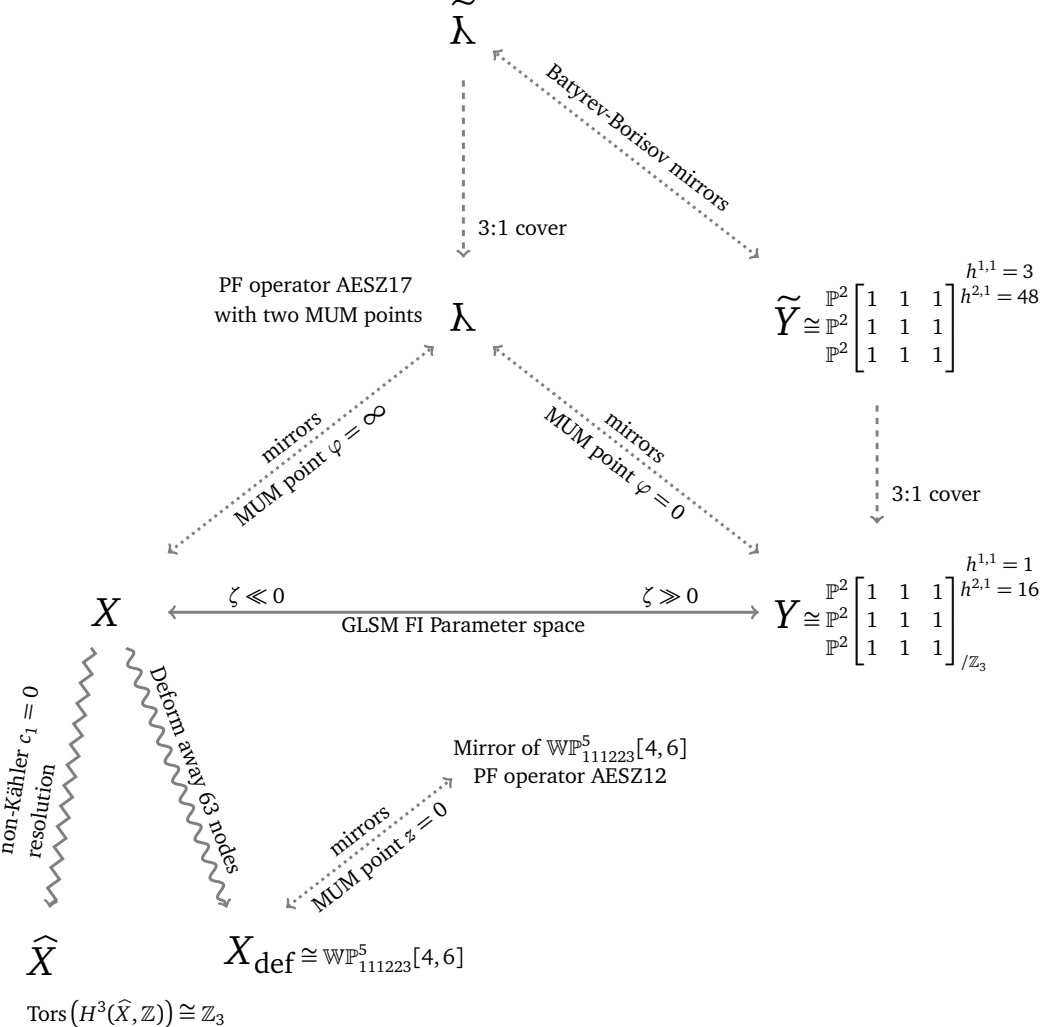

Figure 1: Relations between this paper's geometries.

## 2   GLSM analysis

In this section we analyse the Calabi-Yau and its moduli space from a GLSM perspective. We propose that the associated GLSM is a one-parameter non-Abelian theory with gauge group $G = (U(1) \times U(1) \times U(1)) \rtimes \mathbb{Z}_3$. We recover the geometry $Y$ in (1) as the "large volume" phase. We confirm the existence of three singular points at the phase boundary by a Coulomb branch analysis and compute the topological data of $Y$ using the GLSM hemisphere partition function. We show that the small volume phase is a hybrid model with an unbroken continuous gauge group, together with an extra Coulomb branch.

### 2.1   GLSM data and field content

We consider a GLSM with gauge group

$$G = (U(1) \times U(1) \times U(1)) \rtimes_\alpha \mathbb{Z}_3 . \tag{4}$$

The homomorphism $\alpha : \mathbb{Z}_3 \mapsto \mathrm{Out}\left(U(1)^3\right)$ specifies the outer automorphism of $U(1)^3$ used to define the semidirect product. The $\mathbb{Z}_3$ should be thought of as the group of cyclic permutations of three elements. We write the three elements of this cyclic group as $v_0 = \mathrm{Id} = (1, 2, 3)$, $v_1 = (2, 3, 1)$, $v_2 = (3, 1, 2)$. An arbitrary element of this $\mathbb{Z}_3$ will be written $v$.

Writing an arbitrary element of $U(1)^3$ as $(\lambda_1, \lambda_2, \lambda_3)$ with each $\lambda_i \in U(1)$, the image of the generator $v_1$ of $\mathbb{Z}_3$ under $\alpha$ is the automorphism $\alpha_{v_1} : (\lambda_1, \lambda_2, \lambda_3) \mapsto (\lambda_2, \lambda_3, \lambda_1)$. Note that $\mathrm{Weyl}(G) \cong \mathbb{Z}_3$.

An arbitrary element $g \in G$ is the pair

$$g = ((\lambda_1, \lambda_2, \lambda_3), v) , \qquad \lambda_i \in U(1) \text{ and } v \in \mathbb{Z}_3 . \tag{5}$$

The standard semidirect product multiplication rule that defines $G$ is

$$((\lambda_1, \lambda_2, \lambda_3), v) \circ ((\hat{\lambda}_1, \hat{\lambda}_2, \hat{\lambda}_3), \hat{v}) = ((\lambda_1, \lambda_2, \lambda_3) \cdot \alpha_v(\hat{\lambda}_1, \hat{\lambda}_2, \hat{\lambda}_3), v \cdot \hat{v}) . \tag{6}$$

Note that $G$ is a matrix Lie group, we can identify any element $g$ as in (5) with the matrix

$$\mathrm{M}(g) = \begin{pmatrix} \lambda_1 & 0 & 0 \\ 0 & \lambda_2 & 0 \\ 0 & 0 & \lambda_3 \end{pmatrix} \cdot \mathrm{Perm}(v) \in GL(3, \mathbb{C}) , \tag{7}$$

where $\mathrm{Perm}(v)$ is the permutation matrix effecting the permutation $v$ on a three-vector.

The model has four triples of chiral superfields $P^i, X_i, Y_i, Z_i$ ($i = 1, 2, 3$) with the respective scalar components $p^i, x_i, y_i, z_i$, and three vector multiplets $\Sigma_i$ with scalar components $\sigma_i$. The chirals are charged as follows under the three $U(1)$ gauge symmetries and the vector $U(1)$ R-symmetry:

|  | $P^1$ | $P^2$ | $P^3$ | $X_1$ | $X_2$ | $X_3$ | $Y_1$ | $Y_2$ | $Y_3$ | $Z_1$ | $Z_2$ | $Z_3$ | FI |
|---|---|---|---|---|---|---|---|---|---|---|---|---|---|
| $U(1)_1$ | $-1$ | $-1$ | $-1$ | $1$ | $1$ | $1$ | $0$ | $0$ | $0$ | $0$ | $0$ | $0$ | $\zeta$ |
| $U(1)_2$ | $-1$ | $-1$ | $-1$ | $0$ | $0$ | $0$ | $1$ | $1$ | $1$ | $0$ | $0$ | $0$ | $\zeta$ |
| $U(1)_3$ | $-1$ | $-1$ | $-1$ | $0$ | $0$ | $0$ | $0$ | $0$ | $0$ | $1$ | $1$ | $1$ | $\zeta$ |
| $U(1)_V$ | $2-6q$ | $2-6q$ | $2-6q$ | $2q$ | $2q$ | $2q$ | $2q$ | $2q$ | $2q$ | $2q$ | $2q$ | $2q$ | $-$ |

$$\tag{8}$$

with $0 \leq q \leq \frac{1}{3}$. In addition, the $\mathbb{Z}_3$ permutes the fields as

$$v_1 : \quad X_i \mapsto Y_i \mapsto Z_i \mapsto X_i , \qquad P^i \mapsto P^i , \qquad \Sigma_1 \mapsto \Sigma_2 \mapsto \Sigma_3 \mapsto \Sigma_1 . \tag{9}$$

We provide the explicit matrices that give the representations each of the superfields transform in. Under gauge transformation by $g$ each of the three triples $(X_i, Y_i, Z_i)^T$, $i \in \{1, 2, 3\}$, transforms as

$$\rho_{XYZ}(g) : \begin{pmatrix} X_i \\ Y_i \\ Z_i \end{pmatrix} \mapsto \mathrm{M}(g) \cdot \begin{pmatrix} X_i \\ Y_i \\ Z_i \end{pmatrix}, \tag{10}$$

with $M(g)$ given in (7). Each $P$-field transforms as

$$\rho_{\mathrm{Det}^{-1}}(g) : P^i \mapsto \frac{1}{\lambda_1 \lambda_2 \lambda_3} P^i, \tag{11}$$

where the determinant of the matrix $M(g)$ appears. The $\Sigma$ fields transform in the adjoint representation:

$$\rho_{\mathrm{Adjoint}}(g) : \begin{pmatrix} \Sigma_1 \\ \Sigma_2 \\ \Sigma_3 \end{pmatrix} \mapsto \mathrm{Perm}(v) \cdot \begin{pmatrix} \Sigma_1 \\ \Sigma_2 \\ \Sigma_3 \end{pmatrix}, \tag{12}$$

where we understand $v$ as the $\mathbb{Z}_3$ component of $g$, as displayed in (5).

The $\mathbb{Z}_3$ action on the chirals has specifically been chosen so that the quotient geometry $Y$ displayed in (1) is the vacuum manifold of the geometric phase $\zeta \gg 0$. The above action on the $\Sigma$ fields is necessary because these must transform in the adjoint of $G$ (as they are vector superfields), and the image of the group $G$ under the adjoint representation is $\mathbb{Z}_3$. Consequently, invariance of the action forces the three FI-parameters to be equal:[1] $\zeta_1 = \zeta_2 = \zeta_3 = \zeta$. The $U(1)$-actions and the $\mathbb{Z}_3$-action do not commute, so the GLSM is indeed non-Abelian.

We add a gauge-invariant superpotential[2] with R-charge 2:

$$W = A_i^{jkl} P^i X_j Y_k Z_l = P^i G_i(X, Y, Z). \tag{13}$$

Invariance under $\mathbb{Z}_3$ implies $A_i^{jkl} = A_i^{ljk} = A_i^{klj}$ so that the $G_i$ themselves are $\mathbb{Z}_3$-invariant. One could ponder on having the $\mathbb{Z}_3$ symmetry permute the $P$ fields as well, but an analysis of the choices of gauge invariant superpotential (following the discussion in [25, §3.2.1]) reveals that the specific CICY quotient $Y$ in (1) can only be obtained as a vacuum geometry by choosing $\mathbb{Z}_3$ to leave the $P$ fields invariant.

Since we want to realise a smooth geometry $Y$ in the $\zeta \gg 0$ phase, we must make a further assumption on the $G_i$, which means that the GLSM superpotential $W$ has to satisfy suitable genericity constraints. Namely, the intersection of the three hypersurfaces $G_i = 0$ in $\left(\mathbb{P}^2 \times \mathbb{P}^2 \times \mathbb{P}^2\right)$ should be smooth, and so we require that the intersection be transverse [20]. This requires us to take coefficients $A_i^{jkl}$ so that any solutions $(x, y, z)$ of

$$G_1 = G_2 = G_3 = \mathrm{d}G_1 \wedge \mathrm{d}G_2 \wedge \mathrm{d}G_3 = 0, \tag{14}$$

have at least one of the triples $x_i$, $y_i$, or $z_i$ equalling 0 for all $i$. We know that there is such a choice of superpotential, because we know of the smooth (1). This assumption is the minimal one that guarantees a transverse intersection in $(\mathbb{P}^2)^3$. We spell this out now because this assumption serves as defining data for our model, which we chose to recover a certain geometry (1) in the $\zeta \gg 0$ phase, with implications that we will analyse in the other phase at $\zeta \ll 0$.

---

[1] Depending on whether the quotient action in the GLSM is implemented with a semidirect or a direct product, the GLSM FI parameters will or will not have to be equated in the GLSMs associated to quotient Calabi-Yaus such as those displayed in [25].

[2] We will often use the Einstein summation convention, so sums will not be displayed explicitly.

Following [1], we write the scalar potential $U$ that follows from (13):

$$U(x,y,z,p,\sigma) = |G_1|^2 + |G_2|^2 + |G_3|^2 + \sum_{s \in \{x^1,\ldots,z^3\}} \left| p^i \frac{\partial G_i}{\partial s} \right|^2$$
$$+ \frac{1}{2}\left(|x|^2 - |p|^2 - \zeta\right)^2 + \frac{1}{2}\left(|y|^2 - |p|^2 - \zeta\right)^2 + \frac{1}{2}\left(|z|^2 - |p|^2 - \zeta\right)^2 \tag{15}$$
$$+ 2|p|^2|\sigma_1 + \sigma_2 + \sigma_3|^2 + 2|x|^2|\sigma_1|^2 + 2|y|^2|\sigma_2|^2 + 2|z|^2|\sigma_3|^2,$$

where $|x|^2 = \sum_i |x_i|^2$, et cetera. On the Higgs branch, where $\sigma_1 = \sigma_2 = \sigma_3 = 0$, the ground state is determined by the D-term and F-term equations. The three D-term equations are

$$|x|^2 - |p|^2 - \zeta \ = \ |y|^2 - |p|^2 - \zeta \ = \ |z|^2 - |p|^2 - \zeta \ = \ 0. \tag{16}$$

The twelve F-term equations are

$$G_1 \ = \ G_2 \ = \ G_3 \ = \ 0, \qquad p^i \frac{\partial G_i}{\partial s} \ = \ 0, \qquad s \in \{x_1, x_2, x_3, y_1, y_2, y_3, z_1, z_2, z_3\}. \tag{17}$$

Before we discuss the phases, let us compare this model to other well-studied non-Abelian GLSMs. The field content and symmetries of the present model bear various similarities but also notable differences to models studied by Hori and Tong [9] inspired by a pair of non-birationally equivalent Calabi-Yaus found by Rødland [10], and also models due to Hosono and Takagi [12] whose GLSM realisation has been found in [13]. Both the Rødland and the Hosono-Takagi models have chiral fields with similar charge matrices as the present model (8), i.e. a number of "$P$-fields" that have charge $-1$ under all $k$ $U(1)$-subgroups of the rank $k$ gauge group $G$ and a set of "$X$-type fields" that can be divided up into $k$ components each of which has charge 1 under one of the $k$ $U(1)$s. There is also an action of a discrete group. For Rødland-type models this is the $S_k$ Weyl group of $U(k)$, for the Hosono-Takagi-type models the discrete symmetry comes from the fact that $G = (U(1) \times O(2))/\{\pm 1, \pm \mathbf{1}\} \simeq (U(1) \times U(1)) \rtimes \mathbb{Z}_2$. The latter formulation can be generalised to our model if we increase the rank of $G$ from 2 to 3 and replace $\mathbb{Z}_2$ by $\mathbb{Z}_3$. However, for the present model, the $X, Y, Z$-fields cannot be rearranged into fundamental representations of some $U(k)$ or $O(k)$ gauge group.

A further difference between our GLSM and other one-parameter models with two geometric phases such as the Rødland model and the Hosono-Takagi model, or other examples such as those discussed in [3, 4], is the superpotential. To our knowledge, all the models with two geometric phases that have been studied so far have a superpotential $W$ that is quadratic in the $X$-type fields. In the small volume phase, the $P$-fields obtain a VEV and generate masses for the $X$-fields by way of a mass matrix $M^{ij}(p)$. The IR physics crucially depends on the properties of the mass matrix. The properties of the low-energy theory change at loci where the rank of $M^{ij}(p)$ drops. The low-energy effective theory can be formulated in terms of a non-linear sigma model, potentially with a $B$-field, whose target space is the determinantal variety defined by a rank condition on the mass matrix. In our model, and in the $U(k)$ GLSMs defined by Hori-Tong that have $k > 2$, the potential is a degree $k$ polynomial in the $X$-fields. Rather than being a massive theory, the small volume phase then becomes an interacting theory that can be understood as a Landau-Ginzburg theory on a stack fibred over a geometric base. Indeed, as we will show below, the small volume phase of such models is an interacting gauge theory and any geometric description in the deep IR has to emerge from a different mechanism.

## 2.2 Phases

### 2.2.1 $\zeta \gg 0$-phase

Understanding the classical vacuum in this phase proceeds as usual. The D-term equations

$$|x|^2 - |p|^2 \;=\; |y|^2 - |p|^2 \;=\; |z|^2 - |p|^2 \;=\; \zeta \gg 0, \tag{18}$$

have no solution on the following deleted set:

$$F_{\zeta > 0} = \{x_1 = x_2 = x_3 = 0\} \cup \{y_1 = y_2 = y_3 = 0\} \cup \{z_1 = z_2 = z_3 = 0\}. \tag{19}$$

This implies nonzero values for $|x|^2, |y|^2, |z|^2$, and therefore $\sigma_1 = \sigma_2 = \sigma_3 = 0$. The F-term equations $G_1 = G_2 = G_3 = 0$ then constrain the $x, y, z$ fields to furnish a complete intersection.

If any of the $p^i$ are nonzero, without losing generality suppose $p^1 \neq 0$, then the remaining F-terms $p^i \partial_s G_i = 0$ imply that $dG_1 = -\frac{p^2}{p^1} dG_2 - \frac{p^3}{p^1} dG_3$, and hence $dG_1 \wedge dG_2 \wedge dG_3 = 0$. But we chose $A_i^{jkl}$ such that this would not happen for all of $|x|, |y|, |z|$ nonzero, and as a result the vacuum configuration has $p^1 = p^2 = p^3 = 0$. Consequently from (18) we obtain $|x|^2 = |y|^2 = |z|^2 = \zeta$.

The triples $x_i$, $y_i$, and $z_i$ each get constrained to take values in $\mathbb{P}^2$ (after modding out each $\mathbb{C}^3 \backslash (0,0,0)$ by the appropriate $\mathbb{C}^*$). The three $\mathbb{P}^2$s have the same radius, and moreover the $\mathbb{Z}_3$ permutes them. Therefore, the ambient geometry is the free quotient $(\mathbb{P}^2 \times \mathbb{P}^2 \times \mathbb{P}^2)/\mathbb{Z}_3$. We recover the expected geometry[3] displayed in (1):

$$Y = \left\{ (x, y, z) \in (\mathbb{P}^2 \times \mathbb{P}^2 \times \mathbb{P}^2)/\mathbb{Z}_3 \,|\, G_1(x, y, z) = G_2(x, y, z) = G_3(x, y, z) = 0 \right\}. \tag{20}$$

For later reference we compute the topological characteristics of $Y$. It is convenient to introduce $\widetilde{Y}$, the simply connected cover of $Y$. This is the complete intersection Calabi-Yau threefold with CICY number[4] 7669:

$$\widetilde{Y} \cong \begin{array}{c} \mathbb{P}^2 \\ \mathbb{P}^2 \\ \mathbb{P}^2 \end{array} \left[ \begin{array}{ccc} 1 & 1 & 1 \\ 1 & 1 & 1 \\ 1 & 1 & 1 \end{array} \right]^{h^{1,1}=3, h^{2,1}=48}. \tag{21}$$

Let $e_1$, $e_2$, $e_3$ denote the generating set for $H^2(\widetilde{Y}, \mathbb{Z})$ given by the pullbacks to $\widetilde{Y}$ of each of the Kähler classes of the three $\mathbb{P}^2$ factors of the ambient space. The adjunction formula [36] gives

$$\widetilde{\kappa}_{ijk} \equiv \int_{\widetilde{Y}} e_i \wedge e_j \wedge e_k = \begin{cases} 0, & i = j = k, \\ 6, & i, j, k \text{ distinct}, \\ 3, & \text{otherwise}, \end{cases} \tag{22}$$

$$\widetilde{(c_2)}_i \equiv \int_{\widetilde{Y}} c_2(\widetilde{Y}) \wedge e_i = 36, \qquad \chi(\widetilde{Y}) = -90. $$

To compute topological data for the quotient $Y$, we will make use of the fact that the $\mathbb{Z}_3$-invariant part of $H^2(\widetilde{Y}, \mathbb{Z})$ is spanned by $e_1 + e_2 + e_3$. Under the quotient map $q : \widetilde{Y} \mapsto Y$, we have that the pullback of the generator $e$ of $H^2(Y, \mathbb{Z})_{\text{Free}}$ is

$$q^*(e) = e_1 + e_2 + e_3. \tag{23}$$

---

[3]This can be viewed as either a complete intersection in the quotient $(\mathbb{P}^2)^3/\mathbb{Z}_3$ or as the $\mathbb{Z}_3$ quotient of a complete intersection in $(\mathbb{P}^2)^3$.

[4]Originally compiled in [20], the full CICY list is displayed online at [47] where topological data is listed, together with this numbering that we reference.

We then compute the triple intersection number for $Y$ via

$$
\begin{aligned}
\kappa_{111} &= \int_Y e \wedge e \wedge e = \frac{1}{|\mathbb{Z}_3|} \int_{\widetilde{Y}} q^*(e) \wedge q^*(e) \wedge q^*(e) \\
&= \frac{1}{3} \int_{\widetilde{Y}} (e_1 + e_2 + e_3) \wedge (e_1 + e_2 + e_3) \wedge (e_1 + e_2 + e_3) = \frac{1}{3} \sum_{i,j,k=1}^{3} \widetilde{\kappa}_{ijk} = 30.
\end{aligned}
\tag{24}
$$

Similarly, we find the second Chern number

$$
c_2 = \int_Y c_2(Y) \wedge e = \frac{1}{3} \int_{\widetilde{Y}} c_2(\widetilde{Y}) \wedge (e_1 + e_2 + e_3) = \frac{1}{3} \sum_{i=1}^{3} \widetilde{(c_2)}_i = 36.
\tag{25}
$$

The Euler characteristic $\int_Y c_3(Y)$ and fundamental group $\pi_1(Y)$ are

$$
\chi(Y) = \frac{\chi(\widetilde{Y})}{3} = -30, \qquad \pi_1(Y) \cong \mathbb{Z}_3.
\tag{26}
$$

The latter relation holds because $\widetilde{Y}$ is simply connected, and the fundamental group of the free quotient of a simply connected space is the quotient group.

### 2.2.2 $\zeta \ll 0$-phase

The vacuum equations necessitate that $\sigma_1 + \sigma_2 + \sigma_3 = 0$, which follows once we establish that $|p| \neq 0$ in this phase's classical vacua. We first look at the branch where $\sigma_1 = \sigma_2 = \sigma_3 = 0$, which is sufficient in models where the gauge and matter sectors decouple. We will then go on to more closely analyse whether this decoupling occurs.

The D-term equations

$$
|x|^2 - |p|^2 = |y|^2 - |p|^2 = |z|^2 - |p|^2 = \zeta \ll 0,
\tag{27}
$$

imply that the deleted set is

$$
F_{\zeta < 0} = \{p^1 = p^2 = p^3 = 0\}.
\tag{28}
$$

Suppose for contradiction that a nonzero value of $x, y, z$ solves the twelve F term equations (17). Then the D-terms (27) would imply that $|x|^2 = |y|^2 = |z|^2$, so all three would be nonzero. Then, since at least one of the $p^i$ is nonzero by (28), we would get a nonzero solution $(x, y, z)$ to (14). This occurs because if, without losing generality, $p^1 \neq 0$ then the F-terms guarantee $dG_1 = -\frac{p^2}{p^1} dG_2 - \frac{p^3}{p^1} dG_3$. By assumption solutions of (14) with none of $|x|, |y|, |z|$ equalling 0 could not occur with our choice of $A_i^{jkl}$. Therefore the classical vacuum in the $\zeta \ll 0$ phase has $x_i = y_i = z_i = 0$ for all $i$.

Note that to reach this conclusion $|x| = |y| = |z| = 0$ we have had to use the D-term constraints that $|x|^2 = |y|^2 = |z|^2$, so that there cannot for instance be vacua[5] with $|x| = 0$ but $|y|, |z| \neq 0$.

Now the D-term equations (27) can be seen to imply $|p|^2 = -\zeta$. It remains to mod out by the gauge symmetry. The vacuum is then given[6] by

$$
\{ p_i \in \mathbb{C}^3 \mid |p|^2 = -\zeta \} / ( p_i \sim (\lambda_1 \lambda_2 \lambda_3)^{-1} p_i ).
\tag{29}
$$

---

[5]If we considered the related Abelian $U(1)^3$ model with no $\mathbb{Z}_3$ gauging and three nonequal FI parameters $\zeta_1, \zeta_2, \zeta_3$, then the phase analysis would be completely different due to the possibility of branches with only one of $|x|, |y|, |z|$ equalling zero. Indeed that Abelian $U(1)^3$ model with three distinct FI parameters would have four geometric phases, each giving intersections in $(\mathbb{P}^2)^3$ but with different triples from $(p, x, y, z)$ providing the ambient $(\mathbb{P}^2)^3$.

[6]Alternatively, we can write this as a GIT quotient $\{p_i \in \mathbb{C}^3 - \{0\}\}/\mathbb{C}^*$ which produces the same topological space. However, since we will soon discuss the subtleties of both discrete and continuous unbroken symmetries, we proceed with (29).

Topologically, the vacuum manifold (29) is a $\mathbb{P}^2$. The symmetry is broken to $(U(1) \times U(1)) \rtimes \mathbb{Z}_3$ where the elements of the $U(1)^3$-part of the gauge group reduce to

$$(\lambda_1, \lambda_2, \lambda_3) \rightarrow (\lambda_1, \lambda_2, (\lambda_1 \lambda_2)^{-1}), \tag{30}$$

and the $\mathbb{Z}_3$ cyclically permutes the three elements. Our unbroken symmetry group is thus continuous and non-Abelian.

We note that, in contrast to the Rødland, Hosono-Takagi, and KKSS-type models, the dimension of the vacuum manifold is less than three, so we will not get a threefold by constructing geometries that are determinantal varieties inside the vacuum manifold or branched covers thereof. To obtain the low-energy effective theory, we have to turn on fluctuations of the $X, Y, Z$-fields. This generates a potential

$$W_{\zeta < 0} = \langle P^i \rangle G_i(X, Y, Z) = A^{ijk}(\langle P \rangle) X_i Y_j Z_k, \tag{31}$$

where $\langle P^i \rangle$ signifies that the $p^i$ are constrained to the vacuum. We recover the structure of a hybrid model, i.e. a Landau-Ginzburg model fibred over a geometric base. The unbroken symmetry group acts non-trivially on the fibre fields. Since we have a continuous unbroken gauge symmetry, this is not a standard Landau-Ginzburg orbifold. Mathematically, this means that the hybrid model lives on an Artin stack rather than a Deligne-Mumford stack. In addition, it turns out that the gauge degrees of freedom do not decouple. Indeed, following an analysis[7] in [9, §4.2], the low-energy effective theory suffers from a Coulomb branch. When the symmetry is broken and the $p$-fields have a VEV, the low energy effective theory consists of two vector multiplets $\Sigma_1, \Sigma_2$ associated to the two $U(1)$s and the chiral multiplets $X, Y, Z$. As can be seen from a change of basis in (8), the gauge charges of the chiral fields are

|         | $X_i$ | $Y_i$ | $Z_i$ | FI |
|---------|-------|-------|-------|----|
| $U(1)_1$ | 1     | 0     | $-1$  | 0  |
| $U(1)_2$ | 0     | 1     | $-1$  | 0  |

$$\tag{32}$$

These fields are massive for large $\Sigma_i$ and can be integrated out, leading to an effective potential (see §2.3 below for more details)

$$\mathcal{W}_{eff} = -3\sigma_1(\log \sigma_1 - 1) - 3\sigma_2(\log \sigma_2 - 1) - 3(-\sigma_1 - \sigma_2)(\log(-\sigma_1 - \sigma_2) - 1). \tag{33}$$

The critical locus is at $\sigma_2 = e^{\pm \frac{2\pi i}{3}} \sigma_1$, so we indeed have a Coulomb branch and the theory becomes singular in the IR. This renders the phase non-regular. There is no real separation of scale between the Coulomb and the strongly coupled branch and methods like the Born-Oppenheimer approximation do not apply, making the phase hard to analyse.

At this point it is not clear to us how to describe the CFT in the IR, nor how to exhibit any relevant type of geometry. Ignoring the issues around non-regularity and the Coulomb branch for the moment,[8] we investigate one possible source for a geometry emerging from the chiral sector. The fibre fields $x, y, z$ do not have a mass term, so, in contrast to previously studied models involving quadrics, we cannot expect to obtain a geometry from the behaviour of a mass matrix. We note however that the cyclically symmetric three-tensor $A^{ijk}(\langle p \rangle)$ can be interpreted as an array of coupling constants for the interacting theory. Some couplings will vanish when the rank of this tensor drops, leading to the expectation that at some loci of the vacuum manifold there will be a free theory. We thus suspect that the low energy effective theory will change its physical properties when the hyperdeterminant[9] of $A^{ijk}(\langle p \rangle)$ vanishes.

---

[7]We thank K. Hori for explanations and correspondence.

[8]We will explain in §2.3 why this is justified for negative finite FI parameter.

[9]We thank S. Hosono for suggesting to consider the hyperdeterminant locus. We are also aware of similar considerations by T. Schimannek [48].

For various equivalent definitions of the hyperdeterminant, and a discussion of its properties as a generalisation of the usual determinant, see [49]. Let $x^{(1)} \in \mathbb{C}^{L_1}$, $x^{(2)} \in \mathbb{C}^{L_2}$, ..., $x^{(r)} \in \mathbb{C}^{L_r}$ denote vectors in complex spaces of dimension $L_1$, $L_2$, ..., $L_r$. Then consider the multilinear form

$$f(x^{(1)}, x^{(2)}, \ldots, x^{(r)}) = \sum_{\substack{1 \le i_1 \le L_1 \\ \cdots \\ 1 \le i_r \le L_r}} a_{i_i i_2 \ldots i_r} x_{i_1}^{(1)} x_{i_2}^{(2)} \ldots x_{i_r}^{(r)}. \tag{34}$$

The hyperdeterminant H($f$) of $f$, when it exists, is a polynomial in the entries $a_i$ that vanishes if and only if there is a list of nonzero vectors $(x^{(1)}, x^{(2)}, \ldots, x^{(r)})$ such that

$$f(x^{(1)}, x^{(2)}, \ldots, x^{(k-1)}, y^{(k)}, x^{(k+1)}, \ldots, x^{(r)}) = 0, \quad \text{for all } 1 \le k \le r, \\ \text{and for all } y^{(k)} \in \mathbb{C}^{L_k}. \tag{35}$$

Note that, as in Theorem 1.4 of [49], H($f$) exists if and only if $L_k - 1 \le \sum_{j \ne k}(L_j - 1)$ for all $k$.

The degree of H($f$) grows rapidly with the dimensions $L_k$ and the number of vector spaces $r$. For our case of interest, which is H($A^{ijk}(\langle p \rangle)$), we have $r = L_1 = L_2 = L_3 = 3$. This hyperdeterminant is a homogeneous polynomial of degree 36 in the entries $A^{ijk} \equiv A_m^{ijk} p^m$. It is a very large expression.

In fact, until the work [50], the hyperdeterminant of a general $3 \times 3 \times 3$ tensor was not known explicitly. The approach of [50] is to use the fact that the hyperdeterminant of a $3 \times 3 \times 3$ array, with entries $A^{ijk}$, is necessarily a polynomial in invariants of $SL(3, \mathbb{C}) \times SL(3, \mathbb{C}) \times SL(3, \mathbb{C}) \rtimes S_3$. There are three independent invariants, denoted $I_6$, $I_9$, and $I_{12}$ with the subscript giving their homogeneous degree in the $A^{ijk}$. Note that each is defined up to an overall scale, and $I_{12}$ is defined up to adding multiples of $I_6^2$. In [51] these invariants were explicitly calculated, and we proceed with their choice of scale and definition of $I_{12}$. Then the result of [50] is

$$H(A^{ijk}) = I_6^3 I_9^2 - I_6^2 I_{12}^2 + 36 I_6 I_9^2 I_{12} + 108 I_9^4 - 32 I_{12}^3. \tag{36}$$

As a polynomial in $A^{ijk}$, $I_6$ has 1152 terms. $I_9$ has 9216 terms. $I_{12}$ has 209061 terms. This prevents us from straightforwardly investigating the general expression of the hyperdeterminant with regard to our phase analysis. After imposing the $\mathbb{Z}_3$ symmetry $A^{ijk} = A^{jki} = A^{kij}$, required for gauge invariance of our superpotential (13), the number of terms in each invariant decreases. Now $I_6$ has 187 terms, $I_9$ has 680, and $I_{12}$ has 4933.

The hyperdeterminant expression remains too large to work with directly in full generality. We proceed to experiment, repeatedly giving random values to the entries $A^{ijk}$ consistent with the $\mathbb{Z}_3$ symmetry. We observe that every time we do this, the explicit hyperdeterminant factorises. We find in this way that

$$H(A_m^{ijk} p^m|_{A_m^{ijk} = A_m^{jki} = A_m^{kij}}) = Q_8(p_1, p_2, p_3)^3 Q_{12}(p_1, p_2, p_3). \tag{37}$$

That is, we observe that the hyperdeterminant of the coupling tensor for the cubic interactions in the hybrid phase factorises into the cube of a single degree 8 homogeneous polynomial and a single degree 12 homogeneous polynomial. In each such random case, we observe that the hypersurface $Q_8 = 0$ in $\mathbb{P}^2$ has 4 nodal singularities (that is, $Q_8$ and $dQ_8$ vanish but the Hessian matrix is nonsingular). The hypersurface $Q_{12} = 0$ in $\mathbb{P}^2$ has 45 singular points, of which 21 are nodes.

Due to the non-regularity of the theory, is is unlikely that this is the whole story.

## 2.3 Coulomb branch

The Coulomb vacua of the GLSM are determined by the critical values of the effective potential [1, 13, 52]

$$\mathcal{W}_{eff}(\sigma) = -\langle t, \sigma \rangle - \sum_{i=1}^{\dim V} \langle Q_i, \sigma \rangle (\log \langle Q_i, \sigma \rangle - 1) + i\pi \sum_{\alpha > 0} \langle \alpha, \sigma \rangle, \tag{38}$$

where $\langle \cdot, \cdot \rangle : \mathfrak{t}^*_{\mathbb{C}} \times \mathfrak{t}_{\mathbb{C}} \to \mathbb{C}$ is the pairing on the complexified Lie algebra $\mathfrak{t}_{\mathbb{C}}$ of a maximal torus of the gauge group, $V$ is the complex vector space in which the chiral scalars take values, $Q_i$ are their gauge charges, and $\alpha > 0$ are the positive roots. The parameters $t = \zeta - i\theta$ are the FI-theta parameters.

Using the parametrisation (8), $\mathcal{W}_{eff}$ for our model is

$$\mathcal{W}_{eff}(\sigma) = -t(\sigma_1 + \sigma_2 + \sigma_3) - 3(-\sigma_1 - \sigma_2 - \sigma_3)[\log(-\sigma_1 - \sigma_2 - \sigma_3) - 1] \\ - 3\sigma_1[\log \sigma_1 - 1] - 3\sigma_2[\log \sigma_2 - 1] - 3\sigma_3[\log \sigma_3 - 1]. \tag{39}$$

The critical locus is at

$$e^{-t} = -\frac{\sigma_i^3}{(\sigma_1 + \sigma_2 + \sigma_3)^3}, \qquad i = 1, 2, 3. \tag{40}$$

Defining $z = \frac{\sigma_2}{\sigma_1}, w = \frac{\sigma_3}{\sigma_2}$ and dividing these equations yields the constraints

$$z^3 = w^3 = 1. \tag{41}$$

Modulo the relations (41), the equations for the critical locus reduce to

$$e^{-t} = -\frac{1}{(1 + z + w)^3}. \tag{42}$$

Solving (41) for $z$ and $w$ explicitly gives nine solutions in terms of cubic roots of unity that we can insert back into (42). Disregarding the locus $\sigma_1 + \sigma_2 + \sigma_3 = 0$ for now, seven of these solutions determine three Coulomb branch loci near the phase boundary:

$$e^{-t} = e^{-(\zeta - i\theta)} = -\frac{1}{27}, \pm \frac{i}{3\sqrt{3}}. \tag{43}$$

Thus, there is a Coulomb branch at theta angles $-\frac{\pi}{2}, \pi, \frac{\pi}{2}$ mod $2\pi$. The result matches with the Riemann symbol of the AESZ17 operator (see Table 1) up to a sign. The sign discrepancy is due to a shift in theta angle between the GLSM and the non-linear sigma model [53] which requires us to make the identification $t = -\log \varphi + 3\pi i$, where $\varphi$ can be interpreted as the complex structure parameter of the mirror Calabi-Yau near the large complex structure point.

To analyse the Coulomb branch further, we take a closer look at how the three points arise as solutions of the vacuum equations. Let $\alpha = e^{\frac{2\pi i}{3}}$. Then the choices for $z$ and $w$ that solve (42) contribute as follows:

| $e^{-t} = -\frac{1}{27}$ | $e^{-t} = +\frac{i}{3\sqrt{3}}$ | $e^{-t} = -\frac{i}{3\sqrt{3}}$ |
|---|---|---|
| $z = 1, w = 1$ | $z = \alpha^2, w = \alpha^2$ | $z = \alpha, w = \alpha$ |
| | $z = 1, w = \alpha$ | $z = 1, w = \alpha^2$ |
| | $z = \alpha, w = 1$ | $z = \alpha^2, w = 1$ |

(44)

Notice that the first solution is fixed by the $\mathbb{Z}_3$-symmetry. By a conjecture in [13], this signifies that there are three disjoint Coulomb branches at $e^{-t} = -\frac{1}{27}$ rather than one, indicating that

there are three massless hypermultiplets of the same charge at this locus. This is consistent with the results that we obtain in §3.1 on the singular degenerations of the mirror manifold. We further note that the two loci $\pm\frac{i}{3\sqrt{3}}$ both have the same distance from the two phases and are closer to the strongly coupled phase.

Due to the relation $1 + \alpha + \alpha^2 = 0$, we find another solution to the Coulomb branch equations at $1 + z + w = 0$, or equivalently, $\sigma_1 + \sigma_2 + \sigma_3 = 0$:

$$e^{-t} \to \infty: \qquad \begin{aligned} z &= \alpha, & w &= \alpha^2, \\ z &= \alpha^2, & w &= \alpha. \end{aligned} \qquad (45)$$

This happens in non-regular theories and there is a connection with the Coulomb branch we found in the strongly-coupled phase. The locus has to be excluded from the analysis of the GLSM Coulomb branch because the $p$-fields would be massless in this case and cannot be integrated out, making (39) invalid. However, from the perspecive of the strongly coupled phase, the locus $\sigma_1 + \sigma_2 + \sigma_3 = 0$ determines the Coulomb branch in the $\zeta < 0$-phase.

In the context of $U(k)$ GLSMs with a strongly coupled $SU(k)$-phase at $\zeta < 0$ a careful analysis in [9] showed that the Coulomb branch in the strongly coupled phase gets lifted at finite $\zeta < 0$ but not at $\zeta \to -\infty$. In contrast to the Coulomb branches at the phase boundary, we cannot associate a specific (mod $2\pi$)-theta angle value to this Coulomb branch. Rather, there is a Coulomb branch for any value of the theta angle.[10] It is expected that the CFT in the limit $\zeta \to -\infty$ is singular. This is not a problem for string compactifications. Typical examples with singular CFTs are conifold points or pseudo-hybrid points [54] which have interesting physics and mathematics. In physics, this means that there are extra massless states, with implications for supergravity/black holes [41, 55, 56] and the connectedness of the moduli space of Calabi-Yau string vacua [57–59]. Mirror symmetry seems to be fine with these structures as well and leads to sensible results.

Let us give a more detailed analysis of Coulomb branches at infinity and their effects following [9, §4.4]. The argument generalises to one-parameter GLSMs with maximal torus $(U(1)_1 \times \ldots \times U(1)_k)$ and an additional $\mathbb{Z}_k$-action that cyclically permutes the $U(1)$s and this should apply to our model. We consider $N$ types of matter fields $X$ that can be organised into $k$-plets and $N$ $P$-fields with charges

| $\phi$ | $P^1, \ldots, P^N$ | $X_1^1, \ldots X_N^1$ | $X_1^2, \ldots X_N^2$ | $\ldots$ | $X_1^k, \ldots X_N^k$ | FI |
|---|---|---|---|---|---|---|
| $U(1)_1$ | $-1$ | $1$ | $0$ | $\ldots$ | $0$ | $\zeta$ |
| $U(1)_2$ | $-1$ | $0$ | $1$ | $\ldots$ | $0$ | $\zeta$ |
| $\ldots$ | $\ldots$ | $\ldots$ | $\ldots$ | $\ldots$ | $\ldots$ | $\ldots$ |
| $U(1)_k$ | $-1$ | $0$ | $0$ | $\ldots$ | $1$ | $\zeta$ |

$$(46)$$

The $X$-fields may be components of a fundamental $k$-plet as in Rødland-type models or they can be viewed as coordinates of a free quotient by $\mathbb{Z}_k$ of the toric variety determined by $U(1)^k$.

The vacuum equations for the GLSM Coulomb branch are

$$e^{-t} = \pm \frac{\sigma_1^N}{(\sigma_1 + \ldots + \sigma_k)^N} = \ldots = \pm \frac{\sigma_k^N}{(\sigma_1 + \ldots + \sigma_k)^N}, \qquad (47)$$

with the $\pm$ accounting for theta angle shifts due to the presence of $W$-bosons or regularity conditions such as those discussed in [13, 14]. There is a Coulomb branch whenever $(\frac{\sigma_i}{\sigma_j})^N = \pm 1$ for all $i \neq j$. Some of these solutions will lead to the same $e^{-t}$, some have to be discarded because they correspond to some matter fields becoming massless. In this case, it would not

---

[10]When compactifying the moduli space to a punctured sphere, the puncture associated to this Coulomb branch should lie on the south pole.

be allowed to integrate out those fields, making the expression for the effective potential (38), which is derived under the assumption that all matter fields are massive, invalid.[11] For the GLSM (46), this excludes the Coulomb branch at $\sigma_1 + \ldots + \sigma_k = 0$, which is realised by $\sigma_j = \alpha^j$ with $\alpha = e^{\frac{2\pi i}{k}}$.

However, at this locus there remains a Coulomb branch at $\zeta \to -\infty$ coming from Coulomb branch in the strongly coupled phase which exists by a generalisation of the argument of [9, §4.2], as discussed in the previous section. To show that it is lifted everywhere except at $\zeta \to -\infty$, we follow the arguments in [9, §4.4]. In the strongly coupled phase at $\zeta < 0$ the determinantal $U(1)$ subgroup $U(1)_0$ is broken and the $p$-fields get a VEV. At large but finite $-\zeta$ the gauge sector decouples because the broken $U(1)$ dynamically generates t-dependent twisted masses for the $\sigma$-fields everywhere except at $\zeta \to -\infty$ where the masses disappear. We have to understand the effective potential on the Coulomb branch for large but finite $-\zeta$. We give the field strength large, distinct eigenvalues $\widetilde{\sigma}_a$ such that

$$\sum_{a=1}^{k} \widetilde{\sigma}_a = 0 \,. \tag{48}$$

The eigenvalues set an energy scale $m$. The chiral matter fields $\phi$ all have masses of order $m$ due to the effective scalar potential having terms

$$U_{eff} = \ldots + |\widetilde{\sigma}\phi|^2 + \ldots \tag{49}$$

The massive fields must be integrated out at energy scales below $m$. The effective theory consists of a theory of $k$ chiral multiplets $P^\alpha$ that are charged only under $U(1)_0$. There is an effective FI-parameter $\zeta_0(m) = -\zeta \gg 0$. When the energy is decreased, the FI-parameter $\zeta_0$ runs towards smaller values[12] and will become negative. In this region one can integrate out the $P$-fields. Doing this, one gets an effective potential for the scalar component $\sigma_0$ of the vector superfield $\Sigma_0$ associated to $U(1)_0$ with the $\widetilde{\sigma}_a$ being treated as parameters. Since on the Coulomb branch the other matter fields have been massive in the first place, the effective potential is the same as (38) but we now single out the field $\sigma_0$ by defining

$$\sigma_a = \widetilde{\sigma}_a - \frac{\sigma_0}{k} \,. \tag{50}$$

Integrating out $\sigma_0$ yields

$$\sigma_0 = f(e^{-t}, \widetilde{\sigma}) \,. \tag{51}$$

This should be interpreted as follows. The Higgsed $U(1)_0$-sector dynamically creates twisted masses $\sigma_0/k = f(e^{-t}, \widetilde{\sigma})/k$ for the chirals of the strongly coupled theory. Reinserting this into the effective potential, creates a potential for the remaining $\sigma$-fields associated to the residual gauge symmetry. This lifts the Coulomb branch as long as $\zeta \ll 0$ but finite.

Let us show how this works for our model. Following the discussion of [9], we denote by $\sigma_0$ the $\sigma$-field associated to the determinantal $U(1)$. We define

$$\sigma_1 = \widetilde{\sigma}_1 - \sigma_0, \qquad \sigma_2 = \widetilde{\sigma}_2, \qquad \sigma_3 = -\widetilde{\sigma}_1 - \widetilde{\sigma}_2 \,. \tag{52}$$

Inserting this into (39) we get

$$\begin{aligned}
\mathcal{W}_{eff} = \mathrm{t}\sigma_0 &- 3(\widetilde{\sigma}_1 - \sigma_0)(\log(\widetilde{\sigma}_1 - \sigma_0) - 1) - 3\widetilde{\sigma}_2(\log(\widetilde{\sigma}_2) - 1) \\
&- 3(-\widetilde{\sigma}_1 - \widetilde{\sigma}_2)(\log(-\widetilde{\sigma}_1 - \widetilde{\sigma}_2) - 1) - 3\sigma_0(\log(\sigma_0) - 1) \,.
\end{aligned} \tag{53}$$

---

[11]For example, in the Rødland model one has to discard solutions fixed by the Weyl group action because those would correspond to W-bosons being massless. This reasoning does not apply to our model which does not have W-bosons, so solutions fixed by the $\mathbb{Z}_3$ must be kept.

[12]Note that the effective theory is not Calabi-Yau, so the effective FI parameter undergoes RG flow.

Now we integrate out $\sigma_0$. Computing $\partial_{\sigma_0} \mathcal{W}_{eff} = 0$ gives

$$\mathrm{t} + 3\log(\tilde{\sigma}_1 - \sigma_0) - 3\log(\sigma_0) = 0. \tag{54}$$

Solving the equation above for $\sigma_0$ we obtain

$$\sigma_0 = \frac{1}{1 + e^{-\frac{\mathrm{t}}{3}}} \tilde{\sigma}_1 =: f(e^{-\mathrm{t}})\tilde{\sigma}_1. \tag{55}$$

This goes to zero as $\zeta \to -\infty$, consistent with the fact that the determinantal $U(1)$ gets Higgsed at this point. Away from this limit we get a potential for $\tilde{\sigma}_1, \tilde{\sigma}_2$, lifting the Coulomb branch. To analyse this in more detail we reinsert $\sigma_0$ back into (53).

$$\begin{aligned} \mathcal{W}_{eff}(\tilde{\sigma}_1, \tilde{\sigma}_2) = & f\tilde{\sigma}_1 [\mathrm{t} + 3\log(\tilde{\sigma}_1(1-f)) - 3\log(f\tilde{\sigma}_1)] \\ & - 3\tilde{\sigma}_1 \log(\tilde{\sigma}_1(1-f)) + 3(\tilde{\sigma}_1 + \tilde{\sigma}_2)\log(-\tilde{\sigma}_1 - \tilde{\sigma}_2) - 3\tilde{\sigma}_2 \log(\tilde{\sigma}_2). \end{aligned} \tag{56}$$

The term in the brackets vanishes by (55). There is still a Coulomb branch at the critical locus of $\mathcal{W}_{eff}(\tilde{\sigma}_1, \tilde{\sigma}_2)$. Using $\log(-x) = \log(-1) + \log(x) = i\pi + \log(x)$ we compute

$$\begin{aligned} \partial_{\tilde{\sigma}_1} \mathcal{W}_{eff}(\tilde{\sigma}_1, \tilde{\sigma}_2) = & 3i\pi - 3\log(\tilde{\sigma}_1(1-f)) + 3\log(\tilde{\sigma}_1 + \tilde{\sigma}_2), \\ \partial_{\tilde{\sigma}_2} \mathcal{W}_{eff}(\tilde{\sigma}_1, \tilde{\sigma}_2) = & 3i\pi - 3\log(\tilde{\sigma}_2) + 3\log(\tilde{\sigma}_1 + \tilde{\sigma}_2). \end{aligned} \tag{57}$$

Taking the difference of the two equations, we get a condition we can solve for $\tilde{\sigma}_2$:

$$-3\log(\tilde{\sigma}_1(1-f)) + 3\log\tilde{\sigma}_2 = 0 \quad \to \quad \tilde{\sigma}_2 = (1-f)\tilde{\sigma}_1. \tag{58}$$

Inserting this back into the equations for the critical locus we get

$$\frac{(2-f)^3}{(1-f)^3} = -1. \tag{59}$$

Note that

$$1 - f = e^{-\frac{\mathrm{t}}{3}}f, \quad 2 - f = (1 + 2e^{-\frac{\mathrm{t}}{3}})f \quad \to \quad \frac{(2-f)^3}{(1-f)^3} = e^{\mathrm{t}}(1 + 2e^{-\frac{\mathrm{t}}{3}})^3. \tag{60}$$

Solving this leads to the three Coulomb branches we have identified at the phase boundary and no further singularities. So the Coulomb branch in the strongly coupled phase is lifted for finite negative $\zeta$ but gets pushed to $\zeta \to -\infty$.

## 2.4 Tentative conclusion for the $\zeta \ll 0$-phase

Let us summarise the partial results for the strongly coupled phase at $\zeta < 0$. As long as $\zeta$ is large and negative, the low energy theory is a hybrid-model with a base $\mathbb{P}^2$ and a Landau-Ginzburg-type fibre with a cubic potential. Since we have an unbroken $(U(1) \times U(1)) \rtimes \mathbb{Z}_3$-symmetry this is an interacting gauge theory. Quantum effects related to the broken symmetry imply that at finite negative values of $\zeta$ the gauge degrees of freedom decouple. The chiral fields are massless but interact via a cubic superpotential. The couplings are governed by a rank three tensor $A^{ijk}(\langle p \rangle)$ depending on the base coordinates of the fibration. It seems natural to suspect that the physics of the theory changes when some of the couplings vanish. This information is determined by the hyperdeterminant locus of the coupling tensor. In previously studied examples related to quadrics, the geometry of the Calabi-Yau in such a phase could be deduced from the properties of the mass matrix of the hybrid theory in the given phase. We expect that the coupling tensor will take the role of the mass matrix in these interacting theories, but a full analysis goes beyond the scope of the present work.

As $\zeta \to -\infty$ an additional non-compact Coulomb branch emerges. A Coulomb branch indicates the presence of extra massless degrees of freedom. The associated CFT is expected to be singular and the gauge theory degrees of freedom to not decouple. While such phenomena are fairly well-understood for Coulomb branches at phase boundaries, having such a configuration at a point at infinity in the moduli space is a feature which has so far only been observed in non-Abelian GLSMs. It raises the immediate question whether the Coulomb branch and the hybrid model interact at some level. Since the methods to analyse strongly coupled phases do not apply to non-regular GLSMs, we have not been able to answer this question. One possible approach would be to compute the Witten index for the combined hybrid/Coulomb branch system. We hope to address this in future work. Whether or not the Coulomb branch decouples also has consequences for D-branes and categorical equivalences. We give some further comments at the end of the article. Further note that there are also exotic hybrid phases that can lead to singular low-energy CFTs [54, 60] through mechanisms that are not obviously related to Coulomb branches of strongly coupled phases. This can happen in Abelian and non-Abelian GLSMs when the Higgs vacua have a more complicated structure. Since our analysis of the low-energy effective theory in the $\zeta < 0$-phase is incomplete, we cannot exclude that further singularities of this type also occur in our model. Moreover, the phenomenon of non-reguarity has so far only been described in one-parameter models. A generalisation to multiparameter models and possible connections to phenomena, such as exoflops, that arise in the context of singular phases would be interesting to investigate further.

While we have not succeeded in giving a complete description of the low-energy physics of the theory in the $\zeta < 0$-phase and we have not been able to pinpoint a smooth geometry from the GLSM analysis, mirror symmetry implies that we should find some sort of geometry in this phase. Both phases are mirror to a Calabi-Yau threefold whose Picard-Fuchs operator has two points of maximally unipotent monodromy, one of which we can relate to the geometry $Y$ (1) and the other corresponding to our problematic phase. The properties of the phase suggest that we should not expect there to be a smooth manifold but there could be something that can be interpreted as a non-commutative resolution of a singular geometry. Identifying a candidate for such a geometry will be the focus of the next section.

As a final comment, we also would like to draw some parallels between the "mixed branch" at $\zeta \to -\infty$ and phases of GLSMs which are not Calabi-Yau. There, the typical setting is that one phase is geometric and the other phases have a Higgs branch corresponding to a geometry plus additional massive vacua. The key difference here is that in the Calabi-Yau case, the additional vacua are massless.

## 2.5 GLSM B-branes and hemisphere partition function for $Y$

The GLSM hemisphere partition [52, 61, 62] function computes the central charge of a B-type D-brane and thus provides a means to compute topological data of a Calabi-Yau. Identifying a GLSM-brane associated to the structure sheaf of $Y$, we give an independent calculation of the topological characteristics of $Y$.

We briefly recall the definition of GLSM B-branes and the hemisphere partition function. Consider a B-brane $\mathcal{B} = (M, Q, \rho, r_*)$ of a GLSM, where $M$ is a $\mathbb{Z}_2$-graded Chan-Paton module, $Q$ is a $G$-invariant matrix factorisation of $R$-charge 1, $\rho$ is the representation of $G$ on $M$ and $r_*$ is the representation of the vector R-symmetry on $M$. We refer to [52, 61, 62] and later references for further details. The hemisphere partition function for a Calabi-Yau GLSM with gauge group $G$ a compact Lie group is defined as follows:

$$Z_{D^2}(\mathcal{B}) = C \int_\gamma \mathrm{d}^{\mathrm{rk}G}\sigma \prod_{\alpha > 0} \sinh \pi \langle \alpha, \sigma \rangle \prod_{i=1}^{\dim V} \Gamma\left(i\langle Q_i, \sigma \rangle + \frac{R_i}{2}\right) e^{i\langle \mathfrak{t}, \sigma \rangle} f_{\mathcal{B}}(\sigma), \qquad (61)$$

where $C$ is an undetermined constant and $\gamma$ is an integration contour that has to be chosen such that the hemisphere partition function converges in the given phase [52,63]. The contribution from the GLSM brane $\mathcal{B}$ is encoded in the brane factor $f_{\mathcal{B}}(\sigma)$:

$$f_{\mathcal{B}} = \mathrm{Tr}_M e^{i\pi r_*} \rho(e^{2\pi\sigma}). \tag{62}$$

Specialising to our model, we get

$$Z_{D^2}(\mathcal{B}) = C \int \mathrm{d}^3\sigma \, \Gamma(-i(\sigma_1 + \sigma_2 + \sigma_3) + 1 - 3q)^3 \Gamma(i\sigma_1 + q)^3 \Gamma(i\sigma_2 + q)^3 \Gamma(i\sigma_2 + q)^3 \tag{63}$$
$$\times \, e^{it(\sigma_1 + \sigma_2 + \sigma_3)} f_{\mathcal{B}}(\sigma).$$

The poles of the Gamma functions are at imaginary values of the $\sigma_i$.

Before we choose a specific brane, we consider some general properties of the hemisphere partition function in the $\zeta \gg 0$ phase. We can choose an integration contour along real values of the $\sigma_i$. Convergence of the integral then implies that we can close the integration contour at $\mathrm{Im}\,\sigma_i \to \infty$. Thus, the contour encloses the following poles of the Gamma functions:

$$i\sigma_i + q = -k_i + \varepsilon_i, \qquad i = 1,2,3, \qquad k_i \in \mathbb{Z}_{\geq 0}. \tag{64}$$

The brane factor will not introduce further poles, but it can cancel some poles coming from the Gamma functions. By standard manipulations, the hemisphere partition function can be rewritten as

$$Z_{D^2}^{\zeta \gg 0}(\mathcal{B}) = C(i)^3 \sum_{k_i \geq 0} \oint \mathrm{d}^3\varepsilon \, \frac{\Gamma(1 + k_1 + k_2 + k_3 - \varepsilon_1 - \varepsilon_2 - \varepsilon_3)^3}{\Gamma(1 + k_1 - \varepsilon_1)^3 \Gamma(1 + k_2 - \varepsilon_2)^3 \Gamma(1 + k_3 - \varepsilon_3)^3} \tag{65}$$
$$\times \, \frac{\pi^9 (-1)^{k_1 + k_2 + k_3}}{\sin^3 \pi\varepsilon_1 \sin^3 \pi\varepsilon_2 \sin^3 \pi\varepsilon_3} e^{-t(k_1 + k_2 + k_3 - \varepsilon_1 - \varepsilon_2 - \varepsilon_3)} f_{\mathcal{B}}(\varepsilon).$$

Here we have set $q = 0$ to match with the R-charges of the IR CFT. Evaluating this integral for a choice of brane factor computes the exact central charge of the respective B-brane. It can be expanded in terms of the mirror periods $\varpi^{(LV)}(\varphi)$ after taking into account the theta angle shift $t = -\log\varphi + 3i\pi$ between the GLSM and the NLSM of the phase.

The central charge of a D0-brane is proportional to the fundamental period $\varpi_0^{(LV)}(\varphi)$ of the mirror. For this type of examples, it is fairly straightforward to read this off from the hemisphere partition function without explicitly specifying a D0-brane.[13] The integrand has third order poles in each of the $\varepsilon_i$. The pole order can be reduced by suitable brane factors. For the integral to be a power series in $\varphi$, we must have first order poles. Assuming this is the case, we can deduce the following expression for the fundamental period of the mirror:

$$\varpi_0^{(LV)} = \sum_{k_i \geq 0} \frac{\Gamma(1 + k_1 + k_2 + k_3)^3}{\Gamma(1 + k_1)^3 \Gamma(1 + k_2)^3 \Gamma(1 + k_3)^3} \varphi^{(k_1 + k_2 + k_3)} \tag{66}$$
$$= 1 + 3\varphi + 27\varphi^2 + 381\varphi^3 + 6219\varphi^4 + 111753\varphi^5 + \mathcal{O}(\varphi^6).$$

This indeed coincides with the holomorphic solution of the Picard-Fuchs equation, see (81) below.

We proceed to confirm the topological data of the Calabi-Yau in the $\zeta \gg 0$-phase by computing the central charge of the D6-brane associated to the structure sheaf. For this purpose, we consider the following matrix factorisation

$$Q = \sum_{i=1}^{3} p_i \eta_i + \frac{\partial W}{\partial p_i} \bar{\eta}_i, \tag{67}$$

---

[13]This will not be as easy for GLSMs whose gauge groups have positive roots as the integrand of the hemisphere partition function will have a more complicated structure.

where $\eta_i, \overline{\eta}_i$ are Clifford matrices satisfying $\{\eta_i, \overline{\eta}_j\} = \delta_{ij}$ with all other anticommutators zero. It is easy to see that this is $G$-invariant[14] and has $R$-charge 1. As explained in detail in [53], $M$ can be characterised in terms of "Wilson line branes": $M = \oplus \mathcal{W}(q_1, q_2, q_3)_r$ labelled by weights $q_i$, $r$ of irreducible representations of $G$ and $U(1)_V$, respectively. Combining this with the maps encoded in the matrix factorisation, the GLSM B-brane is characterised by a (twisted) complex of Wilson line branes. The object related to (67) that corresponds to the structure sheaf is

$$\mathcal{W}(0,0,0)_0 \ \underset{\longleftarrow}{\longrightarrow} \ \mathcal{W}(1,1,1)_1^{\oplus 3} \ \underset{\longleftarrow}{\longrightarrow} \ \mathcal{W}(2,2,2)_2^{\oplus 3} \ \underset{\longleftarrow}{\longrightarrow} \ \mathcal{W}(3,3,3)_3 \,. \tag{68}$$

The associated brane factor is

$$f_{\mathcal{B}} = -\left(-1 + e^{2\pi(\sigma_1 + \sigma_2 + \sigma_3)}\right) . \tag{69}$$

We evaluate the hemisphere partition function in the $\zeta \gg 0$-phase and express the result in terms of the Frobenius basis $\varpi_i^{(LV)}(\varphi)$ of mirror periods. The overall normalisation is not fixed, however it was demonstrated in [52] that the overall factor should include a scaling $|\mathcal{W}|$, where $\mathcal{W}$ is the Weyl group. Taking into account the $\mathbb{Z}_3$-factor in $G$, this suggests to set $C = i\frac{(2\pi)^6}{3}$. Then we get

$$Z_{D^2}(\mathcal{B}) = 5\varpi_3^{(LV)} + \frac{3}{2}\varpi_1^{(LV)} - \frac{30i\zeta(3)}{8\pi^3}\varpi_0^{(LV)} \,. \tag{70}$$

The central charge of the structure sheaf $\mathcal{O}_Y$ is

$$Z(\mathcal{O}_X) = \frac{H^3}{6}\varpi_3^{(LV)} + \frac{c_2 \cdot H}{24}\varpi_1^{(LV)} + \frac{\zeta(3)c_3}{(2\pi i)^3}\varpi_0^{(LV)} \,. \tag{71}$$

For the $\zeta \gg 0$-phase we read off:

$$H^3 = 30 \,, \qquad c_2 \cdot H = 36 \,, \qquad c_3 = -30 \,, \tag{72}$$

which is the expected result.

# 3 Considerations from mirror symmetry

To gain a better understanding of the Calabi-Yau $X$ in the $\zeta \ll 0$-phase of the GLSM, we make use of mirror symmetry and topological string theory. We will construct the mirror threefold $\lambda$ of $Y$ in §3.1, which has complex structure parameter $\varphi$. We verify that the values of $\varphi$ such that $\lambda$ has conifold/hyperconifold singularities are in agreement with our previous GLSM analysis.

We will eventually, in §3.3, make a number of claims on the nature of the second MUM point at $\varphi = \infty$. Before we do this, we use §3.2 to review a number of details surrounding MUM points. Some of this discussion will include very familiar items from [34–36]. We will also discuss transfer matrices and Kähler transformations, which are important to consider in examples with multiple MUM points as has already been illustrated in [12]. We will further recall certain discoveries of [2,4] that concern recently appreciated properties of MUM points.

In [4, §5.4], it is proposed that MUM points in mirror symmetry can be distinguished into *commutative* and *noncommutative* MUM points. This terminology is related to commutativity or noncommutativity of the underlying Calabi-Yau category. The authors explain that

---

[14]In particular, this brane is invariant with respect to that $\mathbb{Z}_3$-symmetry. It would be interesting to compare D-brane categories of Calabi-Yaus and their free quotients by some discrete group: the discrete symmetry forbids certain GLSM matrix factorisations that would exist for GLSMs without the $\mathbb{Z}_3$.

while about a commutative MUM point one can perform the usual BPS expansions using the Gopakumar-Vafa formula [64,65] to obtain invariants $n_\beta^{(g)}$, the noncommutative MUM points require a different treatment using the torsion refined GV formula [2,4]. We will here delineate MUM points into $N = 1$ and $N > 1$, according to whether the standard GV or the $\mathbb{Z}_N$-torsion refined GV formula should be applied, because we will be taking a bootstrap approach that is sensitive to the value of $N$ in a way that we aim to make clear to the reader. Further study is required to associate a noncommutative Calabi-Yau category to the $\zeta \ll 0$ phase. In fact, the discussion of [4] contains the more general possibility of torsion groups that differ to $\mathbb{Z}_N$, but this will not be relevant for our purposes and so we limit ourselves to discussing $\mathbb{Z}_N$ refinement.

The methods of [4] concern threefolds $X$ with nodal singularities that do not admit a global Kähler resolution. Applying the methods of [4] requires identifying a smooth deformation $X_{\text{def}}$, obtained from $X$ by complex structure deformation. Having identified problems in §3.3, we set about solving them in §3.4 by identifying this $X_{\text{def}}$. We read off topological data of $X_{\text{def}}$ from the topological string free energies, and use this to make our ansatz for this smooth deformation. This turns out to be a familiar hypergeometric threefold.

Having identified $X_{\text{def}}$, we go on in §3.5 and §3.6 to provide two integral symplectic bases of solutions to the PF equation, one basis attached to each MUM point, together with full sets of monodromy matrices.

## 3.1 The mirror threefold $\lambda$ and its Picard-Fuchs equation

The polynomials defining the $h^{1,1} = 3$ intersection (21) have an associated polytope $\Delta$. The Batyrev-Borisov procedure [66] produces the $h^{2,1} = 3$ mirror family $\widetilde{\lambda}$ by first forming a toric variety $\mathbb{P}_{\Delta^*}$. In our case, this variety is six-dimensional. Let $U_1, U_2, V_1, V_2, W_1, W_2$ be coordinates on the dense torus $\mathbb{T}^6 \subset \mathbb{P}_{\Delta^*}$. Then the mirror variety $\widetilde{\lambda}$ is birational to the mutual vanishing locus of the following Laurent polynomials:

$$
\begin{aligned}
F^1 &\equiv 1 - U_1 - V_1 - W_1, \\
F^2 &\equiv 1 - U_2 - V_2 - W_2, \\
F^3 &\equiv 1 - \frac{\varphi^1}{U_1 U_2} - \frac{\varphi^2}{V_1 V_2} - \frac{\varphi^3}{W_1 W_2}.
\end{aligned}
\tag{73}
$$

We will construct the mirror $\lambda$ of $Y$ by quotienting the mirror $\widetilde{\lambda}$ of $\widetilde{Y}$, after setting $\varphi^1 = \varphi^2 = \varphi^3 = \varphi$.

By a residue integral we can obtain the fundamental period of $\widetilde{\lambda}$, which is

$$
\varpi_0^{\widetilde{\lambda}}(\varphi^1, \varphi^2, \varphi^3) = \sum_{m_i \geq 0} \left( \frac{(m_1 + m_2 + m_3)!}{m_1! m_2! m_3!} \right)^3 (\varphi^1)^{m_1} (\varphi^2)^{m_2} (\varphi^3)^{m_3}.
\tag{74}
$$

On the locus $\varphi^i = \varphi$, $\widetilde{\lambda}$ has a freely acting $\mathbb{Z}_3$ symmetry with generator

$$
\begin{aligned}
U_1 &\mapsto V_1 \mapsto W_1 \mapsto U_1, \\
U_2 &\mapsto V_2 \mapsto W_2 \mapsto U_2.
\end{aligned}
\tag{75}
$$

This is freely acting for generic $\varphi$: indeed, a fixed point must have $U_1 = V_1 = W_1$, and then in light of (73) we must have $U_1 = 1/3$. Similarly, $U_2 = V_2 = W_2 = 1/3$. Then we see that the third equation in (73) is only satisfied for one value of $\varphi$, namely $\varphi = \frac{1}{27}$.

After setting $\varphi^i = \varphi$, we can search for septuples $(U_1, U_2, V_1, V_2, W_1, W_2, \varphi)$ such that the vanishing set of (73) is singular. That is, we seek solutions to

$$
F^1 = F^2 = F^3 = \mathrm{d}F^1 \wedge \mathrm{d}F^2 \wedge \mathrm{d}F^3 = 0.
\tag{76}
$$

We find seven solutions which we express as septuplets $(U_1, U_2, V_1, V_2, W_1, W_2, \varphi)$:

$$
\left(\frac{-i}{\sqrt{3}}, \frac{-i}{\sqrt{3}}, \omega, \omega, \omega, \omega, \varphi = \frac{i}{3\sqrt{3}}\right), \qquad \left(\frac{i}{\sqrt{3}}, \frac{i}{\sqrt{3}}, \overline{\omega}, \overline{\omega}, \overline{\omega}, \overline{\omega}, \varphi = \frac{-i}{3\sqrt{3}}\right),
$$

$$
\left(\omega, \omega, \frac{-i}{\sqrt{3}}, \frac{-i}{\sqrt{3}}, \omega, \omega, \varphi = \frac{i}{3\sqrt{3}}\right), \qquad \left(\overline{\omega}, \overline{\omega}, \frac{i}{\sqrt{3}}, \frac{i}{\sqrt{3}}, \overline{\omega}, \overline{\omega}, \varphi = \frac{-i}{3\sqrt{3}}\right),
$$

$$
\left(\omega, \omega, \omega, \omega, \frac{-i}{\sqrt{3}}, \frac{-i}{\sqrt{3}}, \varphi = \frac{i}{3\sqrt{3}}\right), \qquad \left(\overline{\omega}, \overline{\omega}, \overline{\omega}, \overline{\omega}, \frac{i}{\sqrt{3}}, \frac{i}{\sqrt{3}}, \varphi = \frac{-i}{3\sqrt{3}}\right), \qquad (77)
$$

$$
\left(\frac{1}{3}, \frac{1}{3}, \frac{1}{3}, \frac{1}{3}, \frac{1}{3}, \frac{1}{3}, \varphi = \frac{1}{27}\right), \qquad\qquad \omega = \frac{e^{\pi i/6}}{\sqrt{3}}.
$$

The three top left solutions furnish one $\mathbb{Z}_3$ orbit of three points. So do the three top right solutions. The seventh solution in the lower left gives a singular point that is fixed by the $\mathbb{Z}_3$ symmetry.

So, $\widetilde{\lambda}$ is singular if $\varphi \in \{\pm i/(3\sqrt{3}), 1/27\}$. In the cases $\varphi = \pm i/(3\sqrt{3})$, there are three separate nodal points where an $S^3$ shrinks. In the case $\varphi = 1/27$, there is only one shrinking $S^3$.

The quotient variety $\lambda$ is singular for $\varphi = \pm i/(3\sqrt{3})$, with one conifold point, where an $S^3$ collapses. This shrinking $S^3$ is the image of three distinct shrinking $S^3$s by the quotient map. For $\varphi = 1/27$, $\lambda$ acquires a hyperconifold point where a lens space $S^3/\mathbb{Z}_3$ shrinks. So in IIB string theory, the numbers of massless hypermultiplets at the conifold points $\varphi = \pm i/(3\sqrt{3})$ and $\varphi = 1/27$ are respectively 1 and 3. These numbers are the same as the numbers of Coulomb branches identified at these conifold points in §2.3, in agreement with the conjecture in [13, §6].

In setting $\varphi^i = \varphi$, we obtain the fundamental period of $\lambda$ from that of $\widetilde{\lambda}$ in (74) in accordance with the GLSM computation:

$$
\varpi_0^{\lambda}(\varphi) = \sum_{n=0}^{\infty} \left[ \sum_{m_i \geq 0}^{m_1+m_2+m_3=n} \left(\frac{(m_1+m_2+m_3)!}{m_1! m_2! m_3!}\right)^3 \right] \varphi^n. \qquad (78)
$$

This same period function has been obtained in [31]. This period is annihilated by the differential operator AESZ17 [28]:

$$
\begin{aligned}
\mathcal{L}^{\text{AESZ17}} &= \sum_{k=0}^{4} S_k(\varphi)\left(\varphi\frac{\mathrm{d}}{\mathrm{d}\varphi}\right)^k, \\
S_4 &= (1-27\varphi)(5-9\varphi)^2\left(1+27\varphi^2\right), \\
S_3 &= -36\varphi\,(5-9\varphi)\left(7-15\varphi+621\varphi^2-729\varphi^3\right), \\
S_2 &= -6\varphi\left(180-541\varphi+39591\varphi^2-91935\varphi^3+59049\varphi^4\right), \\
S_1 &= -6\varphi\left(75-155\varphi+34155\varphi^2-64233\varphi^3+39366\varphi^4\right), \\
S_0 &= -3\varphi\left(25-30\varphi+21060\varphi^2-32562\varphi^3+19683\varphi^4\right).
\end{aligned} \qquad (79)
$$

In line with what we have said so far, this operator is already understood within [28, 31] to annihilate the periods of $\widetilde{\lambda}$ on the locus $\varphi^i = \varphi$. We take it as the Picard-Fuchs operator for the quotient $\lambda$.

The singularities of this operator lie at $\varphi = 0$, $\varphi = \infty$, and the zeroes of $S_4(\varphi)$. Note that $\varphi = 1/27$ and $\varphi = \pm i/(3\sqrt{3})$ are roots of $S_4(\varphi)$, so that singularities of the geometry $\lambda$ are reflected in the operator. There is an additional apparent singularity of the operator at $\varphi = 5/9$, for which $\lambda$ is smooth, and a basis of power-series solutions (without logarithms) can be found about this point.

## 3.2 Background details on MUM points and refined GV invariants

We can construct Frobenius bases of solutions as expansions about the MUM points $\varphi = 0$ and $\varphi = \infty$. Where the latter is concerned, we will make a change of variables to $\phi(\varphi)$. The coordinates $\varphi$ and $\phi$ are related to each other and to the complexified FI parameter $t = \zeta - i\theta$ by[15]

$$-e^{-t} = \varphi = \frac{1}{729\phi}. \tag{80}$$

We will review some very familiar particulars of $N = 1$ MUM points in order to make clear why this does not apply to $\phi = 0$. Then we will recall relevant formulae from [4] that apply to $N > 1$ MUM points. Before doing either of these, let us first note some features common to both types of MUM point with our model in mind.

The point $\varphi = 0$ corresponds to the large volume point $\zeta \to \infty$ of the GLSM, and $\phi = 0$ corresponds to the hybrid point $\zeta \to -\infty$. So we distinguish the two Frobenius bases by superscripts $(LV)$ and $(H)$, which stand for "Large Volume" and "Hybrid".

$$
\begin{aligned}
\varpi_0^{(LV)} = \varpi_0^{\lambda}(\varphi) = {}& 1 + 3\varphi + 27\varphi^2 + 381\varphi^3 + 6219\varphi^4 \\
& + 111753\varphi^5 + 2151549\varphi^6 + O(\varphi^7), \\[4pt]
\varpi_1^{(LV)} = {}& \varpi_0^{(LV)} \log(\varphi) + a_1(\varphi), \\[4pt]
\varpi_2^{(LV)} = {}& \frac{1}{2}\varpi_0^{(LV)} \log(\varphi)^2 + a_1(\varphi)\log(\varphi) + \frac{1}{2}a_2(\varphi), \\[4pt]
\varpi_3^{(LV)} = {}& \frac{1}{6}\varpi_0^{(LV)} \log(\varphi)^3 + \frac{1}{2}a_1(\varphi)\log(\varphi)^2 + \frac{1}{2}a_2(\varphi)\log(\varphi) + \frac{1}{6}a_3(\varphi), \\[8pt]
\varpi_0^{(H)} = {}& \phi - 9\phi^2 - 837\phi^3 + 32553\phi^4 + 4787019\phi^5 - 253184859\phi^6 \\
& - 43950787299\phi^7 + O(\phi^8), \\[4pt]
\varpi_1^{(H)} = {}& \varpi_0^{(H)} \log(\phi) + b_1(\phi), \\[4pt]
\varpi_2^{(H)} = {}& \frac{1}{2}\varpi_0^{(H)} \log(\phi)^2 + b_1(\phi)\log(\phi) + \frac{1}{2}b_2(\varphi), \\[4pt]
\varpi_3^{(H)} = {}& \frac{1}{6}\varpi_0^{(H)} \log(\phi)^3 + \frac{1}{2}b_1(\phi)\log(\phi)^2 + \frac{1}{2}b_2(\phi)\log(\phi) + \frac{1}{6}b_3(\phi).
\end{aligned}
\tag{81}
$$

The series coefficients of $a_i(\varphi)$ in (81) can be obtained in a closed form using Gamma functions and higher order Harmonic numbers, by taking the closed form (78) for $\varpi_0^{(LV)}$ and applying

---

[15]The number 729 is the minimal choice of positive $\alpha$ such that, after effecting $\varphi = \frac{1}{\alpha\phi}$, the resulting fundamental period $\varpi_0^{(H)}$ in (81) has integral coefficients in its Taylor series. For later reference, our decision to take $\alpha > 0$ is informed by considerations from later on in this paper and so this comment may seem out of place on a first read: we do not find integral refined invariants at genus 1 with $\alpha = -729$.

the method of Frobenius. These series have expansions that start

$$
\begin{aligned}
a_1(\varphi) = {} & 6\varphi + 69\varphi^2 + 1037\varphi^3 + \frac{35373\varphi^4}{2} + \frac{3271887\varphi^5}{10} + O(\varphi)^6 \\
& + \frac{64218101\varphi^6}{10} + O(\varphi^7),
\end{aligned}
$$

$$
\begin{aligned}
a_2(\varphi) = {} & \frac{12\varphi}{5} + \frac{369\varphi^2}{5} + \frac{20677\varphi^3}{15} + \frac{528073\varphi^4}{20} + \frac{262129923\varphi^5}{500} \\
& + \frac{5388194759\varphi^6}{500} + O(\varphi^7),
\end{aligned}
\tag{82}
$$

$$
\begin{aligned}
a_3(\varphi) = {} & -\frac{72\varphi}{5} - 135\varphi^2 - \frac{20861\varphi^3}{15} - \frac{653827\varphi^4}{40} - \frac{1006248201\varphi^5}{5000} \\
& - \frac{12189676543\varphi^6}{5000} + O(\varphi^7).
\end{aligned}
$$

We do not have a closed form for $\varpi_0^{(H)}$. The first few terms of the series expansions for $b_i(\phi)$ are

$$
b_1(\phi) = -66\phi^2 - 2763\phi^3 + 231737\phi^4 + \frac{34305453\phi^5}{2} - \frac{18525326157\phi^6}{10} + O(\phi^7),
$$

$$
\begin{aligned}
b_2(\phi) = {} & -324\phi^2 - 6579\phi^3 + 1326591\phi^4 + \frac{185167221\phi^5}{4} \\
& - \frac{1103193247221\phi^6}{100} + O(\phi^7),
\end{aligned}
\tag{83}
$$

$$
\begin{aligned}
b_3(\phi) = {} & 1944\phi^2 + 24057\phi^3 - 3515751\phi^4 - \frac{650306151\phi^5}{8} \\
& + \frac{20867455404927\phi^6}{1000} + O(\phi)^7.
\end{aligned}
$$

MUM stands for Maximal Unipotent Monodromy: it can be seen that upon circling $\varphi = 0$ or $\phi = 0$ the bases of solutions in (81) transform by a triangular transformation $\varpi \mapsto U\varpi$, with the minimal integer $k$ such that $(U - \mathbb{I})^k = 0$ being $k = 4$.

There can be a mirror geometry associated to each MUM point, and each such manifold has its own set of topological data. We have already identified the mirror at $\varphi = 0$ as $Y$ (1), and will denote the mirror at $\varphi = \infty$ by $X$ (although as we shall explain, we do not anticipate that $X$ is a smooth manifold like $Y$).

The mirror maps about each MUM point are given by

$$
\begin{aligned}
t_0(\varphi) &= \frac{1}{2\pi i} \frac{\varpi_1^{(LV)}(\varphi)}{\varpi_0^{(LV)}(\varphi)} = \frac{1}{2\pi i} \log(\varphi) + O(\varphi), \\
t_\infty(\phi) &= \frac{1}{2\pi i} \frac{\varpi_1^{(H)}(\phi)}{\varpi_0^{(H)}(\phi)} = \frac{1}{2\pi i} \log(\phi) + O(\phi).
\end{aligned}
\tag{84}
$$

Higher genus B-model free energies can be obtained by solving the holomorphic anomaly equations [45, 46]. It is convenient to do this by using the polynomial method [42]. The holomorphic anomaly equations only fix the genus $g$ free energy up to a holomorphic ambiguity $f^{(g)}(\varphi)$ which can be fixed by incorporating data from singular degenerations of $\lambda$ and Castelnuovo vanishing of the Gopakumar-Vafa invariants of the mirror manifolds associated to MUM points, as pioneered in [41] and further developed in [43, 44]. A crucial source of boundary

data is the conifold gap condition [41], which completely fixes the terms in $f^{(g)}(\varphi)$ that are polar at conifold points once the correct normalisation of the mirror map about a conifold is known. An observation in [67, §8.7] provides a means of obtaining this correct normalisation, which in our example is necessary for us to compute any higher genus free energies for $\lambda$. Once the B-model free energies are known, the A-model free energies are computed via

$$F^{(g)}_{\text{A-model}}(t) = (\eta \varpi_0)^{2g-2} \mathcal{F}^{(g)}. \tag{85}$$

In this equation, $\eta$ is a number that gives the necessary change of gauge for $F^{(g)}$ to be a generating function of BPS invariants of $Y$ (or $X$, depending on which MUM point we expand around). One way to characterise $\eta$ is through the Yukawa couplings. When expanded in the $t$-coordinates, we require

$$\widehat{C}_{t_0 t_0 t_0} = (2\pi i)^3 \left( \kappa^{(Y)}_{111} + \text{n.p.} \right), \qquad \widehat{C}_{t_\infty t_\infty t_\infty} = (2\pi i)^3 \left( \kappa^{(X)}_{111} + \text{n.p.} \right), \tag{86}$$

with n.p. denoting nonperturbative genus 0 instanton contributions. Beginning from the B-model Yukawa coupling

$$C_{\varphi\varphi\varphi} \equiv -\int_{\lambda} \Omega \wedge \partial^3_{\varphi\varphi\varphi} \Omega = \frac{30\left(1 - \frac{9}{5}\varphi\right)}{\varphi^3(1 - 27\varphi)(1 + 27\varphi^2)}, \tag{87}$$

one can compute $\widehat{C}_{t_0 t_0 t_0}$, with a change of gauge, by the usual formula [34]

$$\widehat{C}_{t_0 t_0 t_0}(t_0) = \left( \varpi^{(LV)}_0 \right)^{-2} \left( \frac{\mathrm{d}\varphi}{\mathrm{d}t_0} \right)^3 C_{\varphi\varphi\varphi}(\varphi(t_0)). \tag{88}$$

The hat on $\widehat{C}_{t_0 t_0 t_0}$ is to indicate that we have not only made a general coordinate transformation, but also a change of Kähler gauge (the division by $(\varpi^{(LV)}_0)^2$ in (88)). Now, $C_{\varphi\varphi\varphi}$ is a rational function of $\varphi$, and so can be expanded about $\phi = 0$ after the tensor transformation

$$C_{\phi\phi\phi}(\phi) = \left( \frac{\mathrm{d}\varphi}{\mathrm{d}\phi} \right)^3 C_{\varphi\varphi\varphi}(\varphi(\phi)). \tag{89}$$

When we apply (88), with $\phi, t_\infty, \varpi^{(H)}_0$ instead of $\varphi, t_0, \varpi^{(LV)}_0$, our leading term will not in general be $\kappa^X_{111}$. Dividing by some number $\eta^2$ resolves this discrepancy. The reason for this is that the B-model has as gauge symmetry the Kähler transformations $\Omega \mapsto f(\varphi)\Omega$. In writing (87), we chose a gauge such that the correct asymptotics for $\widehat{C}_{t_0 t_0 t_0}$ are produced (as in (86)) after making the transformation (88). There is no guarantee that this gauge leads to the correct asymptotics for $\widehat{C}_{t_\infty t_\infty t_\infty}$. So

$$\widehat{C}_{t_\infty t_\infty t_\infty} = \left( \eta \varpi^{(H)}_0 \right)^{-2} \left( \frac{\mathrm{d}\phi}{\mathrm{d}t_\infty} \right)^3 C_{\phi\phi\phi}(\phi(t_\infty)). \tag{90}$$

The necessary division by $\eta^2$ is the gauge transformation $\Omega \mapsto \eta^{-1}\Omega$, and since the genus-$g$ free energy transforms with Kähler weight $2 - 2g$, one arrives at (85).

An equivalent characterisation of $\eta$ is in terms of the integral symplectic bases $\Pi^{(LV)}$ and $\Pi^{(H)}$ of periods, built from the Frobenius periods about each MUM point and the corresponding mirror geometric data, as reviewed in the following subsection. A transfer matrix $\widetilde{\mathrm{T}}$ will relate the two bases via $\Pi^{(H)} = \widetilde{\mathrm{T}}\Pi^{(LV)}$. $\widetilde{\mathrm{T}}$ will not be an integral symplectic matrix. However, it can be written

$$\widetilde{\mathrm{T}} = \frac{1}{\eta}\mathrm{T}, \tag{91}$$

where T is a symplectic matrix with unit determinant and integer entries.

### 3.2.1  $N = 1$ MUM points

These MUM points can be thought of informally as the "ordinary MUM points", for which the analyses of [34, 36] hold. Since in our model the point $\varphi = 0$ is such an $N = 1$ MUM point (by design, as we constructed $\text{Ⴗ}$ as the mirror to the smooth family $Y$), we will here use the symbol $Y$ to denote the mirror geometry associated to an $N = 1$ MUM point in general, and not write the superscript $(Y)$ on $\kappa_{111}$ and $c_2$.

The genus 0 A-model free energy reads

$$
\begin{aligned}
\frac{1}{(2\pi i)^3} F^{(0)}(t) = & -\frac{1}{6}\kappa_{111}t^3 - \frac{1}{2}Y_{011}t^2 + \frac{c_2}{24}t + \frac{\chi(Y)}{2}\frac{\zeta(3)}{(2\pi i)^3} \\
& -\frac{1}{(2\pi i)^3}\sum_{\beta\in H_2(Y,\mathbb{Z})} n_\beta^{(0)}\text{Li}_3\left(e^{2\pi i\beta\cdot t}\right),
\end{aligned}
\tag{92}
$$

with $Y_{011} = \frac{1}{2}(\kappa_{111}\bmod 2)$.

The topological data of the manifold $Y$ can be used to create a change of basis matrix that we write as a product of two matrices M and $\rho$. This matrix M$\rho$ has the property that upon circling a singularity in the $\varphi$-plane, the vector of functions $\Pi(\varphi) = \text{M}\rho\varpi$ transforms by a symplectic matrix with integer entries. These matrices are

$$
\text{M} = \begin{pmatrix} \frac{\chi\zeta(3)}{(2\pi i)^3} & \frac{c_2}{24} & 0 & \kappa_{111} \\ \frac{c_2}{24} & -Y_{011} & -\kappa_{111} & 0 \\ 1 & 0 & 0 & 0 \\ 0 & 1 & 0 & 0 \end{pmatrix}, \qquad \rho = \begin{pmatrix} 1 & 0 & 0 & 0 \\ 0 & \frac{1}{2\pi i} & 0 & 0 \\ 0 & 0 & \frac{1}{(2\pi i)^2} & 0 \\ 0 & 0 & 0 & \frac{1}{(2\pi i)^3} \end{pmatrix}.
\tag{93}
$$

This matrix M$\rho$ is chosen so that

$$
\Pi(\varphi(t)) = \frac{\varpi_0(\varphi(t))}{(2\pi i)^3}\begin{pmatrix} 2F^{(0)}(t) - t\partial_t F^{(0)}(t) \\ t\partial_t F^{(0)}(t) \\ 1 \\ t \end{pmatrix} = \frac{1}{(2\pi i)^3}\begin{pmatrix} \mathcal{F}_0(X^0(\varphi), X^1(\varphi)) \\ \mathcal{F}_1(X^0(\varphi), X^1(\varphi)) \\ X^0(\varphi) \\ X^1(\varphi) \end{pmatrix}.
\tag{94}
$$

The rightmost formula above introduces an integral symplectic basis of periods $\{\mathcal{F}_I, X^I\}$ that can be computed by integrating the holomorphic three form of $X$ along an integral symplectic basis $\{A^I, B_I\}$ of three-cycles on $X$:

$$
\mathcal{F}_I = \int_{B_I} \Omega(\varphi), \qquad X^I = \int_{A^I} \Omega(\varphi).
\tag{95}
$$

Special geometry implies [34, 68] that the $\mathcal{F}_I$ can be expressed as derivatives of the genus 0 B-model free energy $\mathcal{F}$, which is a function of the conjugate periods $X^I$.

Higher genus B-model free energies $\mathcal{F}^{(g)}$ can be computed by solving the holomorphic anomaly equations [45, 46]. Higher genus A-model free energies, by the Gopakumar-Vafa formula [64, 65], encode the higher genus GV invariants $n_\beta^{(g)}$ as

$$
\begin{aligned}
F^{\text{All Genus}}(t, \lambda) &= \sum_{g=0}^\infty \lambda^{2g-2} F^{(g)}(t) \\
&= \lambda^{-2}c(t) + l(t) + \sum_{g=0}^\infty\sum_{\beta\in H_2(Y,\mathbb{Z})}\sum_{m=1}^\infty n_\beta^{(g)}\frac{1}{m}\left(2\sin\frac{m\lambda}{2}\right)^{2g-2}q^{m\beta}.
\end{aligned}
\tag{96}
$$

Here $c(t)$ is the polynomial part of the genus 0 free energy (92). $l(t) = -\frac{c_2}{24}t$ is the classical part of the genus 1 free energy (which is only defined up to an additive constant).

By using the solution approach to the holomorphic anomaly equations developed in [41], and then applying (85) and comparing with (96), it becomes possible to compute the $n_k^{(g)}$ to as high a genus as available boundary data allows. The boundary data coming from the gap behaviour at the three conifold points, plus the constant term in the expansion about $\varphi = 0$, plus the Castelnuovo vanishing of the invariants $n_\beta^{(g)}$ of $Y$, are sufficient for us to compute A-model topological string free energies for $Y$ up to genus 4. We display the resulting $n_\beta^{(g)}$ in the first row[16] of Table 2, Appendix §B.

### 3.2.2 $N > 1$ MUM points

It can happen that there is no choice of $\kappa_{111}$ and $c_2$ such that there is an integral basis of solutions as described in the previous $N = 1$ discussion, under the assumption that the same Euler characteristic should be attached to all MUM points within a common moduli space. Moreover, the $n_k^{(g)}$ prescribed in (96) may not be integers. This phenomenon has been addressed in [2], and later in [4] where the authors study the possibility that the mirror geometry $X$ associated to a MUM point is not smooth. Instead, to a MUM point one can associate a singular threefold $X$ with some number of nodal singularities, together with a topologically nontrivial flat $B$-field which obstructs the complex structure deformations that would remove these singularities.

These singularities have the property that there is no global resolution that is Calabi-Yau and Kähler. The article [4] considers resolutions $\widehat{X}$ which are non-Kähler, have vanishing first Chern class, and have exceptional curves which are torsional. On $\widehat{X}$, the $B$-field is a two-form valued in $H^2(\widehat{X}, U(1))$. It is explained in [4] that the distinct choices of such a flat $B$-field are labelled by the elements of $H^3(\widehat{X}, \mathbb{Z})_{\text{Torsion}} \cong H_2(\widehat{X}, \mathbb{Z})_{\text{Torsion}}$. The authors of [4] consider the A-model topological string on backgrounds $(\widehat{X}, [k])$, where $\widehat{X}$ is a non-Kähler resolution, with vanishing first Chern class, of a nodal Kähler Calabi-Yau threefold $X$ with $H_2(\widehat{X}, \mathbb{Z}) \cong H_2(X, \mathbb{Z}) \oplus \mathbb{Z}_N$, and $[k] \in \text{Tors}(H^3(\widehat{X}, \mathbb{Z})) \cong \mathbb{Z}_N$ gives the class of the B-field.

The proposal of [2, 4] is that the topological string partition function on $(\widehat{X}, [k])$ encodes integer BPS invariants (these are called "torsion refined Gopakumar-Vafa invariants") $n_{\beta,l}^{(g)}$ via

$$
\begin{aligned}
F_k^{\text{All Genus}}(t, \lambda) &= \sum_{g=0}^{\infty} \lambda^{2g-2} F_k^{(g)}(t) \\
&= \lambda^{-2} c^{(N,k)}(t) + l(t) + \sum_{g=0}^{\infty} \sum_{l=0}^{N-1} \sum_{\beta \in H_2(Y,\mathbb{Z})} \sum_{m=1}^{\infty} n_{\beta,l}^{(g)} \frac{e^{\frac{2\pi i m l k}{N}}}{m} \left( 2 \sin \frac{m\lambda}{2} \right)^{2g-2} q^{m\beta} .
\end{aligned}
\tag{97}
$$

The subscript $k = 0$ means that there is no $B$-field, and so no obstruction to the deformations of $X$ that produce a smooth Calabi-Yau threefold $X_{\text{def}}$, such deformations are guaranteed by a theorem of Namikawa and Steenbrink [69]. The A-model topological string free energy is insensitive to complex structure deformations, and so the $k = 0$ expansion (97) is the 'ordinary' A-model free energy for the smooth target manifold $X_{\text{def}}$. Knowledge of $X_{\text{def}}$ and its BPS expansions is necessary in order to extract the $n_{\beta,l}^{(g)}$, which can only be read off when the full set of $F_k^{(g)}$, $0 \le k \le \lfloor N/2 \rfloor$ is known. For each $k$, there is a different MUM point to expand about belonging to a *different* moduli space. It is quite remarkable that, as shown in [2, 4], integer invariants $n_{\beta,l}^{(g)}$ can be computed by comparing the topological string free energies in expansions about various MUM points in this manner. The extra label $k \in \mathbb{Z}_N$ on $n_{\beta,l}^{(g)}$

---

[16]Although these aforementioned boundary conditions only enable us to reach genus 4, Table 2 in fact goes up to genus 11. This is because we have incorporated additional boundary conditions from the second MUM point with the results of §3.4, as we detail in §A when we discuss torsion-refined invariants.

corresponds to the torsion in $H_2(\widehat{X}, \mathbb{Z})$, so that the sum over $(\beta, l)$ is a sum over the entire second homology $H_2(\widehat{X}, \mathbb{Z})$. At genus 0, the possibility of supporting instanton numbers on torsional curve classes was raised in [70, 71].

For the noncommutative resolutions studied in the work [2] that originally introduced the torsion-refined GV formula (97), the leading linear term in the genus-one free energy could be independently fixed by modularity and was found to equal $\frac{-c_2^{X_{\text{def}}}}{24} \cdot \mathbf{t}$ (note that these examples had two Kähler parameters, unlike the present $h^{11} = 1$ case). Based on that result, the authors of [4] expected that $c_2(\widehat{X}) = c_2(X_{\text{def}})$ would hold for their octic example. We remark that if a $h^{11} = 2$ CY3 $A$ with Kähler parameters $(t_1, t_2)$ admits a conifold transition to $B$ by first sending $t_2$ to 0 and then deforming away the singularities, then $c_2(B) = (c_2(A))_1$. Based on this, although we do not explicitly construct any resolution $\widehat{X}$, we anticipate that $c_2(\widehat{X}) = c_2(X_{\text{def}})$ and that this topological invariant can be read off from the leading term in the genus-one free energy. Similarly, the triple intersection number $\kappa_{111}$ is anticipated to be the same for all three spaces. The Euler characteristics differ, they are related by the formula

$$\chi(X_{\text{def}}) = \chi(\widehat{X}) - 2m_s \,, \tag{98}$$

where $m_s$ is the number of nodes on $X$. The genus-1 linear term $l(t) = \frac{-c_2^{X_{\text{def}}}}{24} t$ in (97) is insensitive to the choice of $k$. The perturbative genus-0 piece $c^{(N,k)}(t)$ is a cubic polynomial[17] in $t$,

$$\frac{1}{(2\pi i)^3} c^{(N,k)}(t) \;=\; -\frac{1}{6} \kappa_{111}^{X_{\text{def}}} t^3 - \frac{1}{2} Y_{011} t^2 + \frac{c_2^{X_{\text{def}}}}{24} t + C_{N,k} \frac{\zeta(3)}{(2\pi i)^3} \,. \tag{99}$$

$c^{(N,k)}$ does depend on $k$, but only through the constant term $C_{N,k}$ which replaces $\chi/2$ in (92). This replacement occurs because in this setup, with singularities supporting fractional $B$-fields, the constant map (degree 0) contribution to the A-model free energies differs to the smooth case. In [4] it is argued that, locally, each singularity together with the supported $B$-field is modelled by a noncommutative conifold.

The Donaldson-Thomas partition function for the resolved conifold is $Z_{\text{DT},A_{\pm}}(t, \lambda)$, where $A_{\pm}$ is either of the two independent resolutions. This depends on the parameter $t = b + i\nu$ and the string coupling $\lambda$. This $t$ is the complexification of the volume $\nu$ of the exceptional curve in the resolution. It is explained in [4] that sending $t$ to zero restores the conifold singularity, but sending $\nu$ to zero while $b$ is kept nonzero leads to a situation where the exceptional curve has zero volume but the string worldsheet physics is regular. As was studied in [72], the volume parameter $\nu$ can be sent to zero so that the partition function becomes, in the form written in [4],

$$\log\left[Z_{\text{DT},A}(b, \lambda)\right] = \sum_{m=1}^{\infty} \frac{1}{m} \frac{e^{2\pi i m b} - 2 + e^{-2\pi i m b}}{4 \sin\left(\frac{m\lambda}{2}\right)^2} \,. \tag{100}$$

This is the partition function of the noncommutative conifold [72]. From this, the constant terms in each topological string free energy are found in [4] to be given by

$$F_k^{\text{All Genus}}(t, \lambda)\big|_{\text{constant}} \;=\; F_0^{\text{All Genus}}(t, \lambda)\big|_{\text{constant}} - \frac{1}{2} \sum_{l=1}^{\lfloor N/2 \rfloor} m^{(\pm l)} \log\left[Z_{\text{DT},A}(kl/N, \lambda)\right] \,. \tag{101}$$

Recalling that $X$ has $m_s$ nodes, and so $\widehat{X}$ has $m_s$ exceptional curves which are pure torsion in homology, $m^{(\pm l)}$ denotes the number of these torsion exceptional curves with $\mathbb{Z}_N$ charge $\pm l$. For the details of the M-theory gauge group and charge lattice, see [4]. One has that $m_s = \sum_{l=1}^{\lfloor N/2 \rfloor} m^{(\pm l)}$. Further, $m^{(0)} = 0$ since there are no homologically trivial exceptional curves on $\widehat{X}$, see [4].

---

[17]We have included the quadratic term $Y_{011} t^2$, which equalled 0 for the main example in [4].

### 3.3 Failure of integrality

Let us now consider the MUM point at infinity belonging to the operator $\mathcal{L}^{\text{AESZ17}}$. We will give two arguments as to why this point $\phi = 0$ cannot be an $N = 1$ MUM point. The first of these uses monodromies, this argument has been made before in [29] and we repeat it so that we can present its solution later on. The second argument makes use of the new topological string free energies that we compute in Appendix §A.

**The monodromy argument:** If the MUM point $\phi = 0$ is of the $N = 1$ type, then there must exist a choice of integers $\kappa_{111}$, $c_2$, $\chi$ such that $M\rho\varpi^{(H)}$ has integral monodromies, with M defined in (93). But we know more: since $\text{\th}$ is the mirror of the large volume geometry $Y$ we must have $\chi(\text{\th}) = -\chi(Y) = 30$. Then, whatever geometry $X$ is attached to the MUM point $\phi = 0$ must have $\chi(X) = -\chi(\text{\th}) = \chi(Y) = -30$ since the Hodge diamond of $X$ will be the same as the Hodge diamond of $Y$ (both are the transpose of the Hodge diamond of $\text{\th}$). Relatedly, $\chi(Y)$ is the Witten index for the nonlinear sigma model with target space $Y$ [73]. One can reach the $\zeta \ll 0$ phase of the GLSM from the $\zeta \gg 0$ phase by varying $\zeta$, which changes the D-terms but does not change the F-terms. The Witten index is invariant under D-term variations, and so must take the same value in each phase. In particular, two smooth geometric phases of any GLSM always have the same Euler characteristics.

For the purposes of numerical work, we introduce another basis of periods. $\widehat{\varpi}^{(H)} = \rho.\varpi^{(H)}$, with $\rho$ given in (93) and $\varpi^{(H)}$ given in (81).

We now show that the assumption that $\phi = 0$ is an $N = 1$ MUM point is inconsistent. We can perform a numerical analytic continuation to compute a monodromy matrix $\widehat{\text{n}}^{(H)}$ such that $\widehat{\varpi}^{(H)} \mapsto \widehat{\text{n}}^{(H)}\widehat{\varpi}^{(H)}$ upon circling $\phi = \frac{-i}{81\sqrt{3}}$. We then conjugate this matrix by M from (93) while leaving $\kappa_{111}$, $c_2$, $\chi$, and $Y_{011}$ as indeterminates. This matrix $M\widehat{\text{n}}^{(H)}M^{-1}$ will be the monodromy matrix of $\Pi^{(H)} = (2\pi i)^{-3}M.\widehat{\varpi}^{(H)}$. The overall factor of $(2\pi i)^{-3}$ does not affect monodromies. Our numerical analysis reveals that

$$M\,\text{n}^{(H)}M^{-1} = \begin{pmatrix} -\frac{1}{2} - 18\nu & 0 & \frac{\kappa_{111}}{8}(1 + 12\nu)^2 & \frac{c_2}{16}(1 + 12\nu) \\ -\frac{3c_2}{4\kappa_{111}} & 1 & \frac{c_2}{16}(1 + 12\nu) & \frac{c_2^2}{32\kappa_{111}} \\ -\frac{18}{\kappa_{111}} & 0 & \frac{5}{2} + 18\nu & \frac{3c_2}{4\kappa_{111}} \\ 0 & 0 & 0 & 1 \end{pmatrix},$$

$$\nu = \left(\frac{\chi}{\kappa_{111}} - 13\right)\frac{\zeta(3)}{(2\pi i)^3}. \tag{102}$$

$Y_{011}$ has dropped out. If $\chi$, $\kappa_{111}$, $c_2$ are rational, then in order for this matrix to have rational entries we must choose $\kappa_{111} = \chi/13 = -30/13$. Our assumption has led to a non-integral triple intersection number. Therefore, $\phi = 0$ cannot be an $N = 1$ MUM point.

**The argument from topological string theory:** Another argument shows that $\phi = 0$ cannot be an $N = 1$ MUM point. This is independent of our above monodromy computations, and also independent of our choice of factor 729 in $\varphi = 1/(729\phi)$. If $\phi = 0$ were an $N = 1$ MUM point, then there would be a choice of $\eta$ in (85) such that the A-model free energies had constant terms given by the formula due to [64, 74, 75], which we display in (A.11). We have at our disposal the B-model free energy expansions about $\phi = 0$ at genera 2,3, and 4 from (A.24). Let us attempt to find an $\eta$ such that

$$\left(\eta\varpi_0^{(H)}(\phi)\right)^{2g-2}\mathcal{F}^{(g)}(\phi)|_{\phi=0} \stackrel{?}{=} \frac{(-1)^{g-1}B_{2g}B_{2g-2}}{2g(2g-2)(2g-2)!} \cdot \frac{\chi(Y)}{2}. \tag{103}$$

The question mark over the equality indicates that we do not claim this is true. Using the explicit expansions (A.24), the condition (103) gives us a different value of $\eta$ at each genus:

$$g = 2 : \frac{89\eta^2}{18895680} \overset{?}{=} \frac{1}{2880} \cdot \frac{\chi(Y)}{2} \implies \eta \overset{?}{=} \left(-\frac{3^9 \cdot 5}{89}\right)^{1/2} ,$$

$$g = 3 : \frac{169\eta^4}{18744952939776} \overset{?}{=} -\frac{1}{725760} \cdot \frac{\chi(Y)}{2} \implies \eta \overset{?}{=} \left(\frac{3^{18}}{13^2}\right)^{1/4} , \qquad (104)$$

$$g = 4 : \frac{7649\eta^6}{110687072614083302400} \overset{?}{=} \frac{1}{43545600} \cdot \frac{\chi(Y)}{2} \implies \eta \overset{?}{=} \left(-\frac{3^{27} \cdot 5}{7649}\right)^{1/6} .$$

This procedure does not consistently return an $\eta$, and each such $\eta$ in (104) actually leads to nonintegral GV invariants from (96). Note that if we attempt to solve the three equations in (104) for $\eta$ and $\chi(Y)$, the only solution is $\eta = \chi = 0$.

## 3.4 Identifying the smooth deformation

Given the failure described in the previous subsection, we shall assume that $\phi = 0$ is an $N > 1$ MUM point, and attempt to apply the torsion-refined formalism of [4]. We do not yet have any geometric realisation of a singular threefold in the GLSM at large negative FI parameter, and proceed solely by making consistency checks with the formulae presented in [4].

We shall use what we know to arrive at a candidate smooth deformation $X_{\text{def}}$. We have at our disposal the genera 0, 1, 2, 3, and 4 topological string free energies. Genus 4 is the highest genus we can reach with information purely coming from the conifold gap conditions [41] and the MUM point $\varphi = 0$ which provides the Castelnuovo vanishing of GV invariants of $Y$ and the constant term in each expansion about $\varphi = 0$. Once we have a candidate $X_{\text{def}}$, this will provide us new information with which to solve the holomorphic anomaly equations at genera beyond 4 and in so doing we make nontrivial checks of our proposed geometry.

We shall identify a candidate smooth deformation from its classical topological data. We must obtain the second Chern number $c_2^{(X_{\text{def}})}$, the scaling $\eta$ in (85), the number of nodes $m_s$, and the triple intersection number $\kappa_{111}^{(X_{\text{def}})}$. Note that we must have $h^{1,1}(X_{\text{def}}) = h^{2,1}(\mathbb{X}) = 1$. The Euler characteristic $\chi(X_{\text{def}})$, and therefore $h^{2,1}(X_{\text{def}})$, will be obtained from (98) once we know $m_s$.

**Obtaining the second Chern number** We will read off $c_2^{(X_{\text{def}})}$ from the genus 1 topological string free energy $\mathcal{F}^{(1)}$. Recall that $\mathcal{F}^{(1)}$ solves the genus 1 holomorphic anomaly equation [45, 46, 76]. To begin with we will work in the $\varphi$-coordinate, and then transform our expressions into the $\phi$-coordinate. The genus 1 HAE reads

$$\partial_{\overline{\varphi}}\partial_\varphi \mathcal{F}^{(1)}(\varphi, \overline{\varphi}) = \frac{1}{2} C_{\varphi\varphi\varphi}(\partial_{\overline{\varphi}} \mathcal{S}^{\varphi\varphi}) + \left(1 - \frac{\chi(Y)}{24}\right) \mathcal{G}_{\overline{\varphi}\varphi} . \qquad (105)$$

$\mathcal{S}^{\varphi\varphi}$ is one of the BCOV propagators, which we review and construct in §A. $\mathcal{G}_{\overline{\varphi}\varphi}$ is the metric on moduli space, which is obtained by differentiating the Kähler potential $\mathcal{K}$ as $\mathcal{G}_{\overline{\varphi}\varphi} = \partial_{\overline{\varphi}}\partial_\varphi \mathcal{K}$.

Recall further that (105) is solved by integrating with respect to $\overline{\varphi}$ and $\varphi$, at the expense of introducing the genus 1 holomorphic ambiguity $f^{(1)}$.

$$\mathcal{F}^{(1)}(\varphi, \overline{\varphi}) = \frac{1}{2}\left(3 + h^{1,1}(Y) - \frac{\chi(Y)}{24}\right)\mathcal{K} - \frac{1}{2}\log\left(\det(\mathcal{G}_{\varphi\overline{\varphi}})\right) + f^{(1)}(\varphi) + \overline{f^{(1)}(\varphi)} . \qquad (106)$$

$f^{(1)}$ is specified, up to an additive constant by which the genus 1 free energy remains undetermined, by the behaviour of $\mathcal{F}^{(1)}$ at the boundaries of moduli space. Let us briefly review this.

At a conifold point $\varphi_c$, where a number $|G_c|$ of massless hypermultiplets arise in IIB string theory, $\mathcal{F}^{(1)}$ diverges as $-\frac{|G_c|}{12}\log(\varphi - \varphi_c)$ [77]. We have argued, both from analysing the mirror geometry and a GLSM computation, that our conifold point $1/27$ has $|G_c| = 3$ and the points $\pm i/\sqrt{27}$ have $|G_c| = 1$. At a MUM point $\varphi = 0$ mirror to a geometry $Y$, $\mathcal{F}^{(1)}$ diverges as $-\frac{c_2^{(Y)}}{24}\log(\varphi)$ [45, 46]. These conditions are satisfied in our example by taking

$$f^{(1)}(\varphi) = \frac{1}{12}\log\left(\frac{\varphi^{-6-c_2^Y/2}}{(1-27\varphi)^3(1+27\varphi^2)}\right). \tag{107}$$

The $-6$ is included to cancel with a divergence at $\varphi = 0$ provided by other terms in (106), so that the sum of all terms has the correct behaviour. Adding any nonconstant holomorphic function of $\varphi$ will introduce a new pole somewhere, so the above $f^{(1)}$ is correct.

We now stress the following important point: our moduli space has two MUM points and three conifold points. We have completely determined $f^{(1)}$ by using data attached to one MUM point and three conifold points. This means that we can use the resulting $\mathcal{F}^{(1)}$ to read off the boundary behaviour at the remaining MUM point $\phi = 0$, which will provide us with $c_2^{(X_{\text{def}})}$. To this end, it is more convenient to consider the first derivative $\partial_\varphi \mathcal{F}^{(1)}$ in the holomorphic limit, which we explain in more detail in §A. First notice that we can recover the second Chern number of $Y$ from

$$\begin{aligned}
\partial_\varphi \mathcal{F}^{(1)} &= \frac{1}{2}C_{\varphi\varphi\varphi}S^{\varphi\varphi} + \left(1 - \frac{\chi(Y)}{24}\right)K_\varphi + \partial_\varphi\left[\frac{1}{12}\log\left(\frac{\varphi^{-6-c_2^Y/2}}{(1-27\varphi)^3(1+27\varphi^2)}\right)\right] \\
&= -\frac{36}{24\varphi} - 6 - \frac{105}{2}\varphi - 531\varphi^2 + \frac{7353}{2}\varphi^3 + 433134\varphi^4 + \dots,
\end{aligned} \tag{108}$$

where the coefficient of $\frac{-1}{24\varphi}$ is our known second Chern number $c_2^{(Y)} = 36$. $S^{\varphi\varphi}(\varphi)$ is the propagator $\mathcal{S}(\varphi, \overline{\varphi})$ in the holomorphic limit, we provide this in §A. $K_\varphi = -\partial_\varphi \log(\varpi_0^{(LV)})$ is the first derivative of the Kähler potential $\mathcal{K}$ in the holomorphic limit.

Notice that every term in (108) is a tensor, which we are able to transform to the $\phi$-coordinate. This will involve a propagator $S^{\phi\phi}$ which we provide in §A, and $K_\phi = -\partial_\phi \log(\varpi_0^{(H)})$. We obtain

$$\begin{aligned}
\partial_\phi \mathcal{F}^{(1)} &= \frac{1}{2}C_{\phi\phi\phi}S^{\phi\phi} + \left(1 - \frac{\chi(Y)}{24}\right)K_\phi + \partial_\phi\left[\frac{1}{12}\log\left(\frac{\varphi^{-6-c_2^Y/2}}{(1-27\varphi)^3(1+27\varphi^2)}\right)\right] \\
&= \frac{1}{2}\left(\frac{\mathrm{d}\varphi}{\mathrm{d}\phi}\right)^3\left(C_{\varphi\varphi\varphi}\big|_{\varphi=\frac{1}{729\phi}}\right)S^{\phi\phi} - \left(1 - \frac{\chi(Y)}{24}\right)\partial_\phi \log(\varpi_0^{(H)}) \\
&\quad + \frac{\mathrm{d}\varphi}{\mathrm{d}\phi}\partial_\varphi\left[\frac{1}{12}\log\left(\frac{\varphi^{-6-c_2^Y/2}}{(1-27\varphi)^3(1+27\varphi^2)}\right)\right]\Bigg|_{\varphi=\frac{1}{729\phi}} \\
&= -\frac{32}{24\phi} + 69 + \frac{2295}{2}\phi - 86490\phi^2 - \frac{166428783}{2}\phi^3 + 6343497909\phi^4 + \dots
\end{aligned} \tag{109}$$

The coefficient of $\frac{-1}{24\phi}$ provides the result that we sought:

$$c_2^{X_{\text{def}}} = 32. \tag{110}$$

**Obtaining $\eta$ and the number of nodes** We saw in §3.3 that there was no choice of $\eta$ such that our topological string expansions $F^{(g=2,3,4)}(t_\infty)$ had the $N = 1$ constant term

$\frac{(-1)^{g-1}B_{2g}B_{2g-2}}{2g(2g-2)(2g-2)!} \cdot \frac{\chi(Y)}{2}$. However, since the constant term is different in the $N > 1$ expansions due to the correction formula (101) of [4], there is hope. In the $N = 3$ case[18] that we are considering, the constant terms are

$$
\begin{aligned}
&F_1^{(g)}(t_\infty)\big|_{\text{Constant}} \\
&= \frac{(-1)^{g-1}B_{2g}\left[B_{2g-2}\frac{\chi(Y)}{2} + m_s(g-1)\left(\text{Li}_{3-2g}(e^{-2\pi i/3}) + \text{Li}_{3-2g}(e^{+2\pi i/3})\right)\right]}{2g(2g-2)(2g-2)!}.
\end{aligned}
\tag{111}
$$

Note that setting $m_s = 0$ recovers (A.11). Let us once again plug our expansions $\mathcal{F}^{(g=2,3,4)}(\phi)$ from §A into (85), but now ask that they reproduce (111). This gives

$$
\begin{aligned}
g &= 2: &\frac{89\eta^2}{18895680} &= \frac{1}{2880} \cdot \frac{\chi(Y)}{2} - \frac{m_s}{720}, \\
g &= 3: &\frac{169\eta^4}{18744952939776} &= -\frac{1}{725760} \cdot \frac{\chi(Y)}{2} + \frac{m_s}{18144}, \\
g &= 4: &\frac{7649\eta^6}{110687072614083302400} &= \frac{1}{43545600} \cdot \frac{\chi(Y)}{2} - \frac{13m_s}{1555200}.
\end{aligned}
\tag{112}
$$

With $\chi(Y) = -30$, the three equations (112) are solved by

$$
m_s = 63, \qquad \eta^2 = -3^9.
\tag{113}
$$

**Obtaining the triple intersection number**   Now that we have $\eta^2 = -3^9$, we can read off $\kappa_{111}^{(X_{\text{def}})}$ from the Yukawa coupling.

$$
\begin{aligned}
\frac{1}{(2\pi i)^3}\widehat{C}_{t_\infty t_\infty t_\infty} &= \frac{1}{(2\pi i)^3} \frac{1}{\eta^2\left(\varpi_0^{(H)}\right)^2}\left(\frac{d\phi}{dt_\infty}\right)^3 C_{\phi\phi\phi} \\
&= 2 - 1908\,e^{2\pi i t_\infty} + 491400\,e^{2\cdot 2\pi i t_\infty} + 9300120\,e^{3\cdot 2\pi i t_\infty} \\
&\quad - 12685736502\,e^{4\cdot 2\pi i t_\infty} + \dots
\end{aligned}
\tag{114}
$$

From the leading term, we read off

$$
\kappa_{111}^{(X_{\text{def}})} = 2.
\tag{115}
$$

**The smooth deformation**   We have obtained a triple intersection number, which is 2. The second Chern number is 32. The Hodge number $h^{1,1}$ is 1, and the Euler number is $\chi(\widehat{X}) - 2m_s = \chi(Y) - 2m_s = -156$. $h^{1,1}$ and $\chi$ fix the remaining Hodge number to be $h^{2,1} = 79$. By Wall's theorem [78] (See also [79] for further discussion in the string theory context), this is enough data to uniquely fix a family $X_{\text{def}}$ if we assume simply connectedness.

We do not have a reason for assuming simply connectedness ab initio. We are not aware of any quotient (so non-simply connected) manifolds with the topological data we have obtained, but that by itself does not preclude their existence. We will justify this assumption of simply connectedness in post, after proceeding with the topological string genus expansion beyond

---

[18]One could ponder on other values of $N$. Note that there is no solution for the $N = 2$ case. Given the success we describe with $N = 3$, and the fact that we can go on to enjoy nontrivial success at genera $5 - 11$, we do not make a serious attempt to find a solution with $N > 3$. Searching for one is complicated by the fact that there can be a different number $m^{(\pm l)}$ of exceptional curves in each torsion class, unlike the $N = 2$ or $N = 3$ case where $m_s = m^{(\pm 1)}$.

genus 4 on the basis of our following claim, and observing a successful reproduction of the constant terms (111) at higher genera and integer refined invariants from (97).

We predict that the MUM point $\phi = 0$ of the operator AESZ17 is an $N = 3$ MUM point corresponding to a codimension two complete intersection in a weighted projective space with 63 nodal singularities. The specific complete intersection is that of a quartic and a sextic in $\mathbb{WP}_{111223}$.

$$\text{Claim:} \qquad X_{\text{def}} \text{ is } \mathbb{WP}^5_{111223}[4,6]. \tag{116}$$

**A sanity check on monodromies**   The earlier $N = 1$ assumption led to the nonsensical $\kappa_{111} = -30/13$, on the basis of (102). But in the $N = 3$ case, one should replace the prepotential (92) by the corrected (99). This amounts to replacing $\chi/2$ by the quantity we refer to in this paper as $C_{3,1}$, which from the formula (101) due to [4] is $C_{3,1} = \frac{\chi}{2} + \frac{4m_s}{9}$. In light of this we can revisit (102) and find that a rational, but not integral, basis is in fact prescribed by the corrected prepotential. The condition $\frac{\chi}{\kappa_{111}} - 13 = 0$ from (102) is replaced by the new condition $\frac{2C_{3,1}}{\kappa_{111}} - 13 = 0$. We verify this:

$$\frac{2C_{3,1}}{\kappa_{111}} - 13 = \frac{2\left(\frac{\chi(Y)}{2} + 4\frac{m_s}{9}\right)}{\kappa_{111}^{(X_{\text{def}})}} - 13 = \frac{2\left(\frac{-30}{2} + 4\frac{63}{9}\right)}{2} - 13 = 0, \tag{117}$$

as required for the matrix in (102) to have rational entries. We will address the prospect of an integral basis in §3.6.

## 3.5   The integral symplectic basis attached to the large volume phase

About the MUM point $\varphi = 0$, a standard integral symplectic basis of solutions $\Pi^{(LV)}$ is $\Pi^{(LV)} = M^{(LV)}\rho\varpi^{(LV)}$. The matrix $M^{(LV)}$ contains the topological data given in §2.2.1.

The matrices $n_{\varphi_*}^{(LV)}$ that give the monodromy transformations of $\Pi^{(LV)}$ upon circling each singularity $\varphi_*$ of the operator AESZ17 are

$$n_{\varphi=0}^{(LV)} = \begin{pmatrix} 1 & -1 & 8 & 15 \\ 0 & 1 & -15 & -30 \\ 0 & 0 & 1 & 0 \\ 0 & 0 & 1 & 1 \end{pmatrix}, \qquad n_{\varphi=\frac{1}{27}}^{(LV)} = \begin{pmatrix} 1 & 0 & 0 & 0 \\ 0 & 1 & 0 & 0 \\ -3 & 0 & 1 & 0 \\ 0 & 0 & 0 & 1 \end{pmatrix},$$

$$n_{\varphi=\frac{i}{\sqrt{27}}}^{(LV)} = \begin{pmatrix} -8 & -3 & 9 & -9 \\ 9 & 4 & -9 & 9 \\ -9 & -3 & 10 & -9 \\ -3 & -1 & 3 & -2 \end{pmatrix}, \qquad n_{\varphi=\frac{-i}{\sqrt{27}}}^{(LV)} = \begin{pmatrix} 10 & -3 & 9 & 9 \\ 9 & -2 & 9 & 9 \\ -9 & 3 & -8 & -9 \\ 3 & -1 & 3 & 4 \end{pmatrix}, \tag{118}$$

$$n_{\varphi=\infty}^{(LV)} = \begin{pmatrix} 19 & 4 & -8 & -12 \\ -9 & -5 & 12 & -24 \\ 21 & 3 & -5 & -27 \\ 9 & 2 & -4 & -5 \end{pmatrix}.$$

Note that $n_{\frac{i}{\sqrt{27}}}^{(LV)} n_0^{(LV)} n_{\frac{-i}{\sqrt{27}}}^{(LV)} n_{\frac{1}{27}}^{(LV)} n_\infty^{(LV)} = \mathbb{I}$.

To compute the above monodromies about the conifolds, we chose contours displayed in Figure 2.

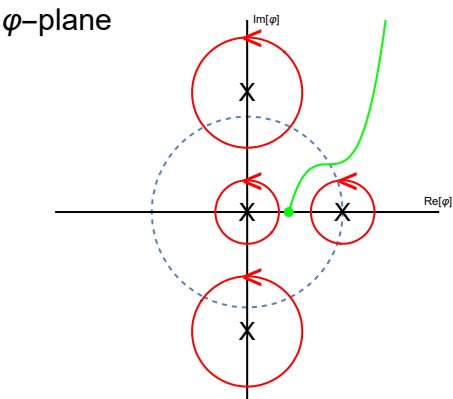

Figure 2: The dashed circle indicates the region of convergence of the Frobenius basis of solutions $\varpi^{(LV)}$ centred at $\varphi = 0$. Each X denotes a singularity in the $\varphi$-plane, and the monodromy matrices in (118) are computed for the red contours circling each of the singularities. To compute $n_\infty^{(LV)}$ we continue along the green curve through the upper-right $\varphi$-quadrant to $\varphi = i\infty$, circle this singularity counter-clockwise (not pictured) and then return along the same green curve.

## 3.6 An integral basis attached to the hybrid point

Now that we have a candidate smooth deformation $X_{\mathrm{def}} = \mathbb{WP}^5_{111223}[4, 6]$, we can proceed to study monodromies of the solution basis

$$\widetilde{\Pi} \; = \; \frac{1}{(2\pi i)^3} \mathrm{M}^{(H)} \rho \, \varpi^{(H)} \,, \tag{119}$$

where for the change of basis matrix $\mathrm{M}^{(H)}$ we choose (93) with $\kappa_{111} = 2$, $c_2 = 32$, replace $\chi$ by $2C_{3,1} = \chi(Y) + \frac{8m_s}{9} = 26$, and for now we leave $Y_{011}$ undetermined. This amounts to using the modified prepotential (99) (instead of (92)) in the manipulation (94) to obtain M. Using numerics, we go on to compute the monodromy matrices $\widetilde{n}^{(H)}_{\phi_*}$ such that circling the singularity $\phi_*$ counter-clockwise effects $\widetilde{\Pi}^{(H)} \mapsto \widetilde{n}^{(H)}_{\phi_*} \widetilde{\Pi}^{(H)}$.

Irrespective of the value of $Y_{011}$, the matrices $\widetilde{n}^{(H)}_{\phi_*}$ found in this way are non-integral and rational. To obtain an integral basis we appeal to a suggestion in [4]: since the resolution $\widehat{X}$ in that example had $\mathbb{Z}_2$ torsion, a 0-brane can decompose into two D2 branes, of central charge $-1/2$, wrapping exceptional torsion curves. The authors of [4] go on to suggest that the correct generator with 6-brane charge has central charge twice that expected from the structure sheaf of $X_{\mathrm{def}}$. The factor of 2 is specific to their example, which had a different $X_{\mathrm{def}}$ and $N = 2$. Based on this suggestion we first seek a diagonal matrix D such that $\Pi^{(H)} = \mathrm{D}\widetilde{\Pi}^{(H)}$ has integral monodromies. We find that, irrespective of $Y_{011}$, there is no such diagonal D. However, we do find a nondiagonal D that suffices:

$$\mathrm{D} \; = \; \begin{pmatrix} 3 & 0 & \frac{1}{2} & 0 \\ 0 & 1 & 0 & 0 \\ 0 & 0 & \frac{1}{3} & 0 \\ 0 & 0 & 0 & 1 \end{pmatrix}. \tag{120}$$

Monodromies of $\Pi^{(H)}$ are integral provided $Y_{011} \in \mathbb{Z}$, so we now set $Y_{011} = 0$.

The diagonal entries of (120) may be explained by the primitive-charge arguments of [4]. However, we do not have a physical explanation for the off-diagonal entry $1/2$.

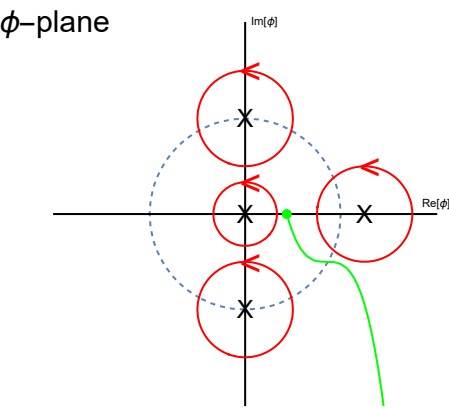

Figure 3: Here the dashed circle contains the region of convergence for the Frobenius solutions $\varpi^{(H)}$ expanded about $\phi = 0$. Each X is a singularity in the $\phi$-plane, which we encircle with the red contours to compute the monodromy matrices (121). To compute $n_\infty^{(H)}$ we continue along the green curve in the lower-right $\phi$-quadrant to $\phi = -i\infty$, which we then encircle counter-clockwise (not pictured) before returning along the green curve.

With $\Pi^{(H)} = D\widetilde{\Pi}^{(H)}$, the monodromy matrices $n_{\phi_*}^{(H)} = D.\widetilde{n}^{(H)}.D^{-1}$ such that $\Pi^{(H)} \mapsto n_{\phi_*}^{(H)}\Pi^{(H)}$ upon circling $\phi_*$ counter-clockwise are as follows:

$$
n_{\phi=0}^{(H)} = \begin{pmatrix} 1 & -3 & 27 & 3 \\ 0 & 1 & -3 & -2 \\ 0 & 0 & 1 & 0 \\ 0 & 0 & 3 & 1 \end{pmatrix}, \qquad
n_{\phi=\frac{1}{27}}^{(H)} = \begin{pmatrix} 1 & -66 & 363 & 0 \\ 0 & 1 & 0 & 0 \\ 0 & 0 & 1 & 0 \\ 0 & -12 & 66 & 1 \end{pmatrix},
$$

$$
n_{\phi=\frac{-i}{81\sqrt{3}}}^{(H)} = \begin{pmatrix} -2 & 0 & 9 & 12 \\ -4 & 1 & 12 & 16 \\ -1 & 0 & 4 & 4 \\ 0 & 0 & 0 & 1 \end{pmatrix}, \qquad
n_{\phi=\frac{i}{81\sqrt{3}}}^{(H)} = \begin{pmatrix} 1 & 0 & 0 & 0 \\ -4 & 1 & 0 & 16 \\ -1 & 0 & 1 & 4 \\ 0 & 0 & 0 & 1 \end{pmatrix}, \tag{121}
$$

$$
n_{\phi=\infty}^{(H)} = \begin{pmatrix} -29 & -318 & 1734 & 132 \\ -19 & -161 & 882 & 90 \\ -4 & -33 & 181 & 19 \\ -3 & -42 & 228 & 13 \end{pmatrix}.
$$

To compute the above monodromies we chose the integration contours displayed in Figure 3.

We verify that $n_{\phi=\frac{-i}{81\sqrt{3}}}^{(H)} n_{\phi=\infty}^{(H)} n_{\phi=\frac{1}{27}}^{(H)} n_{\phi=\frac{i}{81\sqrt{3}}}^{(H)} n_{\phi=0}^{(H)} = \mathbb{I}$.

Let us remark that we could also have chosen the $(3,3)$-entry in (120) to be 1 or $-1/3$ (instead of the value of $1/3$ that we use), if we were only concerned with obtaining integral monodromies. However, note the following justification for choosing $1/3$.

We find a transfer matrix $T_{\text{upper}}$, so that

$$
\Pi^{(H)}(\phi) = \frac{1}{i3^{9/2}} T_{\text{upper}} \Pi^{(LV)}(\varphi(\phi)), \tag{122}
$$

where we analytically continue along an integration contour in the upper-right quadrant in $\varphi$-space. This path is obtained by concatenating the green curves displayed in both Figure 2

and Figure 3. With our choice of D as in (120), we get

$$
T_{\text{upper}} = \begin{pmatrix} -18 & -5 & 11 & 3 \\ -5 & 0 & 0 & 11 \\ -1 & 0 & 0 & 2 \\ -3 & -1 & 2 & 0 \end{pmatrix},
\tag{123}
$$

with determinant equalling 1. This is in agreement with the value we found for $\eta$ in (85), namely $\eta^2 = -3^9$. So our results (113) and (122) are consistent, see our discussion surrounding (91).

With the exception of $\phi = \frac{i}{81\sqrt{3}}$, for every singularity we can verify that $T_{\text{upper}}^{-1} n_{\phi_*}^{(H)} T_{\text{upper}} = n_{\varphi_*}^{(LV)}$. The reason that this fails for $\phi = \frac{i}{81\sqrt{3}}$ has to do with our choices of integration contour. If we instead continue along the lower-right quadrant in $\varphi$-space, we obtain

$$
T_{\text{lower}} = \begin{pmatrix} 15 & -5 & 11 & 3 \\ -5 & 0 & 0 & 11 \\ -1 & 0 & 0 & 2 \\ 3 & -1 & 2 & 0 \end{pmatrix}.
\tag{124}
$$

We verify that $T_{\text{lower}}^{-1} n_{\phi = \frac{i}{81\sqrt{3}}}^{(H)} T_{\text{lower}} = n_{\varphi = \frac{-i}{\sqrt{27}}}^{(LV)}$.

# 4 Outlook

One obvious direction for further research is to obtain a better understanding of the physics of the GLSM in the $\zeta \ll 0$-phase. The phase suffers from two independent challenging issues: the existence of a Coulomb branch at infinity that makes the GLSM non-regular and the fact that, having a continuous non-Abelian unbroken symmetry and a cubic superpotential, the phase is a strongly coupled interacting gauge theory. As there could be potential effects that relate these two phenomena in our model, it may not be a good example for an in-depth study of these effects. A more practicable approach seems to be to first study simpler models where only one of these effects occurs. A source of such examples are the GLSMs studied by Hori and Tong [9], GLSM realisations of the models studied by Hosono and Takagi, in particular in low dimensions, or the non-compact example recently discussed in [40].

The AESZ list of Calabi-Yau operators [27] currently contains many examples of MUM points without a satisfactory geometric understanding. It is already anticipated that further application of the torsion-refinement of [2, 4] will allow for a better understanding of these examples. This approach has been undertaken for certain AESZ operators in [80]. In addition to this, we have explained in this article how to construct nonAbelian GLSMs that describe CICY quotients [21]. Such a construction may prove useful in the analysis of those AESZ operators, in particular for the examples described in [31].

Furthermore, D-brane transport and the non-Abelian duality need to be understood for non-regular GLSMs. This would, for instance, make it possible to determine the integral bases and monodromy matrices computed in §3.5 and §3.6 directly in the GLSM and to establish connections with the underlying mathematical structures such as Seidel-Thomas twists. While an exhaustive discussion of D-branes in the $\zeta \ll 0$-phase or the associated categorical equivalences for this GLSM is left to future work, we would like to point out some peculiarities which are specific to this model and appear to be connected to the Coulomb branch at $\zeta \to -\infty$.

The first comment concerns the evaluation of the hemisphere partition function in the $\zeta \ll 0$-phase. In strongly coupled phases the hemisphere partition function will in general be divergent. To achieve absolute convergence one has to make a suitable choice for an integration contour. On top of this, one also needs to apply a grade-restriction rule associated to the unbroken continuous gauge symmetry in this phase [63, 81]. In examples like the Rødland and Hosono-Takagi models it is possible, provided one has a suitable brane, to make a naïve choice of contour, analogous to calculations for the sphere partition function [82]: take the contour for real $\sigma_i$ and close it in the negative complex plane. The result will not be convergent. However, the divergent sums can be regulated to yield an expansion in terms of solutions of the Picard-Fuchs equation associated to the phase. For the present example, the situation is more complicated. Due to the Coulomb branch at any theta angle, one has to be more careful with the choice of integration contour. One must make sure that the contour does not intersect the extra Coulomb branch. We can naïvely evaluate the hemisphere partition function in the strongly coupled phase by closing the contour at infinity. Due to the fact that our gauge group has rank 3, we obtain a triple sum, where two sums are divergent. We have not been able to regulate the expression. To understand this better, it also seems useful to first study a simpler non-regular model.

The existence of the Coulomb branch at infinity also raises questions about the formulation of equivalences of D-brane categories associated to the two phases. In the non-Calabi-Yau case, the grade restriction rule determines two windows, a big one and a small one [19, 52, 83, 84]. The small window accounts for equivalences of branes localised on the Higgs branches in the respective phases, while the large window also captures branes localised on the Coulomb branch vacua. In the mathematical formulation, the existence of the massive Coulomb vacua is reflected in the necessity to supplement the categorical equivalences between the geometries given by the Higgs vacua by additional exceptional collections. The situation in our example is similar but the additional Coulomb vacua are massless. The question is if the presence of the Coulomb branch modifies the categorical equivalences between the two phases. Evidence comes from the mathematics literature, where similar phenomena were observed for manifolds of complex dimension two. In [85] it was observed that the homological projective dual of the K3 given by a codimension 6 complete intersection of linear equations in $G(2, 6)$ is a Pfaffian cubic in $\mathbb{P}^5$, i.e. the dimensions of the allegedly dual theories do not match and exceptional collections have to be added. A similar phenomenon occurs for an Enriques surface considered by Hosono and Takagi [86, 87]. In this case the homological projective dual has the correct dimension, but suffers from nodal singularities. To obtain an equivalence of categories, one again has to add exceptional collections. The GLSMs associated to these two examples are both non-regular. We suspect that the necessity for adding exceptional collections is related to the Coulomb branch at infinity. On the other hand, a non-Abelian non-regular GLSM associated to a non-compact Calabi-Yau has recently been considered in [40]. The authors were able to find grade restrictions rule and compute monodromies despite the existence of the Coulomb branch at infinity.

A final remark in relation to D-branes concerns the Coulomb branches at the phase boundary. To our knowledge, in all the Calabi-Yau GLSMs that have been studied so far, it has been observed that the D-brane that becomes massless at the singular point closest to the phase is the structure sheaf of the Calabi-Yau in the respective phase. Considering the $\zeta \ll 0$-phase of our model, there are two singular points at the same distance to the phase. It would be interesting to study what this means for D-branes becoming massless in this case.

## Acknowledgments

We thank Thorsten Schimannek for helpful advice and for sharing preliminary results of his work [80] which also contains examples with $\mathbb{Z}_3$-torsion. We further thank Konstantin Aleshkin, Philip Candelas, Will Donovan, Mohamed Elmi, Kentaro Hori, Shinobu Hosono, Pyry Kuusela, Xenia de la Ossa, Mauricio Romo, Emanuel Scheidegger, Eric Sharpe, and Hiromichi Takagi for helpful discussions and comments. We thank BICMR, MATRIX Institute, and the University of Queensland for hospitality. Furthermore, we thank Mainz Institute for Theoretical Physics (MITP) of the Cluster of Excellence PRISMA$^+$ (Project ID 390831469) for hospitality and partial support. JK thanks Gakushuin University, Kavli IPMU, and the University of Vienna for hospitality. We thank the anonymous SciPost referees for their careful reading of our paper, and their insightful comments.

**Funding information**  JK is supported by the Australian Research Council Future Fellowship FT210100514. JM is supported by a University of Melbourne Establishment Grant.

## A    Computing the topological string free energies

Higher genus topological string free energies for compact CY3 are computed and discussed in [41, 43, 44, 88]. We use the polynomial approach to solving the topological string [42, 76], see also [4]. The holomorphic three-form $\Omega$ on $\lambda$ provides the Kähler potential $\mathcal{K}$ for the metric on the moduli space of complex structures $\mathcal{M}$ of $\lambda$, and also the Yukawa coupling $C_{\varphi\varphi\varphi}$, through

$$\mathcal{K}(\varphi, \overline{\varphi}) = -\log\left( i \int_{\lambda} \Omega \wedge \overline{\Omega} \right), \qquad C_{\varphi\varphi\varphi}(\varphi) = -\int_{\lambda} \Omega \wedge \partial_{\varphi}^3 \Omega. \tag{A.1}$$

For our particular example and choice of gauge, $C_{\varphi\varphi\varphi} = \frac{30(1-\frac{9}{5}\varphi)}{\varphi^3(1-27\varphi)(1+27\varphi^2)}$. The metric on the moduli space is $\mathcal{G}_{\varphi\overline{\varphi}} = \partial_{\varphi}\partial_{\overline{\varphi}}\mathcal{K}$, which defines the Levi-Civita connection $\mathcal{C}$ on $\mathcal{M}$. The holomorphic 3-form $\Omega$ is a section of the Kähler line bundle $\mathcal{L}^{-1}$ over $\mathcal{M}$.

The B-model free energies $\mathcal{F}^{(g)}(\varphi, \overline{\varphi})$ are polynomials in certain propagator functions $\mathcal{S}^{\varphi\varphi}, \mathcal{S}^{\varphi}, \mathcal{S}$, as well as the Kähler potential $\mathcal{K}$. These propagators are respectively sections of $\mathcal{L}^2 \otimes \mathrm{Sym}^2(T\mathcal{M})$, $\mathcal{L}^2 \otimes T\mathcal{M}$, and $\mathcal{L}^2$. They are defined as anholomorphic potentials,[19]

$$\begin{aligned}
\partial_{\overline{\varphi}} \mathcal{S}^{\varphi\varphi} &= \mathcal{G}^{\varphi\overline{\varphi}} \mathcal{G}^{\varphi\overline{\varphi}} \overline{C_{\varphi\varphi\varphi}}, \\
\partial_{\overline{\varphi}} \mathcal{S}^{\varphi} &= -\mathcal{K}_{\varphi} \mathcal{G}^{\varphi\overline{\varphi}} \mathcal{G}^{\varphi\overline{\varphi}} \overline{C_{\varphi\varphi\varphi}}, \\
\partial_{\overline{\varphi}} \mathcal{S} &= \frac{1}{2} \mathcal{K}_{\varphi} \mathcal{K}_{\varphi} \mathcal{G}^{\varphi\overline{\varphi}} \mathcal{G}^{\varphi\overline{\varphi}} \overline{C_{\varphi\varphi\varphi}},
\end{aligned} \tag{A.2}$$

where $\mathcal{K}_{\varphi} = \partial_{\varphi}\mathcal{K}$. The integrated special geometry relation is used in practice to compute $\mathcal{S}^{\varphi\varphi}$. This reads[20]

$$C_{\varphi\varphi\varphi}\mathcal{S}^{\varphi\varphi} = -\mathcal{C}_{\varphi\varphi}^{\varphi} + 2\delta_{\varphi}^{\varphi}\mathcal{K}_{\varphi} + s_{\varphi\varphi}^{\varphi}. \tag{A.3}$$

Here $s_{\varphi\varphi}^{\varphi}$ is a propagator ambiguity, a holomorphic function of $\varphi$ that we can freely choose. $s_{\varphi\varphi}^{\varphi}$ is not a tensor, and under a coordinate transformation has the same transformation law as the Christoffel symbol $\mathcal{C}_{\varphi\varphi}^{\varphi}$ so that the LHS of (A.3) is tensorial.

---

[19]Note that these relations are not [76, equation (4.27)]. The equations (A.2) define the tilded propagators in (4.29) of [76], which we display here without tildes.

[20]We are working in the one-parameter setting. In (A.3), one should read $2\delta_{\varphi}^{\varphi}\mathcal{K}_{\varphi}$ as $\delta_i^l \mathcal{K}_j + \delta_j^l \mathcal{K}_i$ with $l = i = j = \varphi$. We shall throughout this section write $\delta_{\varphi}^{\varphi}$ instead of 1 where appropriate, so that there is no abuse of indices.

The BCOV ring $\mathbb{Q}(\varphi)[\mathcal{S}^{\varphi\varphi}, \mathcal{S}^{\varphi}, \mathcal{S}]$ is closed under differentiation. To see this, one takes a covariant derivative of any of (A.2) using the Kähler and Levi-Civita connections. The special geometry relation $[\partial_{\overline{\varphi}}, D_{\varphi}]^{\varphi}{}_{\varphi} = \partial_{\overline{\varphi}} \mathfrak{r}^{\varphi}_{\varphi\varphi} = 2\delta^{\varphi}_{\varphi}\mathcal{G}_{\varphi\overline{\varphi}} - C_{\varphi\varphi\varphi}\overline{C}^{\varphi\varphi}_{\overline{\varphi}}$ allows one to change the order of differentiation, and then everything can be collected under a $\partial_{\overline{\varphi}}$ that is removed at the expense of adding a holomorphic 'constant of integration' tensor $h^{\bullet}_{\bullet}$ with the appropriate legs. This leads to

$$
\begin{aligned}
\partial_{\varphi}\mathcal{S}^{\varphi\varphi} &= C_{\varphi\varphi\varphi}\mathcal{S}^{\varphi\varphi}\mathcal{S}^{\varphi\varphi} + 2\delta^{\varphi}_{\varphi}\mathcal{S}^{\varphi} - 2s^{\varphi}_{\varphi\varphi}\mathcal{S}^{\varphi\varphi} + h^{\varphi\varphi}_{\varphi}, \\
\partial_{\varphi}\mathcal{S}^{\varphi} &= C_{\varphi\varphi\varphi}\mathcal{S}^{\varphi\varphi}\mathcal{S}^{\varphi} + 2\delta^{\varphi}_{\varphi}\mathcal{S} - s^{\varphi}_{\varphi\varphi}\mathcal{S}^{\varphi} - h_{\varphi\varphi}\mathcal{S}^{\varphi\varphi} + h^{\varphi}_{\varphi}, \\
\partial_{\varphi}\mathcal{S} &= \frac{1}{2}C_{\varphi\varphi\varphi}\mathcal{S}^{\varphi}\mathcal{S}^{\varphi} - h_{\varphi\varphi}\mathcal{S}^{\varphi} + h_{\varphi}, \\
\partial_{\varphi}\mathcal{K}_{\varphi} &= \mathcal{K}_{\varphi}\mathcal{K}_{\varphi} - C_{\varphi\varphi\varphi}\mathcal{S}^{\varphi\varphi}\mathcal{K}_{\varphi} + s^{\varphi}_{\varphi\varphi}\mathcal{K}_{\varphi} - C_{\varphi\varphi\varphi}\mathcal{S}^{\varphi} + h_{\varphi\varphi}.
\end{aligned}
\tag{A.4}
$$

There is a freedom in choosing the propagator ambiguities $h^{\bullet}_{\bullet}, s^{\varphi}_{\varphi\varphi}$. We make a choice

$$
s^{\varphi}_{\varphi\varphi} = 0, \qquad h^{\varphi\varphi}_{\varphi} = 0, \qquad h^{\varphi}_{\varphi} = 0. \tag{A.5}
$$

The $h^{\bullet}_{\bullet}$ are tensors, and so choosing any of them to be zero in a patch leads to them being zero in every patch. However, $s^{\varphi}_{\varphi\varphi}$ has the same transformation law as a Christoffel symbol and so our choice $s^{\varphi}_{\varphi\varphi} = 0$ gives $s^{\phi}_{\phi\phi} = -2/\phi$.

The BCOV closure relations (A.4) and integrated special geometry relation (A.3) then provide the remaining propagator ambiguities, in addition to the propagators themselves. First one reads off $\mathcal{S}^{\varphi\varphi}$ from (A.3) - to do this, one must divide by $C_{\varphi\varphi\varphi}$. One then reads off $h_{\varphi\varphi}$ from the fourth equation in (A.4), then reads off $\mathcal{S}^{\varphi}$ from the first equation in (A.4), then reads off $\mathcal{S}$ from the second equation in (A.4), and finally reads off $h_i$ from the third equation in (A.4). Before we actually do this, we will pass to the holomorphic limit. This will need to be done anyway when we read off GV invariants from the A-model $F^{(g)}$, so it is convenient computationally to take this step now while we are computing propagators and propagator ambiguities. This is effected by the replacements

$$
\mathcal{K}_{\varphi} \mapsto K_{\varphi} = -\partial_{\varphi}\log\left(\varpi^{(LV)}_0(\varphi)\right), \quad \mathfrak{r}^{\varphi}_{\varphi\varphi} \mapsto \Gamma^{\varphi}_{\varphi\varphi} = \frac{\mathrm{d}\varphi}{\mathrm{d}t_0}\frac{\mathrm{d}^2 t_0}{\mathrm{d}\varphi^2}, \quad \mathcal{S}^{\bullet}(\varphi,\overline{\varphi}) \mapsto S^{\bullet}(\varphi). \tag{A.6}
$$

Recall that the mirror map $t_0$ about $\varphi = 0$ was given in (84). For our example, with our choices (A.5) made, we obtain

$$
\begin{aligned}
h_{\varphi\varphi} &= \frac{5 - 360\varphi + 822\varphi^2 - 28728\varphi^3 + 32805\varphi^4}{2\varphi^2(5-9\varphi)(1-27\varphi)(1+27\varphi^2)}, \\
h_{\varphi} &= \frac{\left(\begin{smallmatrix} 125 - 50925\varphi + 1718520\varphi^2 - 25570080\varphi^3 + 623720970\varphi^4 - 2457520722\varphi^5 \\ + 19754488656\varphi^6 - 55770861960\varphi^7 + 69555529521\varphi^8 - 31381059609\varphi^9 \end{smallmatrix}\right)}{48\varphi(5-9\varphi)^4(1-27\varphi)(1+27\varphi^2)}, \\
S^{\varphi\varphi} &= \frac{\varphi^2}{30} - \frac{31\varphi^3}{25} + \frac{217\varphi^4}{250} - \frac{16086\varphi^5}{625} + \frac{1287726\varphi^6}{3125} + \frac{145352034\varphi^7}{15625} + \dots, \\
S^{\varphi} &= \frac{\varphi}{60} - \frac{26\varphi^2}{25} - \frac{919\varphi^3}{500} - \frac{58767\varphi^4}{625} - \frac{222633\varphi^5}{625} - \frac{47789343\varphi^6}{15625} + \dots, \\
S &= \frac{1}{120} - \frac{109\varphi}{100} - \frac{1947\varphi^2}{1000} - \frac{112029\varphi^3}{625} - \frac{661347\varphi^4}{1250} - \frac{18030249\varphi^5}{15625} + \dots
\end{aligned}
\tag{A.7}
$$

The exact form of the polynomial expression for $\mathcal{F}^{(g)}$ is obtained by recursively solving the holomorphic anomaly equations [45, 46]. These were recast into the following PDE form

in [42, 89] (one can also see this manipulation explained in [90]).

$$\frac{\partial \mathcal{F}^{(g)}}{\partial \mathcal{S}^{\varphi\varphi}} = \frac{1}{2}\partial_\varphi\left(\partial'_\varphi\mathcal{F}^{(g-1)}\right) + \frac{1}{2}(C_{\varphi\varphi\varphi}\mathcal{S}^{\varphi\varphi} - s^\varphi_{\varphi\varphi})\left(\partial'_\varphi\mathcal{F}^{(g-1)}\right) + \frac{1}{2}(C_{\varphi\varphi\varphi}\mathcal{S}^\varphi - h_{\varphi\varphi})c_{g-1}$$
$$+ \frac{1}{2}\sum_{h=1}^{g-1}\left(\partial'_\varphi\mathcal{F}^{(h)}\right)\left(\partial'_\varphi\mathcal{F}^{(g-h)}\right),$$

$$\frac{\partial \mathcal{F}^{(g)}}{\partial \mathcal{S}^\varphi} = (2g-3)\left(\partial'_\varphi\mathcal{F}^{(g-1)}\right) + \sum_{h=1}^{g-1}c_h\left(\partial'_\varphi\mathcal{F}^{(g-h)}\right), \tag{A.8}$$

$$\frac{\partial \mathcal{F}^{(g)}}{\partial \mathcal{S}} = (2g-3)c_{g-1} + \sum_{h=1}^{g-1}c_h c_{g-h},$$

$$c_h = \begin{cases} \frac{\chi(Y)}{24}-1, & h=1, \\ (2h-2)\mathcal{F}^{(h)}, & h>1, \end{cases} \qquad \partial'_\varphi\mathcal{F}^{(h)} = \begin{cases} \frac{1}{2}C_{\varphi\varphi\varphi}\mathcal{S}^{\varphi\varphi} + \partial_\varphi f^{(1)}, & h=1, \\ \partial_\varphi\mathcal{F}^{(h)}, & h>1. \end{cases} \tag{A.9}$$

For genera $g \geq 2$, the equations (A.8) recursively ensure that $\mathcal{F}^{(g)}$ is in the BCOV ring, and can be written as a polynomial in the propagators with rational coefficients. Any instances of $\partial_\varphi\mathcal{F}^{(h<g)}$ on the LHS of (A.8) should be expanded by the Leibniz rule, and then any derivatives of propagators replaced using the closure relations (A.4). Working in the holomorphic limit this provides at each genus

$$\mathcal{F}^{(g)}(\varphi) = P^{(g)}(S^{\varphi\varphi}, S^\varphi, S) + f^{(g)}(\varphi), \tag{A.10}$$

where $P^{(g)}$ is a polynomial with no degree-0 monomials and $f^{(g)}(\varphi)$ is a rational function of $\varphi$, which the HAE do not fix. $f^{(g)}$ is known as the holomorphic ambiguity, and it must be determined by incorporating additional data.

Assign degrees $1, 2, 3$ respectively to $S^{\varphi\varphi}, S^\varphi, S$. As a consequence of (A.8), the highest order monomials in the polynomials $P^{(g)}(S^{\varphi\varphi}, S^\varphi, S)$ are of degree $3g-3$.

$\mathcal{F}^{(g)}(\varphi)$ can only be singular when the B-model geometry $\lambda_\varphi$ becomes singular. Depending on the type of singularity that $\lambda_\varphi$ acquires at $\varphi = \varphi_{\text{sing}}$, there are known behaviours of $\mathcal{F}^{(g)}$. This is how we can constrain $f^{(g)}(\varphi)$, and for certain low genera these constraints are strong enough to completely fix $f^{(g)}(\varphi)$. In our example, in the space of $\varphi$ over which $\lambda_\varphi$ is fibred there are two MUM points $\varphi \in \{0, \infty\}$ and three conifold points $\varphi \in \{\pm i/(3\sqrt{3}), 1/27\}$. To be precise, $1/27$ is a hyperconifold point because there is an $S^3/\mathbb{Z}_3$ on $\lambda$ that shrinks, instead of an $S^3$.

In an expansion about the $N = 1$ MUM point $\varphi = 0$, one has the constant term due to [64, 74, 75]:

$$F^{(g)}(t_0)|_{\text{constant}} = \left(\varpi_0^{(LV)}(\varphi(t_0))\right)^{2g-2}\mathcal{F}^{(g)}(\varphi(t_0))|_{\text{constant}}$$
$$= \frac{(-1)^{g-1}B_{2g}B_{2g-2}}{2g(2g-2)(2g-2)!}\frac{\chi(Y)}{2}. \tag{A.11}$$

Given the difficulties we have experienced with the other MUM point $\phi = \frac{1}{729\varphi} = 0$, we shall initially only impose (A.11) at $\varphi = 0$. The only constraint we place on the behaviour of $F^{(g)}(\varphi)$ at $\varphi = \infty$ is that $F^{(g)}(t_\infty)$ is nonsingular. We justify this assumption by the argument in [41], that $F^{(g)}$ can only be singular at a point where new massless charged states emerge in the effective 4d spacetime theory. As no D-branes become massless at $\varphi = \infty$, as can be seen from the logarithmic structure of the periods $\varpi^{(H)}$, we therefore expect regular behaviour.

The ansatz for the holomorphic ambiguity is then

$$
\begin{aligned}
f^{(g)}(\varphi) = b_0 + \sum_{k=1}^{2g-2} b_k \varphi^k + \sum_{k=1}^{3g-3} \frac{b_k^{(\text{App})}}{(5-9\varphi)^k} + \sum_{k=1}^{2g-2} \frac{b_k^{(1/27)}}{(1-27\varphi)^k} \\
+ \sum_{k=1}^{2g-2} \frac{b_{k,0}^{(1/\sqrt{-27})} + b_{k,1}^{(1/\sqrt{-27})}\varphi}{(1+27\varphi^2)^k} .
\end{aligned}
\tag{A.12}
$$

The number $b_0$ is fixed by (A.11). We will shortly turn to explaining how to fix the numbers $b_k^{(1/27)}$ and $b_{k,a}^{(1/\sqrt{-27})}$ using the conifold gap condition (A.16). We will subsequently explain how we have used regularity of the free energy at the apparent singularity $\varphi = 5/9$ to fix the numbers $b_k^{(\text{App})}$.

The only numbers not determined by a regularity requirement are the $2g-2$ numbers $b_k$. Note however that the highest power of $\varphi$ in (A.12) being $2g-2$ is a consequence of regularity of $F^{(g)}(t_\infty)$. To fix these remaining $b_k$, we use the Castelnuovo criterion as studied in [41], see also [91]. We aim to use the Gopakumar-Vafa formula (96) to read off GV invariants $n_\beta^{(g)}$ from $F^{(g)}(t)$. The Castelnuovo criterion is that these numbers vanish unless $g \leq g_{\text{Castelnuovo}}(\beta)$, where

$$
g_{\text{Castelnuovo}}(\beta) = \begin{cases} \left\lfloor \frac{\beta}{3} + \frac{2\beta^2}{3\kappa_{111}^{(LV)}} \right\rfloor + 1, & \beta < \kappa_{111}^{(LV)}, \\[2ex] \left\lfloor \frac{\beta}{2} + \frac{\beta^2}{2\kappa_{111}^{(LV)}} \right\rfloor + 1, & \beta \geq \kappa_{111}^{(LV)}. \end{cases}
\tag{A.13}
$$

The second case in (A.13) is the bound identified in [41], while the upper line is a more recent discovery proven for simply connected 1-parameter CY3 in [91]. There is an additional assumption in [91], that what they called the BMT inequality holds. $Y$ is not simply connected, but we assume that the upper line in (A.13) holds and use it as boundary data. If we do not make this assumption and only use the bottom line of (A.13), then we are unable to reach even genus 2. We will return to discussing this assumption at the end of this appendix.

At any fixed genus $g$, (A.13) predicts that some number of the first few $n_\beta^{(g)}$ vanish. For our model, at genera 2,3,4 there are respectively 2,4,6 such vanishing $n_\beta^{(g)}$. Note that this equals $2g-2$ for each case, and so we can fully constrain the holomorphic ambiguity. Below we provide expansions for the $\mathcal{F}^{(g)}(\varphi)$ that we so obtain.

$$
\begin{aligned}
\mathcal{F}^{(2)}(\varphi) = -\frac{1}{192} + \frac{29\varphi}{160} + \frac{297\varphi^2}{320} + \frac{441\varphi^3}{40} + \frac{6633\varphi^4}{40} \\
+ \frac{186957\varphi^5}{40} + \frac{1380831\varphi^6}{8} + \dots ,
\end{aligned}
$$

$$
\begin{aligned}
\mathcal{F}^{(3)}(\varphi) = \frac{1}{48384} + \frac{23\varphi}{4032} + \frac{83\varphi^2}{2688} + \frac{193\varphi^3}{1344} + \frac{437\varphi^4}{256} \\
- \frac{3333\varphi^5}{28} - \frac{233557\varphi^6}{28} + \dots ,
\end{aligned}
\tag{A.14}
$$

$$
\begin{aligned}
\mathcal{F}^{(4)}(\varphi) = -\frac{1}{2903040} + \frac{173\varphi}{806400} + \frac{2599\varphi^2}{537600} + \frac{223\varphi^3}{26880} + \frac{2353\varphi^4}{35840} \\
+ \frac{475457\varphi^5}{89600} + \frac{147354793\varphi^6}{537600} + \dots
\end{aligned}
$$

Note that, by construction, the constant terms in these expansions match with (A.11). By applying the mirror map (85) and comparing with the GV formula (96), we obtain the invariants up to genus 4 listed in Table 2, Appendix §B. The tables in Appendix §B go beyond genus 4, using considerations we will go on to explain in this current appendix.

**Expanding about the three conifold points**   The behaviour of $\mathcal{F}^{(g)}$ at a conifold point $\varphi_c$ is governed by the gap condition derived in [41]. The mirror coordinate at a conifold point is given by

$$t_c = k_c \frac{\varpi_1^{(c)}}{\varpi_0^{(c)}}. \tag{A.15}$$

Here $\varpi_0^{(c)}$ is a solution to the PF equation, with a series expansion centred on the conifold point beginning $\varpi_0^{(c)}(\varphi - \varphi_c) = 1 + O((\varphi - \varphi_c)^2)$. Similarly, $\varpi_1^{(c)} = (\varphi - \varphi_c) + O((\varphi - \varphi_c)^2)$. Here $k_c$ is a normalisation constant, such that [41]

$$F_{\text{conifold}}^{(g \geq 2)}(t_c) = \frac{|G_c|(-1)^{g-1}B_{2g}}{2g(2g-2)t_c^{2g-2}} + O(1). \tag{A.16}$$

The condition (A.16) constrains $f^{(g)}$ by prescribing the behaviour at $\varphi = \varphi_c$. To obtain the numbers $k_c$ at each conifold point, we utilise the observation[21] in [67, §8.7]. This provides

$$k_{1/27} = 81, \qquad k_{\pm i/(3\sqrt{3})} = \frac{9}{2}(-1 \pm i\sqrt{3}). \tag{A.17}$$

In practice, we go back and recompute all propagators in expansions about each conifold point in the variable $\widetilde{\varphi} = \varphi - \varphi_c$. In doing so, when we take the holomorphic limit we replace $\mathcal{K}_{\widetilde{\varphi}} \mapsto -\partial_{\widetilde{\varphi}} \log(\varpi_0^{(c)})$, $\nabla_{\widetilde{\varphi}\widetilde{\varphi}}^{\widetilde{\varphi}} \mapsto \frac{\mathrm{d}\widetilde{\varphi}}{\mathrm{d}t_c}\frac{\mathrm{d}^2 t_c}{\mathrm{d}\widetilde{\varphi}^2}$. As a consistency check, it is useful to note that the $h_{\widetilde{\varphi}\widetilde{\varphi}}, h_{\widetilde{\varphi}}$ so obtained equal the old $h_{\varphi\varphi}, h_{\varphi}$ with replacements $\varphi \mapsto \varphi_c + \widetilde{\varphi}$. When we convert between A-model and B-model free energy expansions about $\varphi_c$, we take $F^{(g)} = (\varpi_0^{(c)})^{2g-2}\mathcal{F}^{(g)}$. Since two of our conifold points are valued in $\mathbb{Q}(1/\sqrt{-27})$ it is convenient computationally to use the methods developed in [43] for implementing the gap condition at conifold points that are roots of a polynomial irreducible over $\mathbb{Q}$, so that we can treat both roots of $1 + 27\varphi^2$ simultaneously and do not need to perform two separate expansions.

**Expanding about the apparent singularity**   Our model has an apparent singularity at $\varphi = 5/9$, where $\lambda_\varphi$ is smooth. However, the propagators $S^{\varphi\varphi}, S^\varphi, S$ that we have constructed have singularities at $\varphi = 5/9$. So too then does $P^{(g)}(S^{\varphi\varphi}, S^\varphi, S)$, but $\mathcal{F}^{(g)}$ must be regular. So, we must include terms in $f^{(g)}(\varphi)$ such that $P^{(g)} + f^{(g)}$ is regular at $\varphi = 5/9$. We use a mirror coordinate

$$t_{\text{App}} = \frac{\varpi_1^{(\text{App})}}{\varpi_0^{(\text{App})}}, \tag{A.18}$$

where $\varphi = 5/9 + \widetilde{\varphi}$ and

$$\begin{aligned}
\varpi_0^{(\text{App})} &= 1 - \frac{81}{70}\widetilde{\varphi}^2 + \frac{33835077}{2450000}\widetilde{\varphi}^5 - \frac{2647934307}{34300000}\widetilde{\varphi}^6 + \frac{4193246637}{15006250}\widetilde{\varphi}^7 + \dots, \\
\varpi_1^{(\text{App})} &= \widetilde{\varphi} - \frac{12}{5}\widetilde{\varphi}^2 + \frac{28203}{9500}\widetilde{\varphi}^3 - \frac{22757841}{1750000}\widetilde{\varphi}^5 + \frac{117665196561}{2327500000}\widetilde{\varphi}^6 + \dots
\end{aligned} \tag{A.19}$$

---

[21]JM thanks Mohamed Elmi and Emanuel Scheidegger for collaboration on the upcoming work [92] which further discusses this normalisation.

Let us explain the above expansions. Note that, as displayed in the Riemann symbol Table 1, the indices of the Picard-Fuchs operator at $\varphi = 5/9$ are $(0, 1, 3, 4)$. This means that there exists a basis of series solutions, starting $1 + \ldots$, $\widetilde{\varphi} + \ldots$, $\widetilde{\varphi}^3 + \ldots$, $\widetilde{\varphi}^4 + \ldots$ The solution $\varpi_0$ is characterised by being the unique solution with leading term 1 and no terms of degree 1, 3, or 4. We then seek a solution $\varpi_1^{(\text{App})}$ such that when we recompute the propagators as expansions in $\widetilde{\varphi}$ the ensuing propagator ambiguities $h_{\widetilde{\varphi}\widetilde{\varphi}}, h_{\widetilde{\varphi}}$ read off from (A.4) equal our previous expressions $h_{\varphi\varphi}, h_{\varphi}$ after a tensor transformation (merely substituting $\varphi = 5/9 + \widetilde{\varphi}$, as the Jacobian is 1). This recomputation involves taking the holomorphic limit with $\mathcal{K}_{\widetilde{\varphi}} \mapsto -\partial_{\widetilde{\varphi}} \log(\varpi_0^{(\text{App})})$ and $\Upsilon_{\widetilde{\varphi}\widetilde{\varphi}}^{\widetilde{\varphi}} \mapsto \frac{d\widetilde{\varphi}}{t_{\text{App}}} \frac{d^2 t_{\text{App}}}{d\widetilde{\varphi}^2}$. Our tensorial requirement on the propagator ambiguities is satisfied by the choice of $\varpi_1^{(\text{App})}$ in (A.19).

Having recomputed the propagators as expansions about $\widetilde{\varphi} = 0$, we choose $b_k^{(\text{App})}$ in (A.12) so that $\mathcal{F}^{(g)}(\widetilde{\varphi})$ is regular (i.e. has a Taylor expansion with no negative powers of $\widetilde{\varphi}$. Equivalently,

$$F_{\text{Apparent}}^{(g \geq 2)}(t_{\text{App}}) = O(1). \tag{A.20}$$

This is similar to the gap condition (A.16) with $|G_c| = 0$, as first observed in [43].

**Expanding about infinity** In the main body of this paper, we consider the MUM point $\phi \equiv \frac{1}{729\varphi} = 0$ and attempt to extract an associated geometry from the genus 2,3,4, topological string free energies. Here we shall outline how to obtain those expansions. Our Yukawa coupling, in the $\phi$-coordinate, is

$$C_{\phi\phi\phi} = \left(\frac{d\varphi}{d\phi}\right)^3 C_{\varphi\varphi\varphi}(\varphi(\phi)) = \frac{-2 \cdot 3^9 \cdot (1 - 405\phi)}{\phi(1 - 27\phi)(1 + 19683\phi^2)}. \tag{A.21}$$

Again, we recompute the propagators as expansions in $\phi$. This time, to take the holomorphic limit we make the replacements

$$\mathcal{K}_{\phi} \mapsto K_{\phi} = -\partial_{\phi} \log\left(\varpi_0^{(H)}(\phi)\right), \quad \Upsilon_{\phi\phi}^{\phi} \mapsto \Gamma_{\phi\phi}^{\phi} = \frac{d\phi}{dt_{\infty}} \frac{d^2 t_{\infty}}{d\phi^2}, \quad \mathcal{S}^{\bullet}(\phi, \overline{\phi}) \mapsto S^{\bullet}(\phi). \tag{A.22}$$

The mirror map $t_{\infty}$ about $\phi = 0$ was given in (84). We proceed to use (A.4) to compute propagators and propagator ambiguities, but we stress again that $s_{\phi\phi}^{\phi} = -2/\phi$ is not zero in this coordinate patch. We arrive at

$$h_{\phi\phi} = \frac{5 - 3192\phi + 66582\phi^2 - 21257640\phi^3 + 215233605\phi^4}{2\phi^2(1 - 27\phi)(1 - 405\phi)(1 + 19683\phi^2)}, \tag{A.23}$$

$$h_{\phi} = \frac{\left(\begin{array}{c}-27 + 43627\phi - 25501080\phi^2 + 6584829552\phi^3 - 591177535446\phi^4 + 110490298672590\phi^5 - 3302124299123040\phi^6 \\ + 161786935677776040\phi^7 - 3495001967306666775\phi^8 + 6253943137374963375\phi^9\end{array}\right)}{104976\phi^3(1 - 27\phi)(1 - 405\phi)^4(1 + 19683\phi^2)},$$

$$S^{\phi\phi} = \frac{1}{13122} + \frac{175\phi}{6561} + \frac{51697\phi^2}{4374} + \frac{10311730\phi^3}{2187} + \frac{1389149402\phi^4}{729} + \frac{187498993498\phi^5}{243} + \ldots,$$

$$S^{\phi} = -\frac{1}{26244\phi} - \frac{7}{2187} + \frac{34799\phi}{8748} + \frac{8592379\phi^2}{2187} + \frac{1852036499\phi^3}{729} + \frac{343741776259\phi^4}{243} + \ldots,$$

$$S = \frac{5}{52488\phi^2} + \frac{359}{26244\phi} + \frac{39689}{17496} + \frac{3681935\phi}{2187} + \frac{2844013139\phi^2}{1458} + \frac{410700083963\phi^3}{243} + \ldots$$

It is reassuring to check that the propagator ambiguities $h_{\phi\phi}$ and $h_{\phi}$ recomputed in (A.23) match with those in (A.7) after a tensor transformation.

At this stage in the analysis, we already have the polynomials $P^{(g)}$ in (A.10) from solving the HAE. We now substitute into these the propagators expanded about $\phi = 0$ in (A.23).[22]

---

[22] We thank Emanuel Scheidegger for discussion on computationally efficient ways of doing this.

This provides us with the following expansions:

$$
\begin{aligned}
\mathcal{F}^{(2)}(\phi) &= \frac{89}{18895680\phi^2} + \frac{161}{1049760\phi} + \frac{15991}{1259712} + \frac{396197\phi}{787320} \\
&\quad + \frac{40078561\phi^2}{262440} + \frac{3732891737\phi^3}{87480} + \dots, \\
\mathcal{F}^{(3)}(\phi) &= \frac{169}{18744952939776\phi^4} + \frac{97}{520693137216\phi^3} \\
&\quad + \frac{56123}{1041386274432\phi^2} + \frac{4507549}{520693137216\phi} + \dots, \\
\mathcal{F}^{(4)}(\phi) &= \frac{7649}{110687072614083302400\phi^6} \\
&\quad + \frac{8153}{2049760603964505600\phi^5} + \frac{1403603}{2459712724757406720\phi^4} + \dots
\end{aligned}
\tag{A.24}
$$

It is of no concern that these B-model expansions are singular at $\phi = 0$. After multiplying by the suitable power of $\varpi_0^{(H)}$, given in equation (81), the resulting A-model expansion from (85) is regular. In the main body of this paper we discuss the fact that we cannot find a choice of $\eta$ in (85) such that (A.11) is reproduced.

**Torsion refined invariants**    In §3.4 we have used the expansions from earlier in this appendix to propose that the MUM point $\phi = 0$ is associated to a noncommutative resolution of the hypergeometric threefold $\mathbb{WP}^5_{111223}[4,6]$, with 63 nodal singularities.

Topological string free energies for a smooth target $\mathbb{WP}^5_{111223}[4,6]$ were computed in [41], and the GV invariants for this model are tabulated in [93]. This information can be combined with the free energies that we have computed for the target $Y$ (1), to obtain the $\mathbb{Z}_3$ refined invariants for the proposed noncommutative resolution that appear in the refined GV formula (97).

To obtain these refined invariants, one first needs to compute free energies for $X_{\text{def}}$. Although these are available at [93], it will make the refined computation of [4] clearer if we explain some aspects. The mirror of $\mathbb{WP}^5_{111223}[4,6]$ has Picard-Fuchs operator numbered 12 in [27]:

$$
\mathcal{L}^{\text{AESZ12}} = \theta^4 - 2^{10}3^3 z \left(\theta + \frac{1}{6}\right)\left(\theta + \frac{1}{4}\right)\left(\theta + \frac{3}{4}\right)\left(\theta + \frac{5}{6}\right), \qquad \theta = z\frac{\mathrm{d}}{\mathrm{d}z}.
\tag{A.25}
$$

It must be stressed that the $z$-plane is completely distinct from the $\varphi$-plane. About the MUM point $z = 0$ can be found a Frobenius basis of solutions, which includes

$$
\begin{aligned}
\varpi_0^{(\text{def})}(z) &= \sum_{n=0}^{\infty} \frac{(4n)!(6n)!}{(n!)^3(2n)!^2(3n)!} z^n \\
&= 1 + 720z + 5821200z^2 + 75473798400z^3 + O(z^4),
\end{aligned}
\tag{A.26}
$$

$$
\varpi_1^{(\text{def})}(z) = \varpi_0^{(\text{def})}\log(z) + 6144z + 54180000z^2 + 724290828800z^3 + O(z^4).
$$

These are used to define another mirror map,

$$
t_{\text{def}}(z) = \frac{1}{2\pi i}\frac{\varpi_1^{(\text{def})}(z)}{\varpi_0^{(\text{def})}(z)}.
\tag{A.27}
$$

The Yukawa coupling for this model is $C_{zzz} = \frac{2}{z^3(1-27648z)}$. One goes on to compute a distinct set of propagators and propagator ambiguities using (A.4), with $z$ instead of $\varphi$. To take the holomorphic limit, one effects

$$\mathcal{K}_z \mapsto K_z = -\partial_z \log\left(\varpi_0^{(\text{def})}(z)\right), \quad \Upsilon_{zz}^z \mapsto \Gamma_{zz}^z = \frac{dz}{dt_{\text{def}}}\frac{d^2 t_{\text{def}}}{dz^2}, \quad \mathcal{S}^\bullet(z,\bar{z}) \mapsto S^\bullet(z). \quad (A.28)$$

With the new propagators, propagator ambiguities, and $\chi = -156$, one proceeds to use (A.8) to compute B-model free energies $\mathcal{F}_{\text{def}}^{(g)}$. This time the ambiguity $f_{\text{def}}^{(g)}$ is constrained using the constant term and Castelnuovo data attached to the MUM point $z = 0$, and by applying the gap condition (A.16) at the conifold $z = 1/27648$. Note that, as discussed in greater length in [41], the A-model free energies $F_{\text{def}}^{(g)}$ are not regular when expanded about the point $z = \infty$.

By comparing with the GV formula (96), the A-model free energies $F_{\text{def}}^{(g)}(t_{\text{def}})$ provide GV invariants $n_\beta^{(g)}$ for the intersection $\mathbb{WP}_{111223}^5[4,6]$. However, we are identifying $F_{\text{def}}^{(g)}(t_{\text{def}})$ with $F_0^{(g)}$ in (97). This allows us to read off the sums of torsion refined invariants $n_{\beta,k}^{(g)}$ for $\hat{X}$. These obey the relation

$$n_{\beta,0}^{(g)} + n_{\beta,1}^{(g)} + n_{\beta,2}^{(g)} = n_\beta^{(g)} \quad \left(\text{of } \mathbb{WP}_{111223}^5[4,6]\right). \quad (A.29)$$

Note that $n_{\beta,2}^{(g)} = n_{\beta,1}^{(g)}$, so the left hand side of the above relation involves only two independent invariants in the combination $n_{\beta,0}^{(g)} + 2n_{\beta,1}^{(g)}$. We need one more set of relations in order to extract the $n_{\beta,l}^{(g)}$, and this is provided by equating the $k = 1$ expansion (97) with our A-model free energies $F^{(g)}(t_\infty)$, obtained by applying (85) to (A.24).

In short, we get the $n_{\beta,l}^{(g)}$ at each genus by comparing the expansions (97) with the equalities

$$
\begin{aligned}
F_0^{(g)}(t_{\text{def}}) &= F_{\text{def}}^{(g)}(t_{\text{def}}) = \left(\varpi_0^{(\text{def})}(z)\right)^{2g-2} \mathcal{F}_{\text{def}}^{(g)}(z)\Big|_{z=z(t_{\text{def}})}, \\
F_1^{(g)}(t_\infty) &= \left(\eta\varpi_0^{(H)}(\phi)\right)^{2g-2} \mathcal{F}^{(g)}(\phi)\Big|_{\phi=\phi(t_\infty)}.
\end{aligned}
\quad (A.30)
$$

This provides us with the refined invariants up to genus 4 listed in tables Table 3 and Table 4.

**Proceeding beyond genus 4**   We can go further, and obtain the rest of the invariants in our three tables Table 2, Table 3, and Table 4. To do this we need additional boundary conditions, as the constraints we have imposed so far do not allow us to go beyond genus 4. With our identification of $X_{\text{def}}$, we can impose new constraints in expansions about the MUM point $\phi = 0$.

The first of these is Castelnuovo vanishing of the refined invariants $n_{\beta,k}^{(g)}$. The authors of [4] identified that the refined invariants they computed vanished unless $g \leq g_{\text{Castelnuovo}}^{\text{def}}(\beta)$, where

$$g_{\text{Castelnuovo}}^{\text{def}}(\beta) = \left\lfloor \frac{\beta}{2} + \frac{\beta^2}{2\kappa_{111}^{(X_{\text{def}})}} \right\rfloor + 1. \quad (A.31)$$

It may be the case that more generally a stronger bound holds for $\beta < \kappa_{111}^{(X_{\text{def}})}$, as in the top line of (A.13). Deriving such a formula through the use of PT-GV relations and wall crossing formula [94–100], as carried out in [91] to obtain (A.13), would be of great interest. This is a moot point for our example, as $\kappa_{111}^{(X_{\text{def}})} = 2$ is too small for an equation like the top line in (A.13) to provide any new constraints beyond genus 4.

Incorporating (A.31) as additional boundary data allows us to expand to genus 10. For genus 11 we need one additional boundary datum. For this, we will use the constant term (111).

In fact, we wish to draw attention to the fact that the expansions at genera 5 to 10 that result from including (A.31) have the constant terms (111), without us imposing this on the expansions. Although seeing integral invariants is reassuring, it is certainly possible in several examples of compact CY3 to make mistakes in the topological string computations so as to produce incorrect invariants that are nonetheless integral. However, reproducing the precise rational numbers prescribed by (111) serves as a nontrivial check on our work. We also propose that this justifies our assumption of Castelnuovo vanishing in the $\phi = 0$ expansion, and further we propose that this serves as a justification of our use of the top line in (A.13) as boundary data in the $\varphi = 0$ expansions.

To illustrate this point, we display the explicit constant terms for genera 2 to 11 predicted by (111):

$$
\begin{aligned}
\Bigg\{ &-\frac{79}{720}, \ \frac{331}{181440}, \ -\frac{1339}{10886400}, \ \frac{5371}{31933440}, \\
&-\frac{14855809}{3923023104000}, \ \frac{59433601}{47076277248000}, \\
&-\frac{1244461403}{2134124568576000}, \ \frac{218365533465689}{613091306060513280000}, \\
&-\frac{3243582914672231}{11672315249998233600000}, \ \frac{298687259364134483}{110794357452364185600000} \Bigg\}.
\end{aligned}
\tag{A.32}
$$

**Holomorphic ambiguities at low genera**

$$
\begin{aligned}
f^{(2)}(\varphi) =\ &-\frac{137041}{93312} - \frac{491473\varphi}{6480} - \frac{190323\varphi^2}{640} - \frac{6125}{3(-5+9\varphi)^3} - \frac{245}{27(-5+9\varphi)^2} \\
&+ \frac{9629}{1458(-5+9\varphi)} + \frac{1}{720(-1+27\varphi)^2} + \frac{28}{1215(-1+27\varphi)} \\
&+ \frac{1-9\varphi}{360(1+27\varphi^2)^2} + \frac{-1633+14454\varphi}{19440(1+27\varphi^2)},
\end{aligned}
$$

$$
\begin{aligned}
f^{(3)}(\varphi) =\ &-\frac{142789749567391}{228562145280} - \frac{2299634741521\varphi}{2116316160} - \frac{1155612170827\varphi^2}{156764160} \\
&- \frac{2542146547\varphi^3}{107520} - \frac{6799792401\varphi^4}{143360} - \frac{60025000}{27(-5+9\varphi)^6} - \frac{2401000}{2187(-5+9\varphi)^5} \\
&- \frac{1588179425}{19683(-5+9\varphi)^4} - \frac{51391653415}{118098(-5+9\varphi)^3} - \frac{729703257365}{4960116(-5+9\varphi)^2} \\
&- \frac{2291052687907}{138883248(-5+9\varphi)} + \frac{1}{27216(-1+27\varphi)^4} + \frac{11}{30618(-1+27\varphi)^3} \\
&+ \frac{397}{393660(-1+27\varphi)^2} - \frac{5026769}{1249949232(-1+27\varphi)} + \frac{-1-9\varphi}{1134(1+27\varphi^2)^4} \\
&+ \frac{1388+12087\varphi}{102060(1+27\varphi^2)^3} + \frac{-1414423-11404647\varphi}{22044960(1+27\varphi^2)^2} \\
&+ \frac{-52996226-654442569\varphi}{308629440(1+27\varphi^2)},
\end{aligned}
\tag{A.33}
$$

$$
\begin{aligned}
f^{(4)}(\varphi) = {}& -\frac{8932977640164985400263}{10663795450183680} - \frac{4481112962456402323\varphi}{9873884676096} \\
& - \frac{13232775153891761567\varphi^2}{73139886489600} - \frac{780493034576907263\varphi^3}{1015831756800} \\
& - \frac{35655370262059649\varphi^4}{11147673600} - \frac{20994644970247\varphi^5}{2867200} \\
& - \frac{445119511290039\varphi^6}{45875200} - \frac{6470695000000}{2187(-5+9\varphi)^9} - \frac{461772325000}{6561(-5+9\varphi)^8} \\
& + \frac{7652059030000}{177147(-5+9\varphi)^7} - \frac{70961247022750}{177147(-5+9\varphi)^6} + \frac{973295402383745}{14348907(-5+9\varphi)^5} \\
& + \frac{17072897670818221}{172186884(-5+9\varphi)^4} - \frac{17061189586641186551}{506229438960(-5+9\varphi)^3} \\
& - \frac{460804513625302358159}{13229462671488(-5+9\varphi)^2} - \frac{856261990881250068875}{92606238700416(-5+9\varphi)} \\
& + \frac{1}{349920(-1+27\varphi)^6} + \frac{113}{4133430(-1+27\varphi)^5} + \frac{128357}{1388832480(-1+27\varphi)^4} \\
& + \frac{7285775033}{94496161939200(-1+27\varphi)^3} - \frac{2267912798387}{11906516404339200(-1+27\varphi)^2} \\
& + \frac{292633228959857}{250036844491123200(-1+27\varphi)} - \frac{2}{1215(1+27\varphi^2)^6} \\
& + \frac{719}{32805(1+27\varphi^2)^5} + \frac{-426113+2088\varphi}{4286520(1+27\varphi^2)^4} \\
& + \frac{264819741455-76489481547\varphi}{2624893387200(1+27\varphi^2)^3} + \frac{2236748471668+1138200226285\varphi}{5444223321600(1+27\varphi^2)^2} \\
& + \frac{1200731262939953+667122095825595\varphi}{685972138521600(1+27\varphi^2)} \, .
\end{aligned}
$$

# B  Tables of GV invariants (also included as ancillary files)

## B.1  GV invariants for $Y$ ($\varphi = 0$)

Table 2: GV invariants for $Y$ ($\mathbb{Z}_3$ quotient of (3,48) CICY).

| $\beta$ | $n_\beta^{(0)}$ | $n_\beta^{(1)}$ | $n_\beta^{(2)}$ | $n_\beta^{(3)}$ | $n_\beta^{(4)}$ |
|---|---|---|---|---|---|
| 1 | 36 | 0 | 0 | 0 | 0 |
| 2 | 117 | 21 | 0 | 0 | 0 |
| 3 | 708 | 252 | 0 | 0 | 0 |
| 4 | 4329 | 3909 | 0 | 0 | 0 |
| 5 | 36648 | 54780 | 1764 | 0 | 0 |
| 6 | 353481 | 799427 | 74742 | 0 | 0 |
| 7 | 3654036 | 12029976 | 2321712 | 6048 | 0 |
| 8 | 40662441 | 183856566 | 62658648 | 1409391 | 609 |
| 9 | 476686680 | 2849013696 | 1546292340 | 99423948 | 590728 |
| 10 | 5813813286 | 44651082186 | 35915621409 | 4750281117 | 111170412 |
| 11 | 73324427076 | 706137679380 | 798821464500 | 183683944260 | 10444072368 |
| 12 | 950505231633 | 11252074225511 | 17196367996830 | 6203747559594 | 677366032962 |
| 13 | 12607759651752 | 180449529859008 | 360898410897432 | 190511304105852 | 34863672200724 |
| 14 | 170555645873421 | 2909722672893741 | 7421796553634874 | 5449679369889894 | 1529796032277798 |
| 15 | 2346703420933176 | 47141157827832400 | 150114768226834008 | 147527713689350088 | 59686649776267320 |

| $\beta$ | $n_\beta^{(5)}$ | $n_\beta^{(6)}$ | $n_\beta^{(7)}$ | $n_\beta^{(8)}$ |
|---|---|---|---|---|
| 9 | 0 | 0 | 0 | 0 |
| 10 | 35649 | 189 | 0 | 0 |
| 11 | 85547916 | -2844 | 0 | 0 |
| 12 | 18534294279 | 54582342 | -13578 | 12 |
| 13 | 2098119061944 | 30574646352 | 34548876 | -10944 |
| 14 | 166399998654636 | 6118201061109 | 52393703445 | 20606400 |
| 15 | 10450906212081840 | 745857469998888 | 18215392085052 | 99155139996 |
| 16 | 555198061714577424 | 66644726190601284 | 3360649526440095 | 58348980183789 |
| 17 | 25976067372741777228 | 4788136603728615780 | 421615028305402260 | 15935196164602572 |
| 18 | 1099493387576181820440 | 292144244544201285729 | 40508686513504838691 | 2758271324963777787 |

| $\beta$ | $n_\beta^{(9)}$ | $n_\beta^{(10)}$ | $n_\beta^{(11)}$ |
|---|---|---|---|
| 13 | 0 | 0 | 0 |
| 14 | 585 | 0 | 0 |
| 15 | 19453764 | 24840 | 0 |
| 16 | 221381920638 | 54614700 | 585 |
| 17 | 209538919878228 | 607436912556 | 103914720 |
| 18 | 82239100740230895 | 867161858493894 | 2042690495325 |
| 19 | 19228772556752251572 | 472968655069196880 | 4198449734171712 |
| 20 | 3159692314819687461015 | 145942943002288020387 | 3078643370480377335 |
| 21 | 400420801496016622049400 | 30559908006207602166924 | 1224136746375402792432 |
| 22 | 414966904278877669793351601 | 4810181836820364733038669 | 320739808835683485479412 |
| 23 | 3659454660447389109193376508 | 606983765326851350266049292 | 61798070005274601128908656 |
| 24 | 282550863034156559331846319914 | 64145564135530412618834578777 | 9382145243514639230834803968 |

## B.2  Torsion refined GV invariants for $X$ ($\phi = 0$)

Table 3: $k = 0$ $\mathbb{Z}_3$ refined GV invariants $n_{\beta,k}^{(g)}$.

| $\beta$ | $n_{\beta,0}^{(0)}$ | $n_{\beta,0}^{(1)}$ | $n_{\beta,0}^{(2)}$ |
|---|---|---|---|
| 1 | 4968 | 8 | 0 |
| 2 | 9289674 | 87110 | 2 |
| 3 | 44627183136 | 1988858872 | 12311104 |
| 4 | 316892167738938 | 42243179822882 | 1500688167410 |
| 5 | 2789703751373878608 | 837382413457312496 | 88315126793702080 |
| 6 | 28134958395843520400736 | 15924461871881584311688 | 3651073534140119626324 |
| 7 | 311646240373418958207954888 | 295665327714977671674948272 | 123683496582390514093422672 |
| 8 | 3697182172381082654538499873482 | 541298117644503994316043768032 | 3694343545753423615445903557708 |

| $\beta$ | $n_{\beta,0}^{(3)}$ | $n_{\beta,0}^{(4)}$ | $n_{\beta,0}^{(5)}$ |
|---|---|---|---|
| 1 | 0 | 0 | 0 |
| 2 | 3 | 0 | 0 |
| 3 | -20080 | -48 | 0 |
| 4 | 5310609048 | -91074234 | 273385 |
| 5 | 3262260868246560 | 14049618917552 | -197505870928 |
| 6 | 391257647196544031155 | 17712247669915992540 | 200182327909764706 |
| 7 | 276089216795175895776148888 | 333609540319457847939889896 | 200962164405608233360088 |
| 8 | 145199224947764837600124461793 8 | 345762853196566001700519846216 | 4945770252199236246536921254 2 |

| $\beta$ | $n_{\beta,0}^{(6)}$ | $n_{\beta,0}^{(7)}$ | $n_{\beta,0}^{(8)}$ |
|---|---|---|---|
| 3 | 0 | 0 | 0 |
| 4 | 28 | 15 | 0 |
| 5 | 4948170224 | -49502888 | 49248 |
| 6 | -249309554327918 | 29065678237849 | -1108971050056 |
| 7 | 4818521064515699814816 | 31655445125885710576 | 186567957494010864 |
| 8 | 4025813847105326014230845224 | 16747517575870876080524032 0 | 3039952549070342755188114 |
| 9 | 1165179659210675423158701771081168 | 126339623722080969384628753918304 | 8124695925836068590817381871520 |

| $\beta$ | $n_{\beta,0}^{(9)}$ | $n_{\beta,0}^{(10)}$ | $n_{\beta,0}^{(11)}$ |
|---|---|---|---|
| 4 | 0 | 0 | 0 |
| 5 | 96 | 0 | 0 |
| 6 | 25588054752 | -280014490 | 638242 |
| 7 | -11821107361280424 | 680621621530728 | -28973551058312 |
| 8 | 19651331703947358377505 | -93643854358933003242 | 10644675479387239296 |
| 9 | 28530348325425331545781236700 0 | 486757737205743114481627090 4 | 315574572679820813553367 36 |

Table 4: $k = 1$ $\mathbb{Z}_3$ refined GV invariants $n^{(g)}_{\beta,k}$.

| $\beta$ | $n^{(0)}_{\beta,1}$ | $n^{(1)}_{\beta,1}$ | $n^{(2)}_{\beta,1}$ |
|---|---|---|---|
| 1 | 5292 | 0 | 0 |
| 2 | 9307251 | 85617 | 63 |
| 3 | 44628685776 | 1988587800 | 12332736 |
| 4 | 316892330863947 | 42243128282871 | 1500695836083 |
| 5 | 2789703772061681016 | 837382403244044628 | 88315129288115136 |
| 6 | 28134958398750564204744 | 15924461869815551672688 | 3651073534891268644446 |
| 7 | 311646240373856919785189244 | 295665327714557085817194828 | 123683496582600963009964440 |
| 8 | 3697182172381152009616186586499 | 541298117644495406466131849400 | 3694343545753479616486502355750 |

| $\beta$ | $n^{(3)}_{\beta,1}$ | $n^{(4)}_{\beta,1}$ | $n^{(5)}_{\beta,1}$ |
|---|---|---|---|
| 1 | 0 | 0 | 0 |
| 2 | 0 | 0 | 0 |
| 3 | -22176 | 0 | 0 |
| 4 | 5309642952 | -90959409 | 263340 |
| 5 | 3262260425227632 | 14049688655880 | -197516325708 |
| 6 | 3912576470057266674126 | 17712247709049044184 | 200182320547197480 |
| 7 | 276089216794445518389822188 | 3336095403214344327122856 | 2009621644010161109976852 |
| 8 | 145199224947762299807179753423 | 3457628531965749040091990199000 | 494577025219897616817812111041 |

| $\beta$ | $n^{(6)}_{\beta,1}$ | $n^{(7)}_{\beta,1}$ | $n^{(8)}_{\beta,1}$ |
|---|---|---|---|
| 3 | 0 | 0 | 0 |
| 4 | 504 | 0 | 0 |
| 5 | 4949529624 | -49628304 | 55440 |
| 6 | -249308142947631 | 29065407357870 | -1108925051130 |
| 7 | 4818521065551114692016 | 31655444874116277168 | 186568022627465184 |
| 8 | 402581384710601626247093130 | 167475175758521745808379400 | 303995254912716751066953 |
| 9 | 1165179659210675852290380918284400 | 12633962372208083877887282744095 | 8124695925836110763195577977928 |

| $\beta$ | $n^{(9)}_{\beta,1}$ | $n^{(10)}_{\beta,1}$ | $n^{(11)}_{\beta,1}$ |
|---|---|---|---|
| 4 | 0 | 0 | 0 |
| 5 | 0 | 0 | 0 |
| 6 | 25581955023 | -279444465 | 604044 |
| 7 | -11821123459037412 | 680625052290684 | -28974138084000 |
| 8 | 19651331684878750251828 | -93643847929875024135 | 10644673481386307847 |
| 9 | 2853034832542375672251396648 | 4867577372064075979498821612 | 3155745726508963516080084 |

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
