# Peer review of "Noncommutative resolutions and CICY quotients from a non-abelian GLSM"

_SciPost Physics, doi:SciPost Phys. 19, 156 (2025)_

## Round 1 · Referee Report · Anonymous (Referee 1) · 2025-7-4

Strengths

1 - The authors provide a GLSM with a phase that at the same time exhibits highly non-trivial gauge theory dynamics - a detailed understanding of which will hopefully be achieved in future work - as well as a compelling geometric interpretation in terms of a non-commutative resolution of a nodal degeneration of the complete intersection $\mathbb{P}^5_{111223}[4,6]$.
2 - They arrive at this interpretation by combining their GLSM analysis with techniques from mirror symmetry and novel results from topological string theory, thus allowing them to overcome the limitations of the gauge theory analysis.
3 - By providing the first example of a non-commutatively resolved Calabi-Yau threefold that admits both a GLSM realization and corresponds to a $\mathbb{Z}_3$ torsional B-field, the authors open up a new pathway for explicitly understanding the corresponding shaves of non-commutating algebras beyond the case of Clifford algebras that had been studied before.

Report

The authors study the GLSM that is associated to a $\mathbb{Z}_3$-quotient of a certain Calabi-Yau threefold $Y$ that is a complete intersection in $(\mathbb{P}^2)^3$, where the $\mathbb{Z}_3$ cyclically permutes the three $\mathbb{P}^2$ factors of the ambient space. The gauge group of the corresponding GLSM is non-abelian and a semi-direct product of $U(1)^3$ with $\mathbb{Z}_3$. In the phase where the FI-parameter $\zeta$ of the GLSM is large, the GLSM flows to a non-linear sigma model on said geometry.
The analysis of this phase is largely standard and the non-abelian gauge symmetry doesn't pose much of a problem. However, the authors observe that in the phase $\zeta\ll 0$ the dynamics of the GLSM are significantly harder to understand. While they observe that the phase appears to be hybrid like, fibered over a non-linear sigma model on $\mathbb{P}^2$, the existence of a non-compact Coulomb branch -- and of a cubic potential -- renders the dynamics of this phase inaccessible with standard techniques.

Leaving a detailed analysis of the gauge theory dynamics in this phase to future work, the authors then turn to mirror symmetry and topological string theory in order to understand the infrared fixed point.
The authors observe two points of maximally unipotent monodromy in the complex structure moduli space of the mirror of $Y$, one being mirror to the limit $\zeta\gg 0$ and the other to $\zeta\ll 0$.
By studying the integral structure of the periods of this mirror, as well as the topological string free energies, they argue that the IR theory in the phase $\zeta\ll 0$ should admit an interpretation as a non-linear sigma model on a singular degeneration $X$ of the complete intersection $\mathbb{P}_{111223}[4,6]$, with 63 isolated nodes, that supports a so-called ``fractional B-field'' -- and therefore should in turn admit an interpretation as a non-commutative resolution of $X$.
To this end they use modifications of the usual formulae for an integral period basis, as well as for the classical terms and the constant map contributions to the free energies, that have recently been proposed for topological strings on such non-commutative resolutions.
The authors fix the holomorphic ambiguity up to genus 11 and calculate the $\mathbb{Z}_3$-torsion refined Gopakumar-Vafa invariants that are encoded in the A-model topological string free energies of the non-commutative resolution of $X$ together with those of its smooth deformation.
The integrality of these invariants, together with the Castelnuovo vanishing and the correct constant map contributions provide highly non-trivial evidence that their interpretation is indeed correct.

The paper is well written and the exposition is very clear.
Moreover, only few examples of non-commutative resolutions of nodal Calabi-Yau threefolds have been discussed in the literature and the authors provide the first example that appears to admit a GLSM description and at the same time corresponds to a geometry with $\mathbb{Z}_3$-torsion.
Even though the analysis of the gauge theory dynamics in the relevant phase has so far largely been constrained to highlighting the associated difficulties, the authors make elegant use of mirror symmetry to find a convincing proposal for what the IR theory should be.
It will be interesting to see if subsequent work can elucidate how the gauge theory reproduces this proposal.

Requested changes

  • On p.7, the D-term equations also exclude solutions where, for example, $x_1=x_2=x_3=0$ but $(y_1,y_2,y_3)\ne 0\ne (z_1,z_2,z_3)$. Is it clear that (2.11) does not admit any solutions of this form? Of course this would affect the structure of the hybrid model in the phase $\zeta\ll 0$, leading to geometric "branches" in addition to the $\mathbb{P}^2$ that is parametrized by $p_1,p_2,p_3$.
  • On p.14, perhaps the authors could slightly elaborate on why solutions where additional matter fields become massless have to be discarded.
  • On p.16, tegarding the statement "having such a configuration at a point at infinity in the moduli space is a feature specific to non-abelian GLSMs.": For example, in 1305.5767 it has been observed that conifold transitions can occur both at phase limits and at phase boundaries, with the two sometimes appearing for different GLSM realisations of the same geometry. This already happens for abelian GLSM and one would naively expect that such GLSMs where the transition is located at a phase limit lead to counterexamples to the statements of the authors. Is the Coulomb branch still expected to be compact in those examples?
  • In the calculation of the hemisphere partition function in Section 2.5, it seems that the discrete part of the gauge group can essentially be ignored. Is it clear why this is the case?
  • On p.20, below eq.(3.5) the authors refer to "The first three solutions"while, given the order of the solutions in (3.5), the reader might easily be misled to consider those the first, the third and the fifth. Perhaps the exposition can slightly be improved to avoid any potential for misunderstanding.
  • On p.26, above eq.(3.26), is it clear that $c_2$ agrees for $X_{\text{def}}$, $X$ and $\widehat{X}$? How would one even define this quantity for the singular variety $X$?
  • It would be useful if, at least for very low genus, some of the holomorphic ambiguities of the free energies could be provided directly in the paper.

In terms of typos we have observed the following:

  • On p.1, "... turns out to be a very useful too ..."
  • On p.11, "The approach of ... is to use that fact that ..." presumably should read "... is to use the fact that ..."
  • On p.17, below eq.(2.61), "... but is can cancel ..."
  • On p.17, in the paragraph above eq.(2.63), "... it is fairly straightforward to this read off from ..."
  • On p.19, last sentence above Section 3.1, it should be either "one ... basis ... attached to each MUM point ...", or "two ... bases, one attached to each MUM point ...".
  • On p.19, "$U^1,....,W^1,W^3$" should be "$U^1,....,W^1,W^2$". Also, in the following these coordinates are sometimes used with upper and sometimes with lower indices. It seems that no problem would arise if lower indices were to be used throughout.
  • On p.24, in eq.(3.20) the sum over curve classes could either be over elements of $H_2$, in which case the upper limit $\infty$ should be dropped, or over natural numbers.
  • On p.26, below eq.(3.25), we are under the impression that, grammatically speaking, "obstruction" instead of "obstructions" would be in order here.
  • On p.28, in the sentence "If $\phi$ was an $N=1$ MUM point ..." shouldn't it read "If $\phi$ were an $N=1$ MUM point ..."?
  • On p.35, last paragraph, in the sentence "The existence of the Coulomb branch at infinity ..." the word "formulation" should be dropped.
  • On p.36, last paragraph of the conclusion, in the sentence "A final remark in relation ..." it seems to us that it should either by "concerns the" or "pertains to the".

Recommendation

Publish (surpasses expectations and criteria for this Journal; among top 10%)

---

## Round 1 · Referee Report · Anonymous (Referee 2) · 2025-9-21

Report

This journal's general acceptance criteria are definitely met. I recommend the manuscript for publication after a minor revision. For details see the attached report.

Attachment

Recommendation

Ask for minor revision

---

## Round 2 · Author Response

We thank both referees for their careful reading of our paper and for very insightful comments that have helped us improve our presentation.

Concerning the second report, we have implemented all requested changes.

Concerning the first report, we have the following additional remarks to the referee's comments, labelled in the order they appear in the report:

1) We have corrected our statement of the transversality condition in the discussion around (2.11). The correct statement is that the superpotential should be such that (2.11) only has solutions where any one of the triples $x_i, y_i, z_i$ vanish. Indeed, for generic polynomials $G_i$ there can be solutions to (2.11) of the form that the referee raises. However, the D-term equations impose |x| = |y| = |z|, so those solutions of (2.11) raised by the referee do not give classical vacua of our $U(1)^3\rtimes\mathbb{Z}_{3}$ model. We have added a footnote on page 11 that concerns the phase structure of the related $U(1)^3$ model with three independent FI parameters, where such geometric branches play an important role.

2) We have added clarifying remarks on p. 14 below (2.42) and reformulated the two paragraphs below (2.44) on p. 15. See also our response on the next request of the referee for further comments.

3) We have modified and extended the last paragraph on the old p. 16 (now first paragraph on p. 18). Here are some further comments which also relate to the referee’s previous request with regards to the Coulomb branch.

We agree that conifold transitions and phenomena that usually occur at phase boundaries can also occur at limiting points in phases, already in abelian GLSMs. However, we believe that the mechanism that generates the singular behaviour is different for the model we discuss in our paper and is indeed due to non-abelian dynamics. In our model, the strongly coupled phase has a non-compact Coulomb branch everywhere in the phase. This was first observed by Hori and Tong in hep-th/0609032 (see Section 4.2 which also applies to our model). Hori and Tong then argue that, due to dynamically generated masses, this Coulomb branch is lifted everywhere except at the limiting point in the phase. The Coulomb branch is not the Coulomb branch of the GLSM but the one of the SQCD in the IR. We have made modifications in the draft to clarify this.

We have considered the example presented in Section 2.5 of arXiv:1305.5767[hep-th] and did not find any indication of the same phenomenon happening. Nor did we find that the GLSM itself has a non-compact Coulomb branch in the “exoflop”-phase. Rather, in this model, and other exotic hybrid models, the source of the singularity seems to come from the Higgs branch. We note, however, that non-regular phases have so far only been described in one-parameter non-abelian GLSMs. A generalisation to multi-parameter models and a potential connection to exoflops is an interesting open question.

At this point, we do not have an argument that could exclude the existence of noncompact Coulomb branches at limiting point in phases in abelian GLSMs which is why we have slightly weakened our statement.

4) According to Hori-Romo arXiv:1308.2438[hep-th] Section 3.1.2, the derivation of the hemisphere partition function holds for any compact Lie group $G$. We have added this to the text above (2.58). The discrete part of the gauge group enters the hemisphere partition function in two ways:

(a) The overall normalisation, which depends on the Weyl group of the gauge group. While the normalisation of the hemisphere partition function is not fixed, the choice which is suggested by the topological data of the Calabi-Yau in the $\zeta>0$ - phase has a factor 1/3 (see above eq.(2.67)) which is expected from the Weyl group. See our comment in the paragraph between (2.66) and (2.67).

(b) The brane factors: The D-branes need to be gauge invariant. The discrete symmetry forbids certain matrix factorisations which would be allowed in its absence. For example, consider the superpotential

$W = (x_1)^3+ (x_2)^3$

and the associated matrix factorisation

$Q = (x_1 + x_2)\eta + (x_1^2− x_ 1 x_2 + x_2^2) \overline{\eta} $

($\eta, \overline{\eta}$ being the usual Clifford matrices).

Such a matrix factorisation will not be invariant under a $\mathbb{Z}_3$ which acts differently on $x_1$ and $x_2$. We have slightly extended footnote 14 on p. 20 to clarify this. In our example, we believe we have not missed any further contributions. For example, we found a consistent way to match the expected topological invariants of the large volume phase of our model by computing the hemisphere partition function for the brane associated to the structure sheaf.

6) We have altered our discussion of $c_2$ here. The fact that the leading term in the genus-1 free energy for X matches $c_2(X_{def})$ was observed in reference [2], where other techniques were available to fix this leading term. We have taken this as a basis for our assumption that in our example this leading term is $c_2(X_{def})$. We have stated this, with extra discussion, above (3.26) on page 28 (in the new version).

---

## Round 2 · List of Changes

A reference to the paper of Batyrev and van Straten was added, used below equation (1.2).

On the first paragraph of page 2, we add references [14-19] for other relevant GLSM constructions.

For the reader's convenience, we have added a diagram in Section 1.1 that collects the geometries we study, displaying their relations.

Around equation (2.11), we have modified our discussion of the transversality requirement. Also we have corrected a typo in (2.12), this does not affect our analysis or conclusions.

Above (2.24), we clarify which combination of $\sigma$-fields vanish.

After (2.25), we have modified our discussion in light of the above transversality condition, and added a footnote with further discussion in light of Referee 1's comments.

Clarifying remarks were added below (2.42) and (2.44), in light of Referee 1's comments.

We changed the wording below (3.5) to prevent confusion as to which solutions were being referred to.

Second paragraph of S3.2.2, we correct our phrasing on Calabi-Yau versus $c_1=0$.

Our discussion of $c_2$ above equation (3.26) has been altered, in line with Referee 1's comments.

Below (3.25), a reference to Namikawa-Steenbrink was added.

End of paragraph below (3.25), we added two relevant references to Braun-Kreuzer-Ovrut-Scheidegger.

At the top of page 30, we improve our phrasing concerning the Witten index.

In the second paragraph of section 4, we discuss briefly the applicability of our methods to other AESZ operators.

Below (A.20) we reference Hosono-Konishi for this gap observation.

Holomorphic ambiguities at genus 2, 3, and 4 have been included on page 49.

---

## Editorial Decision

published